# Inhibiting MARSs reduces hyperhomocysteinemia-associated neural tube and congenital heart defects

Xinyu Mei[1,2,3,4,*,†] (iD), Dashi Qi[1,5,†], Ting Zhang[1,6,†], Ying Zhao[1,2,7,†], Li Jin[1,2], Junli Hou[8], Jianhua Wang[6], Yan Lin[1,2,4], Yu Xue[9] (iD), Pingping Zhu[8], Zexian Liu[9] (iD), Lei Huang[1,2], Ji Nie[1,2], Wen Si[8], Jingyi Ma[10], Jianhong Ye[1,2], Richard H Finnell[11] (iD), Hexige Saiyin[1], Hongyan Wang[1,2,**] (iD), Jianyuan Zhao[1,2,4,***], Shimin Zhao[1,2,4,****] (iD) & Wei Xu[1,2,4,*****] (iD)

## Abstract

Hyperhomocysteinemia is a common metabolic disorder that imposes major adverse health consequences. Reducing homocysteine levels, however, is not always effective against hyperhomocysteinemia-associated pathologies. Herein, we report the potential roles of methionyl-tRNA synthetase (MARS)-generated homocysteine signals in neural tube defects (NTDs) and congenital heart defects (CHDs). Increased copy numbers of *MARS* and/or *MARS2* were detected in NTD and CHD patients. MARSs sense homocysteine and transmit its signal by inducing protein lysine (N)-homocysteinylation. Here, we identified hundreds of novel N-homocysteinylated proteins. N-homocysteinylation of superoxide dismutases (SOD1/2) provided new mechanistic insights for homocysteine-induced oxidative stress, apoptosis and Wnt signalling deregulation. Elevated MARS expression in developing and proliferating cells sensitizes them to the effects of homocysteine. Targeting MARSs using the homocysteine analogue acetyl homocysteine thioether (AHT) reversed MARS efficacy. AHT lowered NTD and CHD onsets in retinoic acid-induced and hyperhomocysteinemia-induced animal models without affecting homocysteine levels. We provide genetic and biochemical evidence to show that MARSs are previously overlooked genetic determinants and key pathological factors of hyperhomocysteinemia, and suggest that MARS inhibition represents an important medicinal approach for controlling hyperhomocysteinemia-associated diseases.

**Keywords** acetyl homocysteine thioether; methionyl-tRNA synthetase; neural tube defects; N-homocysteinylation; reactive oxygen species
**Subject Category** Neuroscience

## Introduction

Hyperhomocysteinemia is a common metabolic disorder that occurs either due to genetic alterations in the one-carbon metabolic pathway or insufficient availability or uptake of folate molecules; this disorder is associated with neural tube defects (NTDs) and congenital heart defects (CHDs), which are caused by maldevelopment of the neural tube or heart (Mills *et al*, 1995; Motulsky, 1996; Hobbs *et al*, 2005). Hyperhomocysteinemia is also associated with many major adverse health outcomes in adults, such as cardiovascular diseases (Wierzbicki, 2007), neurodegenerative diseases (Kruman *et al*, 2000; Mattson & Shea, 2003) and cancers (Wu & Wu, 2002). Clinical treatments are used to lower circulating homocysteine (Hcy) levels via combined folic acid and B-vitamin therapy, which prevents approximately 50–70% of NTD and CHD cases (Wallingford *et al*, 2013); however, this is ineffective against hyperhomocysteinemia-associated diseases in adults (Toole *et al*, 2004; Bonaa *et al*, 2006), and controversial tumour-promoting effects have even

1  State Key Lab of Genetic Engineering, School of Life Sciences and Institutes of Biomedical Sciences, Obstetrics & Gynecology Hospital of Fudan University, Shanghai, China
2  NHC Key Lab of Reproduction Regulation (Shanghai Institute of Planned Parenthood Research), Shanghai, China
3  Interdisciplinary Research Center on Biology and Chemistry, Shanghai Institute of Organic Chemistry, Chinese Academy of Sciences, Shanghai, China
4  Collaborative Innovation Center for Biotherapy, West China Hospital, Sichuan University, Chengdu, China
5  Shanghai Key Lab of Birth Defect, Children's Hospital of Fudan University, Shanghai, China
6  Capital Institute of Pediatrics, Beijing, China
7  Shanghai Laboratory Animal Research Center, Shanghai, China
8  Department of Chemistry, Fudan University, Shanghai, China
9  Department of Medical Engineering, College of Life Sciences and Technology, Huazhong University of Science and Technology, Wuhan, China
10 Ludwig Institute for Cancer Research, Nuffield Department of Clinical Medicine, University of Oxford, Oxford, UK
11 Center for Precision Environmental Health, Departments of Molecular and Human Genetics, Molecular & Cellular Biology and Medicine, Baylor College of Medicine, Houston, TX, USA
   *Corresponding author. Tel: +86 150 0068 3467; E-mail: mxy201512@163.com
   **Corresponding author. Tel: +86 139 1879 9082; E-mail: wanghy@fudan.edu.cn
   ***Corresponding author. Tel: +86 139 1760 3243; E-mail: zhaojy@fudan.edu.cn
   ****Corresponding author. Tel: +86 21 31246779; E-mail: zhaosm@fudan.edu.cn
   *****Corresponding author. Tel: +86 136 8169 8576; E-mail: xuwei_0706@fudan.edu.cn
   †These authors contributed equally to this work

been found in subclinical cancers (Kim, 2003; Ulrich & Potter, 2007). These downsides cast doubts about whether homocysteine is the main pathological factor in hyperhomocysteinemia-associated diseases and whether it represents the ideal target for therapeutic intervention. Missing factors, likely enzymes or metabolites related to Hcy metabolism, are most certainly involved in hyperhomocysteinemia-associated pathologies.

We recently reported that in addition to protein synthesis, tRNA synthetases exhibit additional functions to sense cognate amino acid levels, transmit amino acid signals to signalling networks and regulate various cellular functions (He *et al*, 2018). Remarkably, MARSs bind to Hcy, making them eligible as Hcy sensors. MARSs produce a reactive intermediate, Hcy thiolactone (HTL), through their error-editing activities and form N-homocysteinylation (N-Hcy) on lysines in proteins (Jakubowski *et al*, 2000). Accumulating evidence has shown that MARSs may be involved in hyperhomocysteinemia-associated pathologies (Perla-Kajan *et al*, 2007); for example, N-Hcy is known to induce inflammation (Capasso *et al*, 2012), cell death (Ferretti *et al*, 2004; Paoli *et al*, 2010), autoimmune responses (Undas *et al*, 2004, 2006; Jakubowski, 2005) and protein dysfunction (Kerkeni *et al*, 2006). Moreover, HTL is more toxic than Hcy to cultured human endothelial cells (Kerkeni *et al*, 2006) and mice (Borowczyk *et al*, 2012a,b). Furthermore, L-Hcy, L-HTL and D-HTL can modify lysine directly or be converted by MARSs into lysine modifiers, and they are toxic to developing rat embryos, whereas D-Hcy, which cannot be recognized by MARSs and therefore cannot be converted into D-HTL, is benign and harmless to developing embryos (Vanaerts *et al*, 1993).

MARS and MARS2 expression is elevated in cancer cells (Kushner *et al*, 1976). There are many similarities between tumour progression and embryo development, such as rapid cell proliferation and active differentiation (Pierce, 1983). As MARSs play critical roles in protein synthesis, we suspected that MARS and MARS2 expression could be elevated during embryo development. Notably, *MARS* and *MARS2* gene duplication has been associated with increased ROS (Bayat *et al*, 2012). Oxidative stress-induced aberrant apoptosis may result in NTDs (Yang *et al*, 2008) and CHDs (Morgan *et al*, 2008). When taken together, it is expected that the toxic effects of hyperhomocysteinemia will be more pronounced in proliferating cells due to high MARS levels. Indeed, several large randomized trials have shown that lowering Hcy is not always effective in improving proliferating independent hyperhomocysteinemia diseases, as stroke (Toole *et al*, 2004; Carlsson, 2007) and Alzheimer's disease (Sachdev, 2011). We therefore hypothesize that MARS levels play a critical role in regulating hyperhomocysteinemia toxicity. In the current study, we confirmed the pathological effects of MARSs in CHDs and NTDs using cells cultured *in vitro* and *in vivo* in animal models and human samples.

# Results

## Increased MARS copy numbers and expression levels are associated with NTDs and CHDs

Based on the hypothesis that MARSs sense and generate homocysteine signals and promote hyperhomocysteinemia-associated disease onset, we collected blood samples from clinically confirmed cases of

NTDs and CHDs (Appendix Table S1) and carried out copy number variation (CNV) analysis for MARS and MARS2-encoding genes. Genomic DNA was extracted at high purity (OD260/OD280 = 1.8–2.0) from 2 ml whole blood from each participant. The most commonly detected copy number frequency for both *MARS* and *MARS2* was three copies, although four copies were occasionally observed; these were found to be significantly more common in samples affected by NTDs and CHDs than in unaffected control samples. Among the controls, five samples were detected to have 3 *MARS* copies, while 16 and 5 NTD samples had three copies of *MARS* and *MARS2*, respectively (Table 1). Similarly, 15 and 9 CHD samples were identified to have 3–4 copies of *MARS* and *MARS2*, respectively (Table 2). These results demonstrated that 17 individuals who had three copies of one or both of MARS/MARS2 had a 4.45-fold increased risk of having NTDs (OR = 4.45, 95% CI = 1.54–15.72, $P = 0.002$, Fisher's test), while 21 individuals who had 3–4 copies of one or both of *MARS/MARS2* had a 12.38-fold increased risk of having CHDs (OR = 12.38, 95% CI = 4.36–43.41, $P = 2.22 \times 10^{-9}$, Fisher's test; Tables 1 and 2). These observations, the finding that all amplified MARSs carried no mutations, and no detection of MARS or MARS2 copy number loss in samples from either birth defect cohort all indicated that increased MARS or MARS2 copy numbers are positively associated with increased risks of either NTDs or CHDs. Increased copy numbers were often associated with elevated protein expression (Hyman *et al*, 2002; Heidenblad *et al*, 2005), and we found that both mRNA (Fig 1A and B) and protein expression levels (Fig 1C–F) increased proportionally with MARS and MARS2 copy numbers.

## MARS expression is up-regulated in developing organs and proliferating cells

The increased copy number of MARSs in NTDs and CHDs indicated that increased MARS levels may deregulate neural tube and heart development. This prompted us to investigate the MARS expression levels in developing tissues or proliferating cells. MARS and MARS2 were expressed in the proliferating cells of mice embryos both in the neural tube at embryonic stages E8.5–E10 (Figs 1G and EV1A) and hearts at E10 (Fig 1H). In embryonic monkey (Fig 1I) and human (Fig EV1B) brain sections, cells with higher MARS levels were

**Table 1. MARS and MARS2 copy number variation analysis in NTDs samples.**

| Genes | Copy Numbers[a] | Controls | NTDs | OR (95% CI) | P |
|---|---|---|---|---|---|
| MARS | 2 | 235 | 180 | | 0.003[b] |
| | 3 | 5 | 16 | 4.16 (1.42–14.81) | |
| MARS2 | 2 | 240 | 191 | | 0.018[c] |
| | 3 | 0 | 5 | NA | |
| ALL | 2 | 235 | 179 | | 0.002[b] |
| | 3 | 5 | 17 | 4.45 (1.54–15.72) | |

NA, not available.
[a]AccuCopy assay identified copy numbers of MARS and MARS2 were confirmed in control and NTD samples, respectively.
[b]P values were estimated using Chi-squared test.
[c]P values were estimated using Fisher test.

**Table 2. MARS and MARS2 copy number variation analysis in CHDs samples.**

| Genes | Copy Numbers[a] | Controls | CHDs | OR (95% CI) | P |
|---|---|---|---|---|---|
| MARS | 2 | 235 | 85 | | |
| | 3 or 4 | 5 | 15 | 8.23 (2.74–29.86) | 3.99E-06[b] |
| MARS2 | 2 | 240 | 91 | | |
| | 3 or 4 | 0 | 9 | NA | 1.27E-05[c] |
| ALL | 2 | 235 | 79 | | |
| | 3 or 4 | 5 | 21 | 12.38 (4.36–43.41) | 2.22E-09[b] |

NA, not available.
[a]AccuCopy assay identified copy numbers of MARS and MARS2 were confirmed in control and CHDs samples, respectively.
[b]P values were estimated using Chi-squared test.
[c]P values were estimated using Fisher test.

mainly present in two regions: the germinal layer and the pallium (outer circle). Most germinal layer cells are neural stem cells and neural precursor cells, while cells in the pallium are mostly immature neurons at this stage (Kriegstein & Alvarez-Buylla, 2009; Marin et al, 2010), and both have high proliferative activity (Fig EV1C). MARS and MARS2 levels in proliferating chorion cells derived from human embryos were higher than that in their corresponding maternal deciduae cells (Fig 1J–L). Moreover, MARS and MARS2 levels were higher in foetal rat brains than those in the brain of the corresponding mothers (Fig 1M–O). Furthermore, proliferating urothelial cell carcinoma (UCC) cells had elevated MARS levels compared with their corresponding adjacent noncancerous cells (Fig 1P and Appendix Fig S1), and the proliferation marker Ki67 (Scholzen & Gerdes, 2000) was co-over-expressed with MARS in hepatocellular carcinoma (HCC) cells (Fig 1Q). All of these observations were consistent with the notion that developing and proliferating cells have high MARS expression levels, possibly to meet elevated protein synthesis demands; however, this was inconsistent with the fact that MARS copy numbers were higher in NTDs and CHDs, which are diseases caused by maldevelopment.

## MARS and/or MARS2 over-expression potentiates Hcy to induce apoptosis

Next, the consequences of MARS over-expression were considered. As MARSs charge methionine (Met) for protein synthesis and transform Hcy to HTL (Fig 2A), we tested the effects of methionine and homocysteine on growth. Neuronal NE4C stem cell growth was inhibited by 0.5 mM Hcy, but not by 0.5 mM Met (Fig 2B). Furthermore, the effects of Hcy were abrogated by Bcl-2 over-expression, which prevents apoptosis (Swanton et al, 1999), as analysed by cleaved-caspase3 levels (Erhardt & Cooper, 1996; Fig 2B), suggesting that Hcy, but not Met, inhibits cell growth mainly by augmenting apoptosis. This notion was further confirmed by the finding that Hcy (Fig 2C and D), but not Met (Fig 2E), induced apoptosis in NE4C cells. Notably, over-expressing either MARS (Fig 2C) or MARS2 (Fig 2D) potentiated the abilities of Hcy, whereas simultaneous MARS and MARS2 knockdown weakened (Fig 2F) the ability of Hcy to induce apoptosis in NE4C cells, which demonstrated that

Hcy exerts its toxicity through MARSs and that MARSs over-expression potentiates Hcy to induce apoptosis. In order to profile the downstream metabolite signals affected by MARS over-expression, metabolomics analysis was applied. Among the enriched pathways, data showed that oxidative stress-related pathways were enriched in NE4C cells over-expressing MARS (Fig 2G). As oxidative stress also is related to Hcy-induced apoptosis and, consequently, the onset of NTDs and CHDs, we focused on the regulatory effect of MARS on oxidative stress.

## MARSs over-expression increases ROS by promoting N-Hcy

Notably, in NE4C cells, MARS or MARS2 over-expression increased cellular ROS (superoxide) levels (Fig 3A and B), whereas knockdown of MARS or MARS2 decreased these levels (Fig 3C and D). The variation in MARS-level-dependent cellular superoxide levels suggested that MARSs are upstream regulators of ROS. Moreover, homocysteine, but not methionine, induced MARS expression-dependent elevation of superoxide (Fig 3E). HTL is an Hcy metabolite produced by both MARS and MARS2 (Senger et al, 2001), and its levels were positively correlated with either MARS and MARS2 over-expression (Fig 3A and B) or knockdown (Fig 3C and D). It was found that HTL elevated cellular superoxide levels in a dose- (Fig 3F) and time-dependent (Fig EV2A). The above results imply that Hcy increases ROS through conversion to HTL induced by MARSs.

To examine whether ROS were regulated by N-Hcy, which is a spontaneous lysine modification induced by HTL proteins (Jakubowski et al, 2000), we generated a specific pan-anti-homocysteinyl-lysine antibody (α-N-Hcy) that allowed us to specifically monitor N-Hcy level changes in cells and isolated proteins (Appendix Fig S2A). We found that HTL treatment of NE4C cells dose-dependently increased N-Hcy protein levels in cell lysates (Fig 3H). In addition, N-Hcy levels were increased by MARS over-expression and decreased by MARS knockdown (Fig 3I), and MARS/MARS2 double knockdown abrogated the ability of Hcy, but not HTL, to elevate N-Hcy (Fig 3J) and superoxide levels (Fig 3K). These results confirmed that MARSs sense Hcy levels and increase ROS levels by modulating HTL production.

## MARS over-expression promotes N-Hcy-induced inactivation of SOD1/2

The concomitant fluctuations in MARSs, HTL, N-Hcy and superoxide concentrations prompted us to examine the intrinsic relationships among them. One possibility is that MARS over-expression promotes ROS accumulation by inducing hyper-N-Hcy. We performed a cell-wide proteomics survey to look for N-Hcy substrates by employing our previously reported affinity enrichment proteomic strategy (Appendix Fig S2B; Guan et al, 2010), which identified 2525 N-Hcy sites representing 870 different proteins here (Fig 4A and B, and Dataset EV1). Notably, N-Hcy heavily modified superoxide dismutase 1 and 2 (SOD1/2) (Table EV1 and Appendix Fig S3), which are two highly abundant key ROS-scavenging enzymes (Zelko et al, 2002) known to be regulated by lysine modifications (Qiu et al, 2010; Chen et al, 2011), suggesting that their activities are potentially regulated by N-Hcy. Incubating purified SOD1 or SOD2 with HTL in vitro (in

solution) increased N-Hcy levels but decreased their specific activities (Fig 4C), whereas exchanging N-Hcy substrate lysine (K) sites in SOD1 or SOD2 for unmodifiable tryptophan (W) (SOD1$^{3KW}$ and SOD2$^{5KW}$) to simulate the bulky side chain effects of N-Hcy rendered them less active (Fig 4D) and irresponsive to HTL-induced N-Hcy inactivation (Fig 4C). These treatments, as well as 20 μM Hcy treatment *in vitro*, had negligible effects on N-Hcy levels and SOD1 or SOD2 activities (Fig EV2C). This result confirmed that N-Hcy induced by HTL, rather than by disulphide bond formation between thiol groups of Hcy and proteins,

negatively regulates the activities of SOD1/2. Unlike in solution treatment, both HTL and Hcy treatments increased N-Hcy and decreased the specific activities of SOD1/2 in cells (Fig 4E and F). MARS and MARS2 single and double knockdowns inhibited HTL production, which resulted in decreased N-Hcy and increased specific activities of SOD1/SOD2 (Figs EV2D and E, and 4G). These results collectively suggested that MARS overexpression enhanced Hcy-induced ROS accumulation via the inactivation of SOD1/2 by modulating HTL production and inducing N-Hcy.

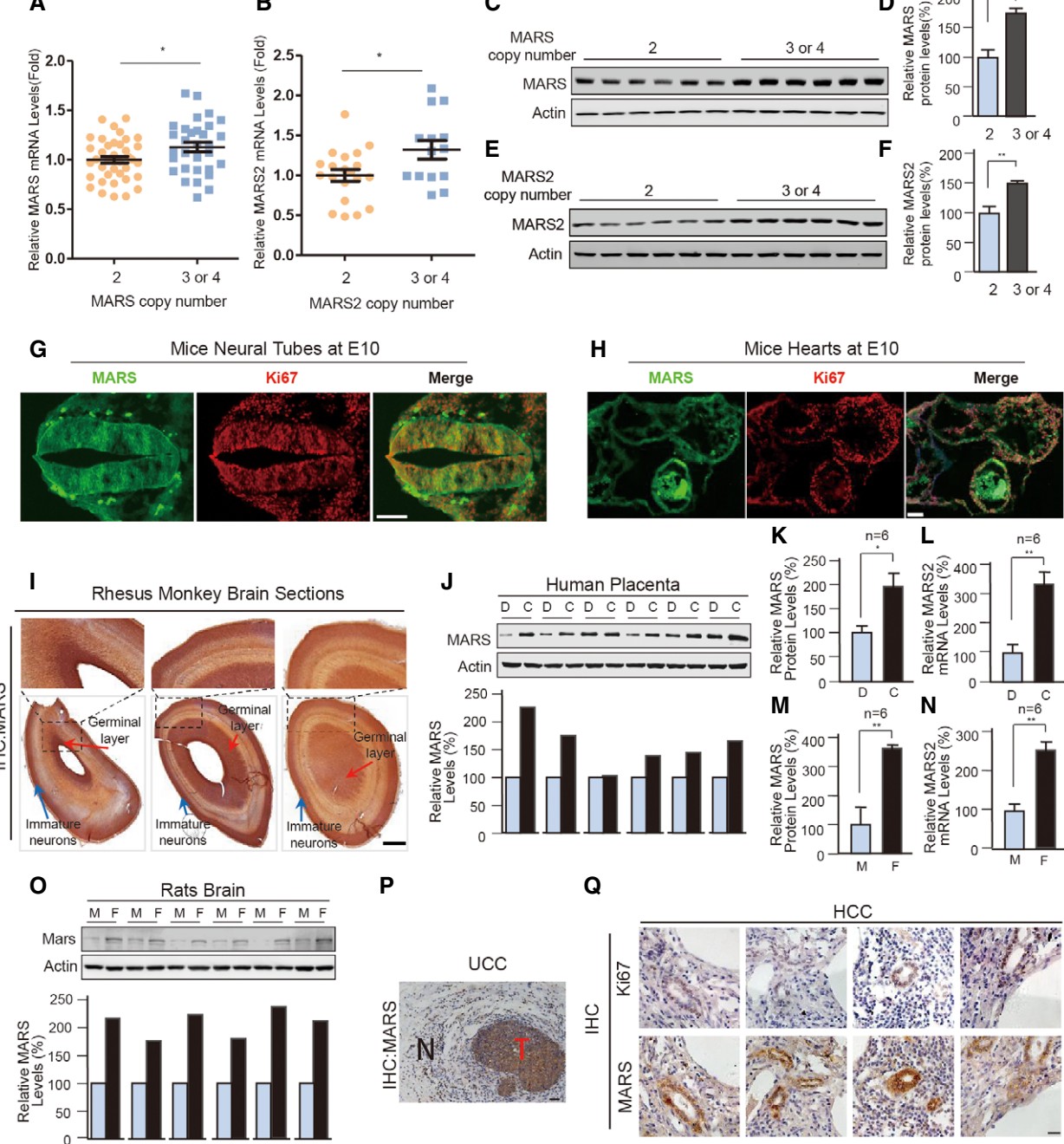

**Figure 1.**

**Figure 1. MARS expression levels were increased in samples with gains in copy numbers and developing/proliferating cells.**

A, B MARS (A) and MARS2 (B) mRNA expression levels were measured by real-time PCR in blood samples with 2 and 3–4 copies. Relative expression levels were normalized to GAPDH levels, and data were quantified relative to the mRNA expression levels of 2 copy samples.

C, D Immunoblotting analysis was performed to detect MARS protein levels in samples with 2 and 3–4 copies. Protein-level intensity shown in (D) was quantified using ImageJ and divided by the protein-level intensity of actin ($n = 6$). The results were then graphed relative to samples with 2 copies.

E, F MARS2 protein expression levels were detected (E) and quantified (F) ($n = 6$). Actin was used as a loading control.

G Cell proliferation was assessed by Ki67 (red) immunofluorescent staining in mice neural tube sections at E10. MARS (green) was co-stained. Scale bar: 150 μm.

H Double immunofluorescent staining for MARS (green) and Ki67 (red) in developing heart sections at E10. Scale bar: 150 μm.

I MARS expression was detected in E80 rhesus monkey brain sections ($n = 3$) by IHC. Local large magnification is shown as framed. Red arrow indicates germinal layer, and blue indicates immature neurons. Scale bar: 1mm.

J, K MARS protein levels in embryonic chorion (C) cells and their corresponding maternal deciduae (D) cells were determined by Western blot. Each D/C lane pair was quantified relative to D as shown at the bottom. The mean MARS levels in all D and C groups were quantified by ImageJ and shown in (K) ($n = 6$).

L MARS2 mRNA expression in embryonic chorion (C) and maternal deciduae (D) were detected by qPCR ($n = 6$).

M MARS protein quantification in the brains of foetal rats (F) and their corresponding mothers (M) from experiment (O) ($n = 6$).

N MARS mRNA levels ($n = 6$) in the brains of foetal rats (F) and their corresponding mothers (M) were determined by qPCR ($n = 6$).

K MARS protein levels in foetal rats (F) and their corresponding mothers (M) were determined by Western blot. Each M/F lane pair was quantified relative to M as shown at the bottom.

P Representative result of MARS expression was detected by IHC in tumour tissues (T) and adjacent noncancer tissues (N) of UCC samples. Scale bar: 50 μm.

Q Ki67 and MARS expression was analysed by IHC in HCC samples. Scale bar: 20 μm.

Data information: Data are presented as the mean ± SEM. *$P \leq 0.05$, **$P \leq 0.01$. Unpaired Student's *t* test was used for (A, B, D and F). Wilcoxon matched pairs test was used for (K, L, M and N).

Source data are available online for this figure.

## MARS over-expression activates Wnt/β-catenin signalling by increasing ROS

To confirm that MARSs sense Hcy and regulate NTDs, we further investigated the roles of MARSs in regulating the NTD- and CHD-associated Wnt signalling pathway (Hurlstone *et al*, 2003; Grigoryan *et al*, 2008; Gray *et al*, 2010). Both Hcy and HTL are capable of increasing a number of Wnt signalling associated markers, including β-catenin levels (Fig 5A and B), the activities of β-catenin-dependent Tcf transcription factors (Fig 5C) and its downstream targets of β-catenin in NE4C cells, namely c-Myc and cyclin D1 (Fig EV2F). This was consistent with the hypothesis that hyperhomocysteinemia causes Wnt signalling deregulation by promoting β-catenin accumulation (Han *et al*, 2009; Beard *et al*, 2011). MARS knockdown by *shRNA* abolished the ability of Hcy to increase β-catenin levels (Fig 5D) and partially abrogated the ability of Hcy, but not HTL, to activate TCF targets (Fig EV2F), suggesting that Hcy sensing by MARSs is essential for Wnt/β-catenin signalling, which in turn is required for normal neural tube and heart formation (Greene *et al*, 2009; Fossat *et al*, 2011; Zhao *et al*, 2014), regulation and likely for NTD development.

Moreover, Hcy and HTL treatments both decreased the interaction between dishevelled 1 (DVL1) and nucleoredoxin (NRX) (Fig 5E and F), a mechanism that is known to use ROS to activate Wnt/β-catenin signalling by decreasing the DVL1-NRX interaction (Funato *et al*, 2006). However, MARS knockdown blocked the ability of Hcy to decrease this interaction (Fig 5G). SOD1/2 over-expression not only partly reversed Hcy- or HTL-induced β-catenin accumulation (Fig 5H), but also abolished the ability of HTL to decrease the DVL1-NRX interaction (Fig 5I), which is similar to the effect of an antioxidant N-acetylcysteine (NAC) treatment (Issels *et al*, 1985; Fig EV2G and H). These results further confirmed that MARSs sense Hcy and abnormally activate Wnt/β-catenin signalling by accumulating ROS due to SOD1/2 inactivation induced by HTL production and N-homocysteinylation.

## Inhibiting homocysteine sensing by targeting MARSs

We tested the inhibition of Hcy sensing by targeting MARSs. First, acetyl Hcy thioether (AHT) was synthesized, which is an Hcy analogue intended to inhibit MARSs; however, it lacks a thiol-reducing moiety to prevent the complex from reducing the efficacy (Figs 6A and EV3A). AHT was found to be a potent MARS inhibitor that limited HTL production of MARS and MARS2 *in vitro* (Fig 6B), and the IC$_{50}$ for MARS and MARS2 was 36 and 40 μM, respectively, at an Hcy concentration of 20 μM (Fig 6C). AHT could effectively inhibit HTL production when administered to cells (Fig 6D), mice (Fig 6E) and rats (Fig EV3B). Furthermore, AHT treatment did not alter Hcy levels in NE4C or H9C2 (rat embryonic cardiomyoblast) cells (Fig 6F) and rat plasma (Fig EV3C). These results showed that AHT is a potent Hcy signal inhibitor via decreasing HTL production.

The ability of AHT to reverse the effects of Hcy was further tested. AHT dose-dependently inhibited N-Hcy levels in NE4C (Fig 6G and H) and H9C2 (Fig EV3D and E) cells. Consistently, AHT dose-dependently inhibited cellular superoxide levels when administered to NE4C (Fig 6I) and H9C2 cells (Fig EV3F), and AHT abolished the apoptosis-inducing ability of Hcy (Fig 6J). Moreover, AHT abolished the ability of Hcy, but not HTL, to increase β-catenin in NE4C cells (Fig 6K). These results collectively showed that AHT-inhibited Hcy sensing reversed the efficacy of Hcy.

## Inhibiting Hcy sensing decreases the prevalence of NTDs and CHDs

Finally, we tested whether interfering with Hcy sensing by MARSs would decrease the birth defect onset rate. All-trans retinoic acid (ATRA) NTD-complicated pregnancy rate (Li *et al*, 2012) and mouse models (Zhang *et al*, 2009) were employed. We found that ATRA increased Hcy levels in mice and rats livers (Fig EV4A and B), which confirmed that ATRA models are appropriate for simulating Hcy-induced congenital complications and in line with the results

that ATRA increased Hcy in human hepatocarcinoma cells (Xu et al, 2010). Moreover, the increase in MARS expression was enhanced by ATRA (Fig 7A–D), especially in neural epithelial cells. AHT suppressed the levels of N-Hcy (Fig 7A–D and Appendix Fig S4A), but not MARS (Fig 7A–D). These results showed that Hcy signals were elevated by ATRA and suppressed by AHT.

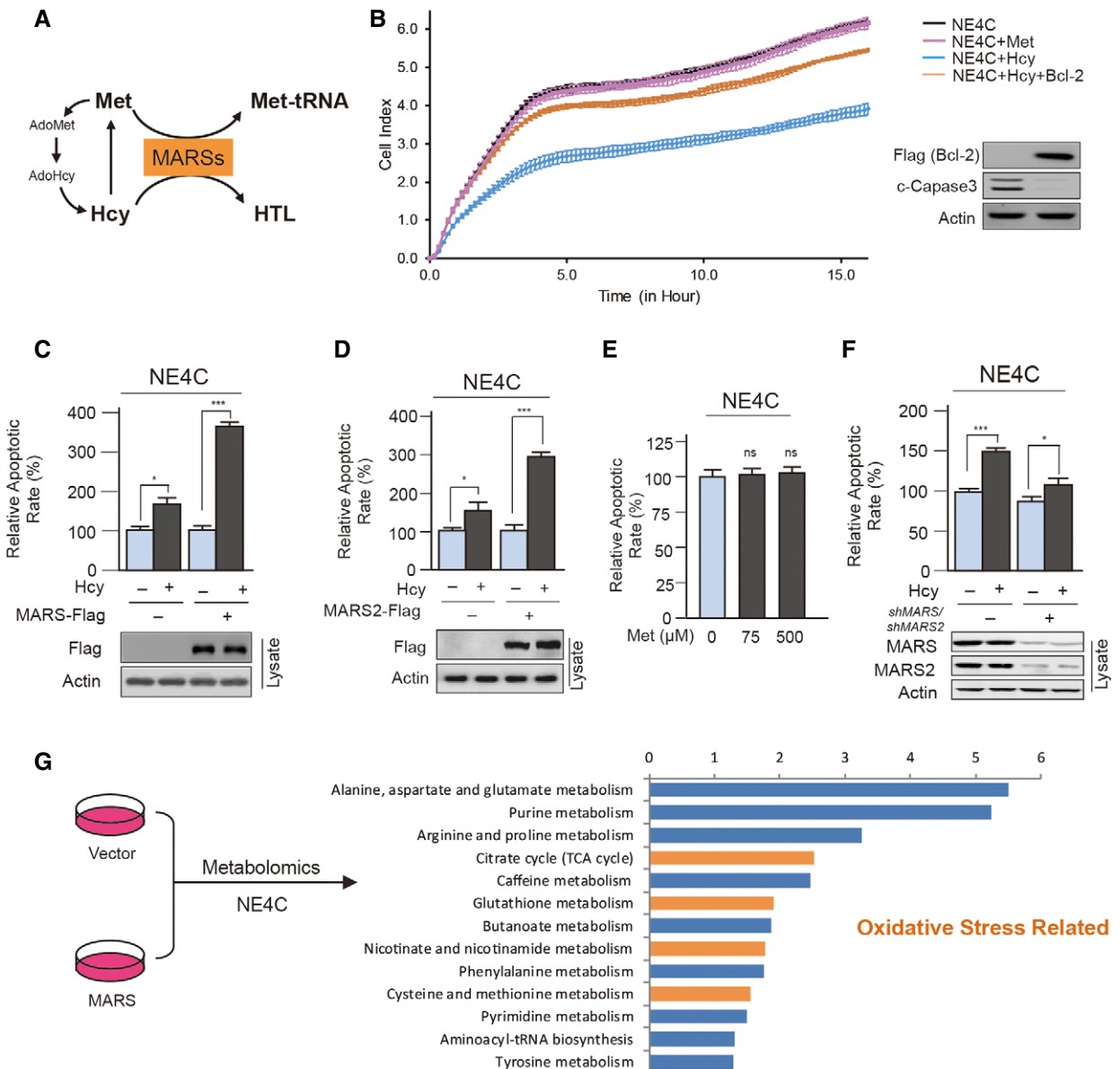

**Figure 2. MARSs potentiate Hcy-induced apoptosis and growth inhibition.**

A    Schematic representation of MARS functions.
B    Cell growth was determined in the absence and presence of 0.5 mM homocysteine or 0.5 mM methionine in the culture media of NE4C cells transfected with pcDNA3.1 vector or pcDNA3.1-Flag-Bcl-2 plasmids ($n = 3$). 5 μg plasmids were transfected in each $1 \times 10^6$ cells. The cell index responds to changes in cell number and cell adhesion. Cleaved-caspase3 (c-caspase3) levels were detected to determine apoptosis levels (right).
C, D    Apoptotic cells were detected by flow cytometry. The apoptotic rates of untreated, MARS (C)- and MARS2 (D)-over-expressing NE4C cells when cultured in the absence and presence of 20 μM homocysteine were normalized to the untreated NE4C cells ($n = 4$).
E    Cells were treated with different methionine concentrations as indicated. Apoptotic rates were detected by flow cytometry and normalized to those of the untreated cells ($n = 4$).
F    Apoptotic rates were determined in the absence and presence of 20 μM homocysteine in the culture media of control and MARS/MARS2 knockdown NE4C cells ($n = 4$).
G    Metabolic characterization of untreated and MARS-over-expressing NE4C cells based on pathway analysis of significantly changed metabolites. Columns show significantly enriched pathways, and yellow columns indicate oxidative stress-associated pathways.

Data information: Data are presented as the mean ± SEM and were compared using an unpaired Student's t test. [ns]not significant, $*P \leq 0.05$, $***P \leq 0.001$.
Source data are available online for this figure.

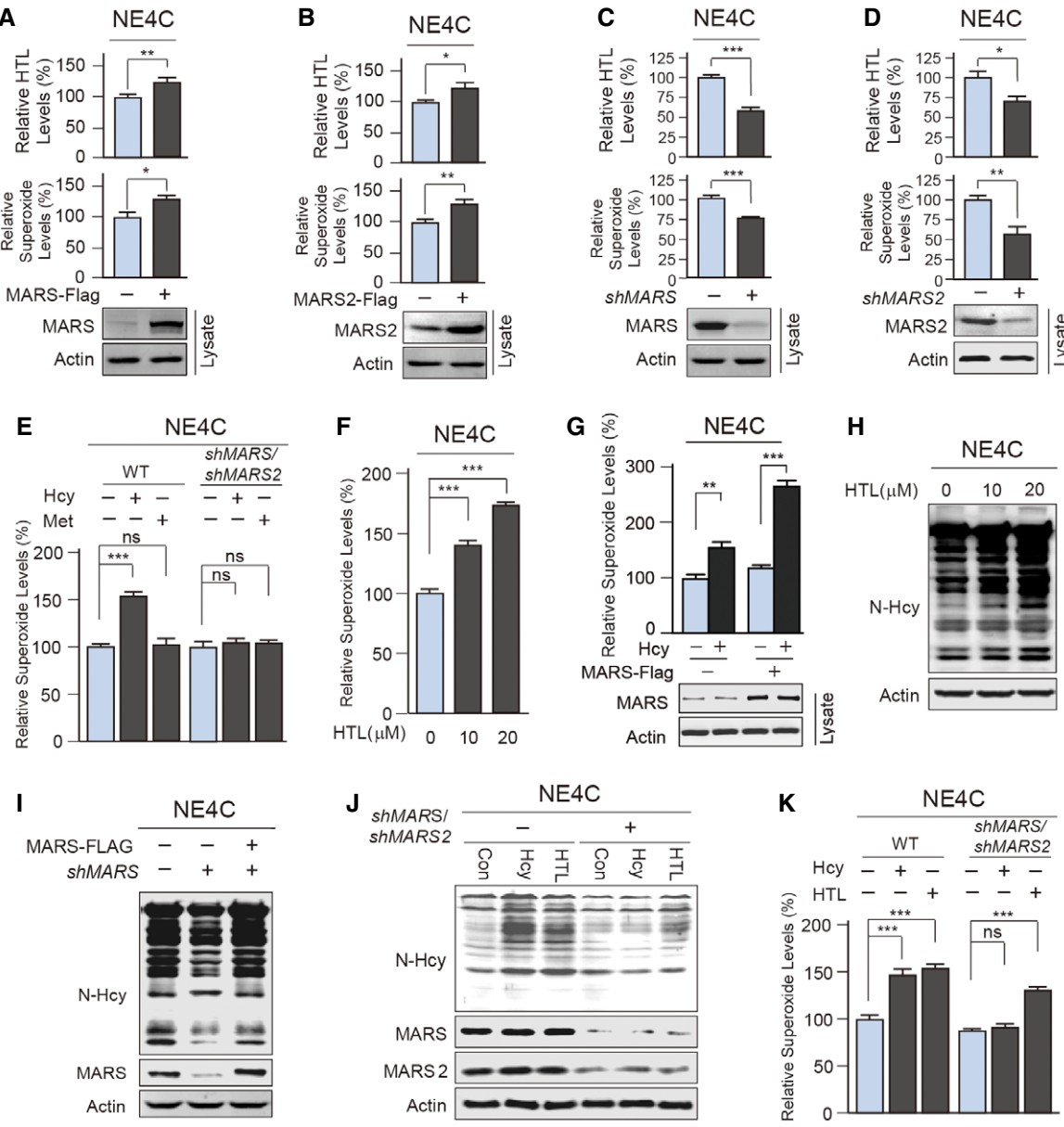

**Figure 3. MARS/MARS2 results of ROS accumulation by inducing N-Hcy.**

A, B   HTL and ROS (superoxide) levels were compared (*n* = 4) between NE4C cells transfected with the empty vector, MARS (A) or MARS2 (B) plasmids. Data were normalized against that of vector-transfected cells. HTL levels were determined by LC-MS, and ROS levels were determined by DHE staining assay.

C   MARS was knocked down by small hairpin RNAs in NE4C cells. MARS knockdown efficiencies were confirmed by Western blot. HTL and superoxide levels were quantified (*n* = 4) relative to vector-transfected NE4C cells. Cells were cultured in 20 μM Hcy-containing media.

D   Relative superoxide levels were determined by DHE staining assay and compared (*n* = 4) between NE4C cells with and without MARS knockdown by shRNA. Cells were cultured in 20 μM Hcy-containing media. MARS2 knockdown efficiencies were confirmed by Western blot.

E   NE4C cells with and without MARS knockdown were treated with either Hcy (20 μM) or Met (75 μM), and cellular superoxide levels were detected 6 h after the start of the respective treatment and quantified relative to the untreated NE4C cells (*n* = 4).

F   The cellular ROS (superoxide) levels were determined (*n* = 4) by DHE staining of NE4C cells treated with various HTL concentrations for 4 h. The ROS levels were normalized to those of untreated NE4C cells.

G   Superoxide levels in response to Hcy treatment (20 μM, 4 h) in MARS-expressing and control cells were determined relative to the untreated NE4C cells (*n* = 4).

H   NE4C cells were cultured in DMEM supplemented with different HTL levels as indicated. N-Hcy levels in HTL-treated and control cells were detected by Western blot.

I   Protein N-Hcy levels in response to MARS knockdown (by shRNA) and over-expression were detected by Western blot.

J, K   Untreated and MARS/MARS2 knockdown NE4C cells were treated with Hcy (20 μM) or HTL (10 μM), and the N-Hcy (J) and superoxide (K) levels (*n* = 4) in cells were detected 6 h after the start of the respective treatment and quantified relative to untreated NE4C cells.

Data information: Error bars indicate SEM. Data were compared using an unpaired Student's *t* test. ^nsnot significant, *P ≤ 0.05, **P ≤ 0.01, ***P ≤ 0.001. One-way ANOVA with Dunnett's correction was used for multiple comparisons.

Source data are available online for this figure.

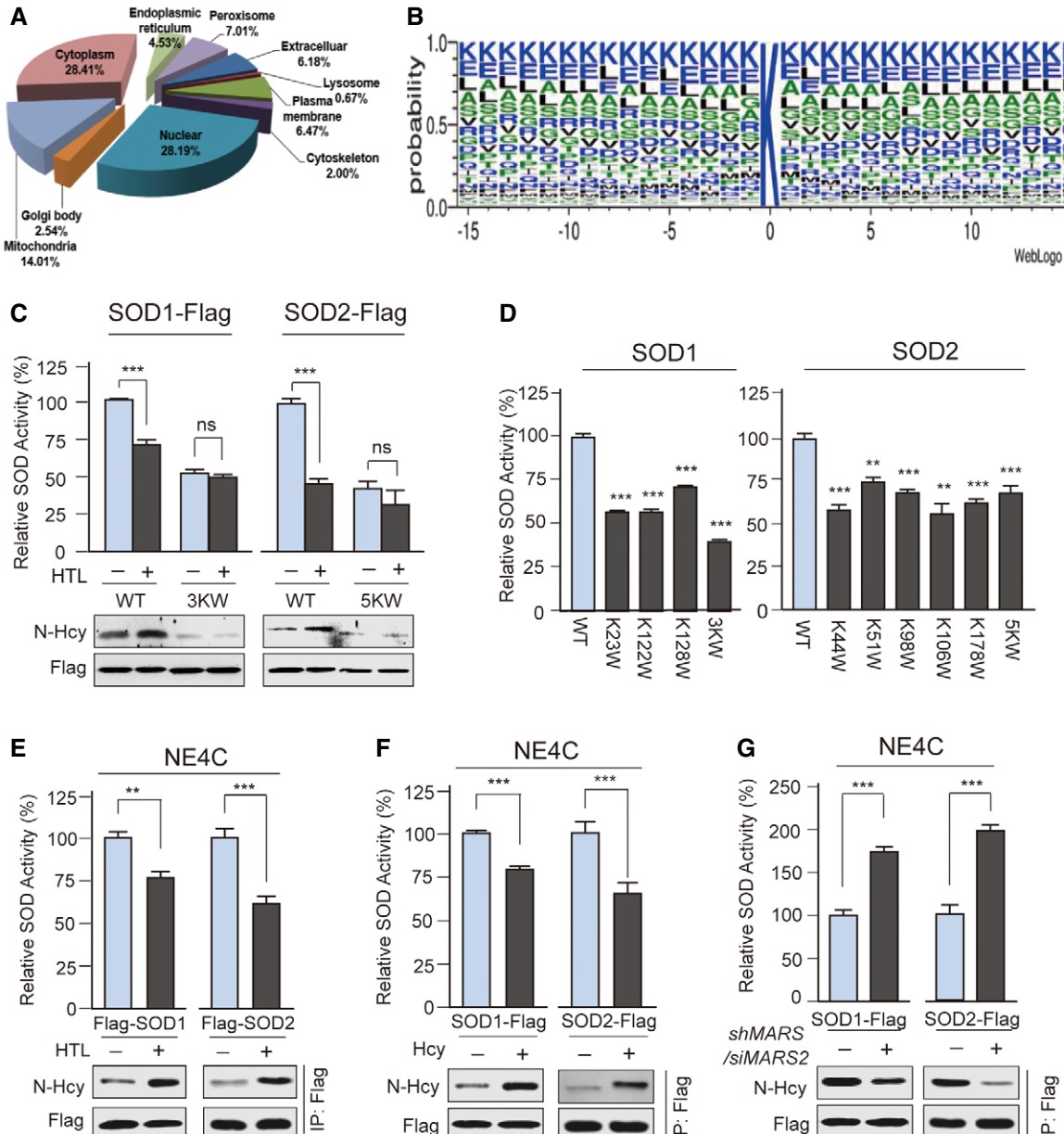

**Figure 4. HTL inactivates SOD1 and SOD2 through N-Hcy.**

A   Compartments of N-Hcy substrates identified by proteomic survey were predicted by WoLF PSORT.

B   WebLogo visualization of the amino acids spanning N-Hcy-modified lysine sites.

C   SOD activities were down-regulated by *in vitro* N-homocysteinylation. Flag-tagged SOD1, SOD1$^{K23W,K122W,K128W}$ (3KW) mutant, SOD2 and SOD2$^{K44W,\ K51W,K98W,K106W,}$ $^{K178W}$ (5KW) mutant were expressed in HEK293T cells, affinity-purified and incubated with or without 10 μM HTL in solution (*in vitro*). After 12 h, the reaction mixture was desalted and concentrated with filter tubes. N-Hcy levels and SOD activities were determined and normalized to untreated WT SOD1 and SOD2, respectively (*n* = 4).

D   Upon switching N-Hcy substrate lysine (K) sites to nonmodifiable tryptophan (W), the N-Hcy unmodifiable mutants displayed lower specific activity than the wide-type (WT) SOD. SOD1 (WT, K23W, K122W, K128W and 3KW) and SOD2 (WT, K44W, K51W, K98W, K106W, K178W and 5KW) were expressed in HEK293T cells. The catalytic activity of affinity-purified SOD proteins was determined and normalized to protein levels. Wild-type SOD1 and SOD2 activities were set at 100% (*n* = 4).

E, F   Hcy and HTL treatment in NE4C cells increased N-homocysteinylation of SODs and decreased their activities. Flag-tagged SOD1 and SOD2 were each ectopically expressed in NE4C cells. The cells were transferred to serum-free DMEM with or without 20 μM Hcy (E) or 10 μM HTL (F) 6 h before harvesting. After affinity purification, the specific activities of SOD1 and SOD2 were determined and normalized to SOD1 and SOD2 activity in untreated cells (*n* = 4).

G   N-Hcy SOD levels were down-regulated in MARS/MARS2 knockdown cells. Flag-tagged SOD1 and SOD2 were each over-expressed in untreated and MARS/MARS2 knockdown NE4C cells. After affinity purification, the N-Hcy levels and relative specific activities of purified SOD1 and SOD2 were determined and quantified relative to the NE4C cells (*n* = 4).

Data information: Data are presented as the mean ± SEM and were compared using an unpaired Student's *t* test. $^{ns}$not significant, **$P \leq 0.01$, ***$P \leq 0.001$. One-way ANOVA with Dunnett's correction was used for multiple comparisons.
Source data are available online for this figure.

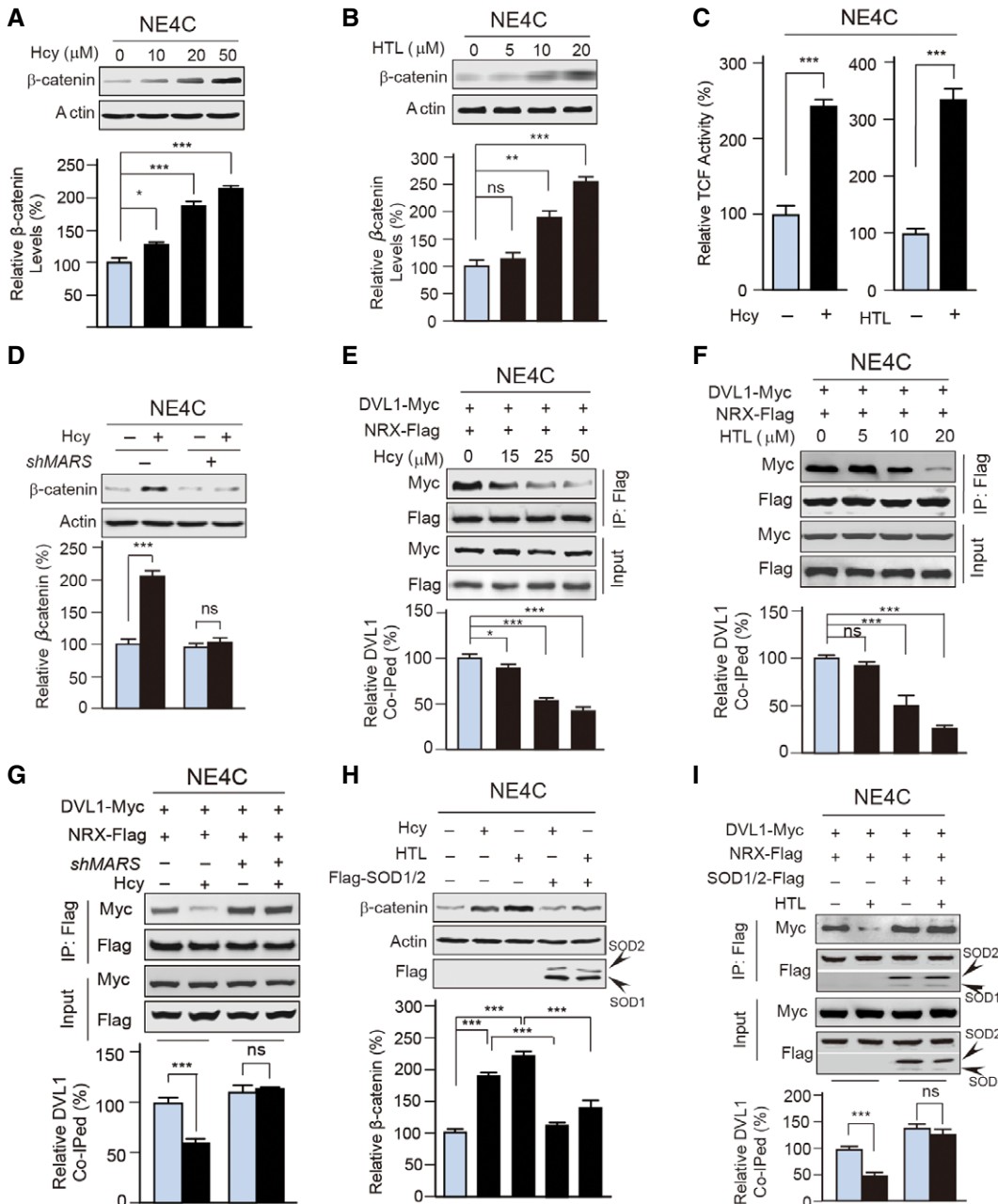

**Figure 5. N-Hcy activates Wnt signalling through ROS-induced disruption of DVL1-NRX interactions.**

A    Levels of β-catenin in response to Hcy treatment in NE4C cells were detected by Western blot (*n* = 3). Representative Western blots are shown, and band intensities were quantified relative to the untreated group.

B    Levels of β-catenin in response to HTL treatment in NE4C cells were detected. Band intensities were normalized to that of untreated cells (*n* = 4).

C    TCF activities were measured with a Topflash/Fopflash luciferase reporter system (*n* = 4). TCF activities responding to either 20 μM Hcy- or 10 μM HTL-treated were measured by quantifying Topflash/Fopflash relative to the untreated groups.

D    Levels of β-catenin in untreated and MARS knockdown NE4C cells were detected (*n* = 3) with the presence or absence of 20 μM Hcy in the culture media.

E, F    NE4C cells were co-transfected with NRX-Flag and DVL1-Myc plasmids and treated with different concentrations of Hcy (E) or HTL (F). The relative amount of DVL1 that co-immunoprecipitated (co-IPed) with NRX-Flag was determined (n=3).

G    NRX-Flag and DVL1-Myc were co-transfected in untreated or MARS knockdown NE4C cells. The relative DVL1 levels that co-IPed with NRX-Flag were determined and quantified (*n* = 3) for treated and control cells cultured with or without 20 μM Hcy.

H, I    Flag-tagged SOD1 and SOD2 were co-over-expressed in NE4C cells cultured in DMEM and supplemented with Hcy or HTL. (H) The levels of β-catenin in cells with and without SOD1/2 expression were determined (*n* = 3). (I) Relative levels of DVL1 that co-IPed with NRX-Flag were quantified in SOD-expressing and control cells cultured with or without 10 μM HTL (*n* = 3).

Data information: Data are presented as the mean ± SEM and were compared using an unpaired Student's *t* test. [ns]not significant, *$P \leq 0.05$, **$P \leq 0.01$, ***$P \leq 0.001$. One-way ANOVA with Dunnett's correction was used for multiple comparisons.

Source data are available online for this figure.

All-trans retinoic acid administration to dams typically resulted in NTDs such as spina bifida and exencephaly (Figs 7E and EV4C) and CHDs such as transposition of the great arteries (TGA), double outlet right ventricle (DORV) defect (Fig 7F), single ventricle defects (SVD) and ventricular septal defects (VSD; Fig 7G).

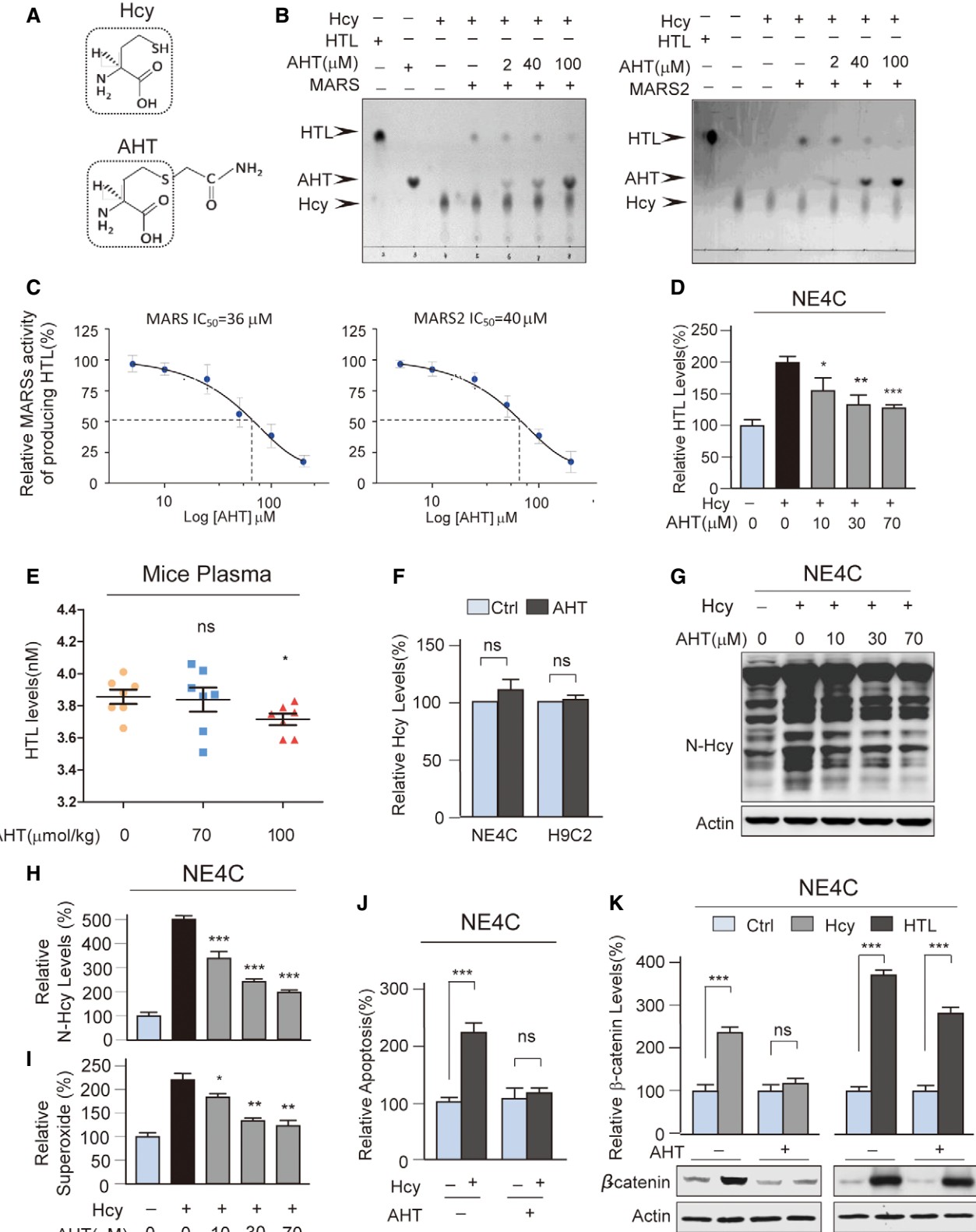

Figure 6.

Figure 6. AHT inhibits Hcy sensing.

A    Chemical structures of Hcy and its structural analogue acetyl Hcy thioether (AHT) are shown. The analogous parts are circled with dashed lines.
B    MARS- or MARS2-catalysed HTL production from Hcy (20 μM) was carried out *in vitro* in the presence of different AHT concentrations. The amount of HTL produced in each reaction was determined by thin layer chromatography.
C    MARS and MARS2 activities were determined ($n = 4$) in the presence of the indicated AHT concentrations in the reactions. AHT $IC_{50}$ values were determined for MARS and MARS2 with 20 μM Hcy.
D    Relative HTL levels were determined ($n = 4$) for NE4C cells treated with various AHT concentrations for 4 h.
E    AHT (71 or 110 μg/kg per day) was injected into C57BL/6 mice for 6 days. Plasma HTL levels were detected by LC-MS.
F    Hcy levels were compared in NE4C or H9C2 cells without and with 100 μM AHT treatments ($n = 6$).
G, H  NE4C cells were cultured in DMEM with or without 20 μM Hcy and supplemented with the indicated AHT levels; N-homocysteinylation levels were detected by Western blot. Band intensities were quantified relative to the untreated group using ImageJ (H) ($n = 4$).
I    Cellular superoxide levels were determined ($n = 4$) for NE4C cells treated with various AHT concentrations for 4 h. Superoxide levels were normalized to those of untreated NE4C cells.
J    Apoptotic rates were determined in the absence or presence of 20 μM Hcy in the culture media in control and 70 μM AHT-treated NE4C cells ($n = 4$).
K    Levels of β-catenin in untreated, Hcy- or HTL-treated NE4C cells were detected ($n = 3$) with the presence or absence of 70 μM AHT in the culture media. Mean β-catenin levels in untreated NE4C cells were set at 100%.

Data information: Data are presented as the mean ± SEM and were compared using an unpaired Student's *t* test. [ns]not significant, *$P ≤ 0.05$, **$P ≤ 0.01$, ***$P ≤ 0.001$. One-way ANOVA with Dunnett's correction was used for multiple comparisons.
Source data are available online for this figure.

In rats, we found that a daily dose of 16 mg/kg AHT significantly decreased ATRA-induced NTD rates (Table EV2), and maternal rat liver N-Hcy levels were positively correlated to the NTD rates (Fig EV4D), which strongly suggested that inhibiting MARS-mediated Hcy signals was effective for preventing NTDs. Notably, AHT decreased superoxide (Fig EV4E) and β-catenin (Appendix Fig S4B) levels similar to those with NAC treatment, supporting the notion that AHT decreases Hcy signalling and alters ROS-mediated β-catenin signalling.

In C57BL/6 mice, AHT treatments also significantly decreased ATRA-induced NTD (Table 3) and CHD (Table 4) rates, and maternal liver N-Hcy levels in mice were positively correlated to the NTD rates (Fig 7H). The recovery of ATRA effects in mice suggested again that increased Hcy signals are causal to NTDs and CHDs. Moreover, AHT treatment induced decreased N-Hcy levels and activated SOD1/SOD2 in neural tubes (Fig 7I), suggesting that the reduction of superoxides by SOD1/SOD2 N-Hcy inhibition governs the efficacy of AHT to reduce NTDs. This notion was further supported by the findings that AHT also induced similar effects in foetal rat neural tubes (Fig EV4F), that NAC exhibited similar NTD- and CHD-reducing abilities when employed to treat ATRA-induced NTDs and CHDs (Tables 3 and 4, and Table EV2), and that AHT decreased superoxide (Fig 7J and K), apoptosis and β-catenin levels in neuron tubes (Figs 7J, L and M, and EV4G–I) and hearts (Fig EV4J–L), similar to the results with NAC.

Finally, as hyperhomocysteinemia NTD and CHD models are available only in chicken embryos, but not rats or mice (Rosenquist *et al*, 1996; Kobus-Bianchini *et al*, 2017), we tested whether inhibiting Hcy sensing by targeting MARS could lower the onset of NTDs and CHDs in chicken embryos. It was found that Hcy treatment induced NTDs and CHDs (Fig 7N), and AHT decreased protein N-Hcy levels (Fig 7O). Further, AHT and NAC treatment decrease superoxide levels (Fig 7P and Q) and NTD and CHD rates (Tables 5 and 6). These results directly showed that targeting MARS is an effective way to reduce NTDs and CHDs induced by hyperhomocysteinemia.

## Discussion

In the current study, by employing clinical samples, animal models and cultured cells, we provided genetic and biochemical evidence to show that the MARS-mediated homocysteine signal plays indispensable roles in the pathology of NTDs and CHDs. Increased MARS or MARS2-encoding gene copy numbers are more frequently detected in NTDs and CHDs. Inhibiting homocysteine sensing by targeting MARS lowers NTD and CHD onsets without influencing homocysteine levels. These results demonstrate that MARSs are major factors in hyperhomocysteinemia, and perhaps even the principal pathological factor for this condition.

The inclusion of MARSs in the consideration of hyperhomocysteinemia pathology opens doors to understanding some previously puzzling observations. The paradoxical observations of homocysteine-lowing treatments may be due to differing MARS levels in the population. Our data showed that developing and proliferating cells, such as those in embryos or cancer cells, have higher MARS levels than in cells in adult tissues. The high MARS levels enhance sensitivity to changes in homocysteine levels in these developing and proliferating cells. Although hyperhomocysteinemia is associated with many major diseases, it consistently requires a more severe homocysteine elevation (> 150 μM) to induce adult diseases such as neurodegenerative disorders (Kruman *et al*, 2000; Mattson & Shea, 2003), whereas birth defects are associated with even a very slight Hcy elevation (8–15 μM) (Mills *et al*, 1995; van der Put *et al*, 1998; Bakker *et al*, 2009; Peker *et al*, 2016). This notion is substantiated by the finding that folate supplementation failed to prevent cardiovascular diseases (Wierzbicki, 2007) or other adult disorders (Brattstrom *et al*, 1998) and promoted the growth of preneoplastic cells and subclinical cancers (Kim, 2003; Ulrich & Potter, 2007), but prevented birth defects. Moreover, the efficacy of folate supplementation varies in preventing birth defects, perhaps due to differing MARS backgrounds in the subjects.

*MARS2* gene duplication has been associated with increased ROS (Bayat *et al*, 2012), and the promotion of oxidative stress has long been observed in hyperhomocysteinemia (Kanani *et al*, 1999). Our study provided a molecular mechanism for such observations, as MARSs exert their effects by modulating N-homocysteinylation, which regulates SOD activities and ROS levels.

While we showed that MARS-regulated ROS levels are critical in hyperhomocysteinemia pathology, we cannot conclude that SODs are the sole target of homocysteine signals. While ROS induced by altered metabolism, such as maternal diabetes, folate

deficiency, retinoic acid and some drugs, may be sufficient to cause NTDs and CHDs, there may be other genetic and environmental causes of NTDs and CHDs that do not increase ROS. It is likely that other downstream targets of Hcy signalling may also contribute to hyperhomocysteinemia pathologies to some extent, as we have now identified that approximately 1,000 proteins are

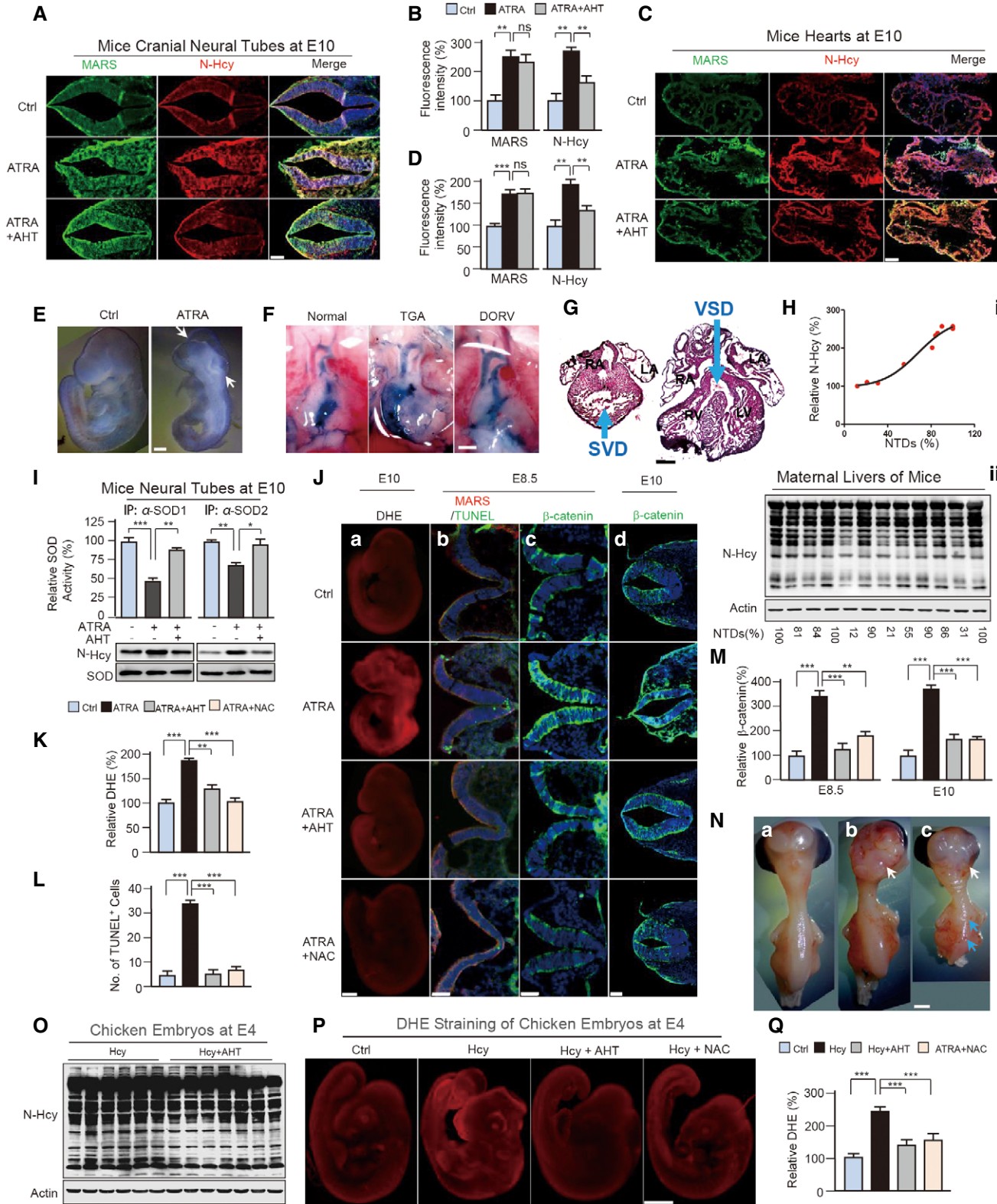

**Figure 7.**

**Figure 7. Hcy sensing inhibition by MARSs prevented NTD and CHD onset.**

A, B  Microscope images of E10 neural tube sections with the indicated treatments were stained with antibodies against MARS (green) and N-Hcy (red). Scale bar: 150 μm. The relative (to untreated) fluorescence intensities were quantified (n = 3) and are presented in (B).

C, D  Double immunofluorescent staining (C) for MARS (green) and N-Hcy (red) in E10 heart sections. Scale bar: 150 μm. The relative (to untreated) fluorescence intensities were quantified (n = 3) and are presented in (D).

E  Goss morphology of E10.5 ATRA-treated and untreated mice embryos. The ATRA-treated embryo exhibits an open neural tube (indicated by white arrows) compared to littermate controls. Scale bar: 500 μm.

F  Examples of normally developed and maldeveloped mice hearts: in the normal heart (i), the ink travelled from the right ventricle (RV) into the pulmonary trunk (PT); however, in ATRA-treated group hearts, ink travelled into the aorta (Ao) (ii, transposition of great arteries (TGA)) or both the PT and Ao (iii, double outlet right ventricle (DORV)). Scale bar: 1,000 μm.

G  Cardiac malformations were analysed by haematoxylin and eosin (H&E) staining of heart sections from E14.5 mice. The cardiac defects observed in ATRA-treated mice are as follows: ventricular septal defect (VSD) and single ventricle defects (SVD). Heart compartments are marked: RA: right atria, RV: right ventricle, LA: left atria, LV: left ventricle. Scale bar: 1,000 μm

H  Total protein N-Hcy levels in maternal mice liver homogenates were determined and quantified in correlation to the corresponding prevalence of NTDs in affected foetuses (i). Western blot was used to detect N-Hcy levels (ii).

I  Neural tube tissues at E10 were homogenized from litters of untreated, ATRA-treated and ATRA + AHT co-treated pregnant mice. Neural tube homogenates of each group were mixed from 6 embryos. Endogenous SOD1 and SOD2 were affinity-purified from the homogenates. N-Hcy levels and the specific activities of each purified SOD1 and SOD2 were determined. The SOD activities of the untreated group were set at 100%; error bar indicates assay replicates (n = 4).

J  Superoxide (a), apoptosis (b) and Wnt signalling (c, d) were over-activated in ATRA-treated mice neural tubes and restored by AHT or NAC treatment. (a) Whole-mount DHE staining of E10.5 mice embryos. Scale bar: 500 μm. Neural tube defect (white arrows) and heart (green arrow) regions show especially higher fluorescence intensities. (b) TUNEL (green) staining detected apoptotic cell death in E8.5 neural tube (E) sections. MARS (red). Scale bar: 300 μm. (c, d) IHC staining for β-catenin (green) in E8.5 (c) and E10 (d) mice neural tube sections. Scale bar: 300 μm (c), 150 μm (d). DAPI (blue).

K  Average (n = 3) DHE fluorescence intensities in experiment (J) (a) were quantified relative to the untreated group.

L  Quantification of TUNEL-positive cells in experiment (J) (b). Three sections from three embryos were pooled for analysis.

M  Quantification of β-catenin fluorescence intensities in experiment (J) (b) and (J) (c). The average (n = 3) β-catenin level in the untreated group was set at 100%.

N  Hcy-treated eggs exhibited NTD phenotypes at embryonic stage 8 (E8). A normal embryo (a) and neural tube defect phenotypes (b, c) are shown. Exencephaly (white arrows) and spina bifida (blue arrows) are indicated. Scale bar: 500 μm.

O  Total protein N-Hcy levels of 72-h chicken embryo homogenates from Hcy and Hcy + AHT-treated groups were detected by Western blot.

P  Whole-mount DHE staining of 72 h chicken embryos from untreated, Hcy-, Hcy + AHT- and Hcy + NAC-treated groups. Scale bar: 2000 μm.

Q  Quantification of DHE fluorescence intensities in (P) (n = 3).

Data information: Data are presented as the mean ± SEM and were compared using an unpaired Student's t test. [ns]not significant, *$P \leq 0.05$, **$P \leq 0.01$, ***$P \leq 0.001$. One-way ANOVA with Dunnett's correction was used for multiple comparisons.

Source data are available online for this figure.

substrates of N-Hcy (Dataset EV1). Nevertheless, our results suggest that MARSs are novel intermediary targets for Hcy-associated diseases. By employing the Hcy analogue AHT to inhibit MARSs to decrease the homocysteine signal (i.e. HTL production), we successfully decreased NTD and CHD onset rates in animal models (Fig 7). Remarkably, the inhibition of the MARS error-editing activity was well-tolerated in animal models, raising the possibility that folate supplementation and MARS inhibition could be employed in combination to reduce birth defect rates and to treat other hyperhomocysteinemia-associated diseases, especially in patients with high MARS levels.

**Table 3. NTDs incidence and AHT/NAC protecting effects in mice embryo ATRA model.**

|  | Ctrl | ATRA | ATRA + AHT | ATRA + NAC |
|---|---|---|---|---|
| No. of Litters | 8 | 7 | 7 | 6 |
| No. of implantation | 64 | 52 | 53 | 49 |
| No. of live foetuses | 64 | 48 | 52 | 47 |
| Resorption | 0 (0.0%) | 4 (7.7%) | 1 (1.9%) | 2 (4.1%) |
| Total NTDs[a,b,c] | 0 | 45 (93.7%) | 21 (40.4%) | 18 (38.3%) |
| Spinal[a,b] | 0 | 8 | 1 | 3 |
| Cranial[a,b,c] | 0 | 22 | 10 | 9 |
| Multiple[a,c] | 0 | 15 | 11 | 6 |

C57BL/6 mice were treated with ATRA or ATRA together with either AHT or NAC. NTDs in dams were determined.
Significance analysis ($P < 0.05$): [a]Ctrl vs. ATRA, [b]ATRA vs. ATRA + AHT, [c]ATRA vs. ATRA + NAC. Fisher's exact test.

**Table 4. CHDs incidence and AHT/NAC protecting effects in mice embryo ATRA model.**

|  | Ctrl | ATRA | ATRA + AHT | ATRA + NAC |
|---|---|---|---|---|
| No. of Litters | 6 | 7 | 7 | 6 |
| No. of implantation | 47 | 55 | 56 | 46 |
| No. of Live foetuses | 47 | 51 | 54 | 45 |
| Resorption | 0 (0.0%) | 4 (7.3%) | 2 (3.6%) | 1 (2.2%) |
| Total CHDs[a,b,c] | 0 (0.0%) | 40 (78.4%) | 19 (35.2%) | 11 (24.4%) |
| TGA[a,b,c] | 0 | 17 | 5 | 2 |
| DORV[a] | 0 | 8 | 3 | 2 |
| SVD[a,b,c] | 0 | 19 | 8 | 7 |
| VSD[a,b] | 0 | 20 | 6 | 7 |

C57BL/6 mice were treated with ATRA or ATRA together with either AHT or NAC. CHDs in dams were determined.
Significance analysis ($P < 0.05$): [a]Ctrl vs. ATRA, [b]ATRA vs. ATRA + AHT, [c]ATRA vs. ATRA + NAC. Fisher's exact test.

**Table 5. NTDs incidence and AHT/NAC protecting effects chicken embryo Hcy model.**

|  | Ctrl | Hcy | Hcy + AHT | Hcy + NAC |
|---|---|---|---|---|
| Survivors/ treated | 56/60 | 42/60 | 47/60 | 49/60 |
| Total NTDs[a,b,c] | 0 (0.0%) | 31 (73.8%) | 13 (27.7%) | 12 (25.5%) |
| Spinal[a,c] | 0 | 8 | 4 | 2 |
| Cranial[a,c] | 0 | 12 | 7 | 4 |
| Multiple[a,b] | 0 | 11 | 2 | 6 |

Chicken embryos were treated with Hcy or Hcy together with either AHT or NAC. NTDs were determined.
Significance analysis ($P < 0.05$): [a]Ctrl vs. Hcy, [b]Hcy vs. Hcy + AHT, [c]Hcy vs. Hcy + NAC. Fisher's exact test.

**Table 6. CHDs incidence and AHT/NAC protecting effects in chicken embryo Hcy model.**

|  | Ctrl | Hcy | Hcy + AHT | Hcy + NAC |
|---|---|---|---|---|
| Survivors/ treated | 52/60 | 39/60 | 37/60 | 40/60 |
| Total CHDs[a,b,c] | 0 (0.0%) | 21 (53.8%) | 9 (24.3%) | 10 (25.0%) |
| TGA[a] | 0 | 4 | 3 | 4 |
| DORV | 0 | 3 | 2 | 1 |
| SVD[a] | 0 | 8 | 2 | 3 |
| ASD[a,c] | 0 | 5 | 1 | 0 |
| VSD[a] | 0 | 6 | 3 | 6 |

Chicken embryos were treated with Hcy or Hcy together with either AHT or NAC. CHDs were determined.
Significance analysis ($P < 0.05$): [a]Ctrl vs. Hcy, [b]Hcy vs. Hcy + AHT, [c]Hcy vs. Hcy + NAC. Fisher's exact test.

# Materials and Methods

### Study participants

We enrolled 100 and 196 consecutive Han Chinese newborn children who received a diagnosis of CHDs and NTDs, respectively, between June 2010 and July 2015 at the Capital Institute of Pediatrics, Beijing. A total of 240 unrelated routine healthy check-up Han Chinese individuals who were confirmed to have neither CHDs nor NTDs served as population controls (Appendix Table S1). The human studies were approved by the institutional review boards of Fudan University and the Committee of Medical Ethics of the Capital Institute of Pediatrics. We obtained written informed consent from the guardians of all participants. The studies were conducted in accordance with the Declaration of Helsinki and the Department of Health and Human Services Belmont Report.

### CNV detection by AccuCopy assay

Quantitative copy number variation was genotyped with AccuCopy assay (Du *et al*, 2012), an accepted CNV genotyping method (You *et al*, 2013; Wu *et al*, 2014) based on multiplex competitive amplification (Genesky Biotechnologies; Shanghai, China). Five total multiplex PCR panels were designed to amplify five target segments,

including three segments in MARS and two in MARS2. In addition to primers for these target segments, three reference segments that were utilized for normalization were chosen at the region lacking copy number variation in the DGV database. PCR primers for each reaction are listed in Appendix Table S2. Raw data were analysed by GeneMapper 4.0 (ABI), and the height data for all specific peaks were exported into a Microsoft Excel file. The peak ratio of sample DNA to competitive DNA (S/C) for each segment was calculated. After normalization by the reference segment peak ratio, the copy number ratio of each target segment to the reference can be easily determined by dividing its ratio by that of the reference peak when the competitive DNA segments are well balanced in molecular number in the DNA mixture. The results of AccuCopy assay were confirmed by quantitative real-time PCR using SYBR-Green dye as the probe.

### Cell culture and treatments

NE4C, a neural stem cell, was used to study NTDs because it is related to neural tube development. H9C2, a rat embryonic cardiomyoblast cell line, was used to analyse CHDs because it is related to heart development. To satisfy the large demand for proteins of *in vitro* assays (in tubes), HEK293T was used to affinity purify SODs and MARSs, because HEK293T has high transfection and expression efficiency. Cell lines were maintained at 37°C and 5% $CO_2$. NE4C, H9C2 and HEK293T cells were maintained in Dulbecco's modified Eagle's medium (DMEM) containing glucose (1 g/l). All media contained 2 mM glutamine and were supplemented with 10% foetal bovine serum. The cell line resources are shown in Appendix Table S3. Cells were regularly tested for mycoplasma (Selleck). L-homocysteine, L-homocysteine thiolactone hydrochloride, folic acid and all-trans retinoic acid (ATRA) were purchased from Sigma Chemical (St. Louis, MO, USA). Acetyl homocysteine thioether (AHT) was synthesized in our laboratory for this study, and the synthesis protocols are provided in methods. Homocysteine, HTL and AHT were freshly prepared in PBS, and ATRA was dissolved in DMSO prior to use. To treat cells, homocysteine (20 μM) and HTL (10 μM) were added to the culture media to reach the final indicated concentration 4–6 h before harvesting. AHT (70 μM) was added to the culture media 3–4 h before harvesting, unless indicated otherwise.

### Antibodies

Commercial antibodies were purchased from the indicated companies as shown in Appendix Table S4. Then, α-N-Hcy was prepared following our reported protocol. Briefly, 100 μM bovine serum albumin (BSA) was homocysteinylated by incubation with 1 mM HTL in 0.1 M $Na_2CO_3$ (pH 8.0) with 1:10 (v/v) pyridine at 25°C for 14 h. The modified proteins were purified by passing reaction mixtures through a Sephadex G-25 gel filtration column with 50 mM Tris buffer as the mobile phase in an AKTA-FPLC system (GE Healthcare, Chicago, IL, USA) to remove organic reagents. After verifying gel mobility to ensure the modification, the modified proteins were used to immunize rabbits. The antiserum was collected after four rounds of immunization and checked for specificity before affinity purification by employing cross-linked synthesized Hcy-lysine-containing peptides.

**Western blotting**

Cells or tissues were washed with PBS 3 times and then homogenized in SDS loading buffer, and total protein extracts were immunoblotted with specific antibodies as shown in Appendix Table S4. Western blot signals were obtained by detecting chemiluminescence using a Typhoon FLA 9500 Biomolecular Imager (GE Healthcare, Chicago, IL, USA). Blot intensity was quantified by densitometry using ImageJ software.

**Immunohistochemistry**

For immunohistochemical staining, 12-μm serial coronal brain or heart sections were used as described previously with the following modifications (Wang *et al*, 2014). Briefly, coronal sections were blocked in TBS combined with 1% Triton X-100 and 10% donkey serum for 2 h. All primary antibodies were applied and incubated for 2 days at 4°C. The primary antibodies used for IHC are shown in Appendix Table S4. Secondary antibodies against the appropriate species were applied and incubated for 4 h at room temperature. Fluorescently stained sections were counterstained with DAPI (200 ng/ml; Sigma, St. Louis, MO, USA) for 2–5 min and then mounted on a coverslip with Gel Mount medium (BioMeda, Foster City, CA, USA).

**Cell growth analysis**

Neuronal NE4C stem cell growth was measured using the xCELLigence RTCA MP instrument with an E-plate (ACEA Biosciences, San Diego, CA, USA), which is normally used to monitor cell attachment and growth in real time. NE4C stem cells were seeded on the E-Plate at a density of $1 \times 10^4$ cells/well and incubated overnight. The impedance of electron flow caused by adherent cells is reported using a unitless parameter, cell index (CI), where CI = (impedance at time point $n$ − impedance in the absence of cells)/nominal impedance value. The cell index responds to changes in cell number and cell adhesion; more cells result in a larger cell index.

**Ectopic expression of MARSs**

MARS cDNA (NM_004990.3) and MARS2 cDNA (NM_175439.3) were cloned in the pRK7-Flag vector to generate expression plasmids. Lipofectamine® 3000 transfection reagent (L30075, Thermo Fisher, Waltham, MA, USA) was used to transfect MARS and MARS2 plasmids into the NE4C cells. Then, 5 μg plasmids were transfected in $1 \times 10^6$ cells, which were harvested after 48 h.

**Apoptosis analysis**

In cultured cells, apoptosis was detected by FITC Annexin V Apoptosis Detection Kit 1 (BD Biosciences, San Jose, CA, USA). Data were collected on an Accuri C6 flow cytometer (BD Biosciences). Immunoblot assay using anti-cleaved-caspase3 antibodies was also used to detect apoptosis (Erhardt & Cooper, 1996; Fig 2B). In frozen tissue sections, the TdT-mediated dUTP Nick-End Labeling (TUNEL) assay (Promega, Madison, WI, USA) was used to detect apoptosis. Briefly, air-dried sections were fixed at 25°C with 4% PFA for 15 min and washed twice with PBS. Next, the sections were immersed in equilibration buffer for 10 min, which was then replaced by a mixture including 1 μl TdT enzyme, 5 μl nucleotide mix and 45 μl equilibration buffer. The sections were then held at 37°C for 90 min. To terminate the TdT enzymatic reaction, 2 × SCC was added at room temperature for 15 min.

**Measurement of superoxide levels**

Superoxide levels were measured by monitoring the fluorescence of dihydroethidium (S0063, Beyotime, Haimen, China)-stained cells (lex = 300 nm, lem = 610 nm; Erhardt & Cooper, 1996) following the manufacturer's instruction. Briefly, cells were harvested from quadruplicate cultures, resuspended, incubated for 30 min in the dark and washed twice with PBS solution, and fluorescence was recorded. Whole-mount fluorescent DHE staining was conducted in mice and chicken embryos at E10.5 and E4, respectively. Embryos were incubated with DHE (1:1,000) in serum-free Iscove's modified Dulbecco's medium for 30 min at 37°C. After incubation with DHE, embryos were washed with PBS 3 times and examined immediately using a fluorescence microscope. Image analysis was performed blinded. All superoxide levels were expressed as relative levels to controls, which were arbitrarily set at 100%.

**Detection of homocysteine and HTL**

Intracellular homocysteine concentrations in cell cultures were measured by a Homocysteine-EIA Kit (Bio-Rad, Hercules, CA, USA) following the manufacturer's protocols. HTL was assayed as follows: cells were harvested by washing with PBS and then denatured using pre-chilled 60% methanol dissolved in ddH$_2$O. Cell lysates were collected and centrifuged at 10,000 *g* for 5 min at 4°C. The supernatant was vacuum dried, redissolved in ddH$_2$O and then subjected to ultrafiltration on a PVDF low-protein-binding membrane (Millex-GV$_4$ and Millex-HV$_4$; Millipore, Bedford, MA, USA). The metabolites were extracted, and HTL was analysed by LC-MS (Barathi *et al*, 2010).

**Identification of N-Hcy substrates**

*Liquid chromatography-mass spectrometry*
The peptides were resuspended with 25 μl solvent A (A: water with 0.1% formic acid; B: ACN with 0.1% formic acid), separated by nanoLC and analysed by online electrospray tandem mass spectrometry (Nie *et al*, 2015). The experiments were performed on a nano Acquity UPLC system (Waters Corporation, Milford, USA) connected to a LTQ Orbitrap XL mass spectrometer (Thermo Fisher Scientific, San Jose, CA, USA) equipped with an online nano-electrospray ion source. Each sample was loaded onto the Thermo Scientific Acclaim PepMap C18 column (100 μm × 2 cm, 3 μm particle size), with a flow of 10 μl/min for 3 min and subsequently separated on the analytical column (Acclaim PepMap C18, 75 μm × 15 cm) with a linear gradient, from 5% B to 45% B in 75 min. The column was re-equilibrated at initial conditions for 15 min. The column flow rate was maintained at 300 nl/min, and column temperature was maintained at 40°C. The electrospray voltage of 1.8 kV versus the inlet of the mass spectrometer was used. LTQ Orbitrap XL mass spectrometer was operated in the data-dependent mode to switch automatically between MS and MS/MS

acquisition. Survey full-scan MS spectra with one microscan (m/z 400–1,800) were acquired in the Orbitrap with a mass resolution of 60,000 at m/z 400, followed by MS/MS of the eight most-intense peptide ions in the LTQ analyser. The automatic gain control (AGC) was set to 1,000,000 ions, with maximum accumulation times of 500 ms. The minimum MS signal for triggering MS/MS was set to 500, and single charge state was rejected. Dynamic exclusion was used with two microscans and 90-s exclusion duration. For MS/MS, precursor ions were activated using 35% normalized collision energy at the default activation $q$ of 0.25 and an activation time of 30 ms. For MS/MS, we used an isolation window of 3 m/z and automatic gain control (AGC) was set to 20,000 ions, with maximum accumulation times of 120 ms.

### Data processing, validation and analysis

Raw MS files were analysed by MaxQuant version 1.4.1.2. MS/MS spectra were searched by the Andromeda search engine against the SwissProt-human database (Release 2014-04-10) containing forward and reverse sequences (total of 40,492 entries including forward and reverse sequences). Additionally, the database included 248 common contaminants. In the main Andromeda search, precursor mass and fragment mass had an initial mass tolerance of 5 ppm and 0.05 Da, respectively. The search included variable modifications of methionine oxidation. Minimal peptide length was set to seven amino acids, and a maximum of four miscleavages was allowed. The false discovery rate (FDR) was set to 0.01 for peptide and protein identifications.

### In vitro homocysteinylation of SOD1 and SOD2

Purified FLAG-tagged SODs were incubated with HTL-HCl in 0.1 M sodium phosphate buffer (pH 8.0) at 22°C for 6–12 h. The modified SODs were desalted and recovered by passing the reaction mixture through Amicon Ultra-4 Centrifugal Filter Devices (Millipore, Bedford, MA, USA) in a desktop centrifuge.

### SOD assay

SOD1 and SOD2 activity assays were conducted using the SOD Assay Kit (Dojindo Molecular Technology Inc., Rockville, MD, USA), which use highly water-soluble tetrazolium salt (WST-1) as a substrate, according to the manufacturer's instructions. Briefly, WST-1 produces a water-soluble formazan dye upon reduction with a superoxide anion. The rate of the reduction of $O_2$ free radicals is linearly related to xanthine oxidase (XO) activity and is inhibited by SOD. The SOD activities were determined by a colorimetric method.

### TopFlash reporter assay

A TCF/LEF reporter assay was used to examine the effect of Hcy/ HTL on WNT-induced transcriptional activity. First, 2.5 μg TOPFLASH (gifted from Dr. Tao Zhong's lab, Fudan University (Ni et al, 2011)) and 2.5 μg FOPFLASH plasmids (Millipore, Bedford, MA, USA; Appendix Table S6) were transfected in $1 \times 10^6$ NE4C cells using Lipofectamine® 3000 transfection reagent (L30075, Thermo Fisher, Waltham, MA, USA) for 48 h. Hcy/HTL was added in the culture medium 4 h before cell collection.

Reporter activities were measured with a plate reader (LD400; Beckman Coulter, Fullerton, CA, USA) using a dual-luciferase reporter assay system (Promega, Madison, WI, USA). TCF/LEF activity was defined as the ratio of TOPFLASH:FOPFLASH reporter activities.

### Synthesis of AHT

DL-homocysteine (5 g) was added to a 250-ml round-bottom flask, followed by addition of 160 ml methanol. A total of 4 ml $SOCl_2$ was added to the reaction mixture on ice. The reaction was allowed to incubate under room temperature for 24 h before the solute was removed. The product was redissolved in dichloromethane and 10.35 ml triethylamine ($Et_3N$) on ice, followed by addition of Di-tert-butyl dicarbonate ($Boc_2O$) before stirring under room temperature for 2.5 h. The reaction was stopped, and the solute was evaporated. $CH_3OH$ (80 ml), $H_2O$ (35 ml) and tributylphosphine ($PBu_3$, 9.2 ml) were sequentially added to the products and stirred at 50°C for 30 min. The product was extracted with diethyl ether (200 ml) and water (100 ml), and the diethyl ether phase was obtained and washed with saturated NaCl and dehydrated with $Na_2SO_4$ to get intermediate 2 (Equation 1).

Compound 2 (3 g) was added to methanol (120 ml), $Et_3N$ (9.24 ml) and keep stirring at room temperature for 30 min, and iodoacetamide (2.64 g) was then added and keep stirring for another 12 h. Intermediate compound 3 was obtained by saturated NaCl washing and dehydrated with $Na_2SO_4$ (Equation 2).

To intermediate 3 (4 g), add DCM (60 ml), TFA (10 ml) and keep stirring under room temperature for 12 h, and intermediate 4 was obtained by removing solute (Equation 3).

To intermediate 4 (3 g), add NaOH (11 g) and keep stirring at room temperature for 12 h, followed by removing THF. The final product was obtained by ethanol devolvement followed by filtration and evaporation (Equation 4).

**The paper explained**

**Problem**

Hyperhomocysteinemia has been associated with a number of disease states. However, risk reduction by homocysteine-lowering therapy has not been confirmed by randomized trials, suggesting that missing factors may be involved in hyperhomocysteinemia pathologies.

**Results**

Methionyl-tRNA synthetases (MARSs) produce a reactive homocysteine intermediate, homocysteine thiolactone, and form protein N-homocysteinylation. Increased copy numbers of MARS or MARS2-encoding genes were more frequently detected in patients of neural tube defects (NTDs) and congenital heart defects (CHDs). Homocysteine sensing inhibition by targeting MARS using the homocysteine analogue acetyl homocysteine thioether (AHT) lowered NTDs and CHDs in both All-trans retinoic acid (ATRA)- and hyperhomocysteinemia-induced animal models without affecting homocysteine levels. Our mechanistic studies revealed that MARS and MARS2 over-expression augments homocysteine signals by promoting N-homocysteinylation. Additionally, 870 N-Hcy proteins were identified. N-homocysteinylation inactivates superoxide dismutases (SODs) and leads to the accumulation of reactive oxygen species, which promote apoptosis and cause deregulation of Wnt/β-catenin signalling. Developing and proliferating cells, such as those in embryos or cancer cells, have higher MARS levels, which may result in increased sensitivity to homocysteine.

**Impact**

Our study identified MARSs as new risk factors for NTDs and CHDs and showed that homocysteine signalling is determined by both homocysteine levels and MARS reactivity, indicating that MARSs are previously overlooked genetic determinants and key pathologic factors of hyperhomocysteinemia. This suggests that measuring homocysteine signalling by detecting N-homocysteinylation levels or assessing levels of both homocysteine and MARSs will be of better diagnostic value for predicting hyperhomocysteinemia-associated diseases such as NTDs and CHDs. Moreover, the development of AHT for targeting MARS activity may hold broad implications on preventing hyperhomocysteinemia-associated diseases, because it prevents detrimental effects due to both elevated homocysteine levels and MARS expression.

The success of synthesis was confirmed by NMR analysis for final product 5.

**Detection of MARS activities for producing Homocysteine Thiolactone by Thin Layer Chromatography**

The TLC system (silica gels developed in 4:1:1 butanol:acetic acid: water; Merck, Kenilworth, NJ, USA) used to detect homocysteine thiolactone was performed according to the published procedures (Jakubowski & Fersht, 1981). Reaction mixtures (50 µl) contained 50 mM Hepes (pH 8.0), 10 mM dithiothreitol, 10 mM MgCl₂,

1.5 mM ATP, 20 µM homocysteine, and 1–10 µM MARS or MARS2. After 100 min at 25°C, 2–4 µl aliquots were spotted onto the origin line of the cellulose plates, and spotting was repeated after spots dried until the total volume reached 40 µl. The chromatograms were developed for 6–8 h at room temperature, and then, spots were sprayed with ninhydrin. Hcy (Rf = 0.34) and HTL (Rf = 0.57) showed different shifts that were clearly distinguishable.

**Animals**

All animal care and experiments were conducted in accordance with the Institutional Animal Care and Use Committees of Fudan University.

*Rats and mice models*

Sprague Dawley (SD) rats and C57BL/6 mice were obtained from Sino-British SIPPR/B&K Lab Animal Ltd. (Shanghai, China), housed in polycarbonate cages and given free access to food and water with a 12-h light:dark cycle. Male and female animals were paired (1:2) overnight. Mating was verified by observation of a vaginal plug the next morning, which was considered as E0.5 of pregnancy. Pregnant females were randomly assigned to control and experimental groups ($n \geq 7$ per group). NTDs were induced by administering an oral dose of vehicle (olive oil) containing ATRA (50 and 40 mg/kg body weight for rats and mice, respectively) to pregnant rats on E10.5 and mice on E6.75, and the control group was treated similarly with vehicle alone. The ATRA + AHT group received AHT at 16 mg/kg body weight/day for rats from E6.5 to E12.5 by tail vein injection and 20 mg/kg body weight/day for mice from E0.5 to E10.5. The ATRA + NAC group received NAC at 300 mg/kg body weight/day for rats from E0.5 to E12.5 and 200 mg/kg body weight/day for mice from E0.5 to E10.5 by oral administration. Embryos were dissected from decidual capsules in the uteri and were grossly evaluated to confirm any NTD phenotypes at E18.5 for rats and E10.5 for mice. Morphological analysis of hearts was conducted at E14.5 for mice.

*Chicken embryo model*

All chicken experiments were conducted in accordance with the guidelines of the ethics Committee for Animal Research of Fudan University. In the chicken hyperhomocysteinemia model, fertilized White Leghorn chicken eggs were incubated in a vertical position at 38.0°C and 65.0% humidity. Hcy-treated chick embryos received a single 50 µl injection of 15 µM D,L-Hcy dissolved in saline solution at 28 h after incubation; control embryos received only 50 µl saline solution. In the AHT- or NAC-treated group, we also dissolved D,L-Hcy with an additional 50 µM AHT or 2 mM NAC (A–9165; Sigma-Aldrich Chemie, Zwijndrecht, The Netherlands) in 50 µl saline solution. After injection, eggs were resealed with Scotch tape and returned to the incubator at 38.0°C for further development. Morphological analyses were conducted at E8 for NTDs and E9 for CHDs.

**Haematoxylin–eosin staining (HE staining)**

Next, 6-µm serial coronal brain or heart sections were prepared and stained with haematoxylin and eosin. Histological evaluations were performed using HE staining to assess cardiac malformation.

## Quantification and statistical analysis

Chi-squared and Fisher's exact tests were used to assess variation frequencies in the MARS copy number between the controls and individuals with either CHDs or NTDs. The statistical significance of comparisons between the two groups of data was assessed using unpaired two-tailed Student's *t* test. The statistical significance of comparisons between multiple groups was determined by one-way ANOVA with Dunnett's correction using GraphPad 6 (Prism). The exact *P* values were provided in Appendix Table S7.

## Data availability

Sequencing data of CNV analysis for this study are deposited in the NODE database (https://www.biosino.org/node/project/detail/OEP 000640).

**Expanded View** for this article is available online.

## Acknowledgements
This work was supported by Grants from the State Key Development Programs of China (Nos. 2018YFA0801300, 2018YFC1004700, 2018YFA0800300), the National Science Foundation of China (Nos. 31330023, 81722021, 31671483, 81771627, 31521003, 31821002, 91753207, 31930062, 31871432, 81974175, 81500977, U1432242, 31425008, 31801167), Science and Technology Municipal Commission of Shanghai, China (Nos. 16JC1405300, 17YF1424200), Shanghai Rising-Star Program (No. 18QA1400300), Shanghai Medical Center of Key Programs for Female Reproduction Diseases (2017ZZ01016) and grant from Key Laboratory of Reproduction Regulation of NPFPC. We appreciate Dr. Yufang Zheng, Dr. Ruoyi Gu and Dr. Yixiang Lin for guidance on neural tube and heart development experiments.

## Author contributions
SZ, XM, WX and HW conceived the project, over-sighted the research. XM and DQ performed mouse, chicken, monkey and rat developing experiments; TZ and JW supplied the NTD/CHD patient blood samples; JZ performed copy number sequencing; XM and DQ detected gene/protein expression in CNV samples. HW, YZ, XM and JY provided rats and performed rat anatomy. XM, LH, JN, YL and JM performed the cellular and biochemical experiments. WX and XM produced the N-Hcy antibodies; XM, YX and ZL performed the LC-MS and bioinformatics analysis; XM, JH, PZ and WS synthesized AHT; DQ, XM and HS performed immunohistochemistry experiments. SZ and XM wrote the manuscript, HW, RHF and LJ helped to edit the manuscript. XM, DQ and SZ submitted and revised the manuscript.

## Conflict of interest
The authors declare that they have no conflict of interest.

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
