## [Review Process File · EMBO Molecular Medicine]

Inhibiting MARSs reduces hyperhomocysteinemia-associated neural tube and congenital heart defects

Xinyu Mei, Dashi Qi, Ting Zhang, Yin Zhao, Li Jin, Junli Hou, Jianhua Wang, Yan Lin, Yu Xue, Pingping Zhu, Zexian Liu, Lei Huang, Ji Nie, Wen Si, Jingyi Ma, Jianhong Ye, Richard H. Finnell, Hexige Saiyin, Hongyan Wang, Jianyuan Zhao, Shimin Zhao and Wei Xu

Review timeline:

Submission to The EMBO Journal:	17 June 2018
Decision from The EMBO Journal:	22 June 2018
Transfer to EMBO Molecular Medicine:	24 June 2018
Editorial Decision:	14 August 2018
Revision received:	6 March 2019
Editorial Decision:	4 April 2019
Revision received:	1 August 2019
Editorial Decision:	15 August 2019
Revision received:	16 December 2019
Accepted:	19 December 2019

Editor: Lise Roth

Transaction Report:

Decision from The EMBO Journal

22 June 2018

Thank you for submitting your manuscript to The EMBO journal. I have now read your study carefully and discussed the work with other members of the editorial team. I am afraid that we have decided not to pursue publication of this manuscript in The EMBO Journal. However, I have taken the liberty to discuss your study with my colleague Lise Roth at our sister journal EMBO Molecular Medicine and she would be happy to offer peer review if you were to transfer your manuscript there.

From our side, we appreciate that you find MARS/MARS2 expression to correlate with HHCy disease severity in human patient samples and to be required for HCy-induced toxicity and apoptosis in cell culture. You go on to show that MARS-dependent conversion of HCy to HTL increases ROS production and blocks beta-catenin signaling via protein N-HCy modification. Finally, you show that MARS-inhibition using a HCy-derived small molecule drug rescues neural tube closure in a rat model for HHCy. However, while your study is thus the first to implicate MARS/MARS2 as a risk factor for HHCy, we are concerned that the ability of these two aaRS proteins to trigger the HCy-to-HTL conversion - as well as the deleterious effect of HTL via protein modifications - had been established in earlier studies. We therefore see the main advance of the current study to be in identifying MARS/MARS2 as a therapeutic target for HHCy and as such, it would be a much stronger fit for our sister journal EMBO Molecular Medicine, a journal that focuses on research at the interface between molecular biology and pre-clinical/translational work. While I am thus sorry to say that we have decided not to send the manuscript out for peer-review for The EMBO Journal, I would strongly recommend you to transfer the manuscript to EMBO Molecular Medicine for peer review there following the link provided below. I would like to emphasise that transferring your work to EMBO Molecular Medicine does not involve any reformatting.

Thank you for giving us the opportunity to consider this manuscript. I regret that we have to disappoint you on this occasion, but hope that you will use this opportunity to transfer your work to EMBO Molecular Medicine.

Thank you for the submission of your manuscript to EMBO Molecular Medicine. We have now heard back from the three referees whom we asked to evaluate your manuscript.

As you will see from the reports below, while they all mention the novelty and potential high medical interest of the study, they also raise substantial concerns about your work, regarding both the substance (inadequate cell lines, lack of appropriate statistics, disregard of alternative explanations) and the form (inadequate references, missing method description, mislabeling of figures, overstatements regarding the conclusions), which should be convincingly addressed in a major revision of the present manuscript. In particular, in a cross-commenting exercise, the referees underlined the need for an animal model of hyperhomocysteinemia, as well as the importance of performing all experiments in a single cell type.

Addressing the reviewers concerns in full will be necessary for further considering the manuscript in our journal. Still, revising the manuscript according to the referees' recommendations appears to require a lot of additional work and experimentation. We are therefore ready to extend the deadline to 6 months with the understanding that acceptance of the manuscript would entail a second round of review. EMBO Molecular Medicine encourages a single round of revision only and therefore, acceptance or rejection of the manuscript will depend on the completeness of your responses included in the next, final version of the manuscript. For this reason, and to save you from any frustrations in the end, I would strongly advise against returning an incomplete revision and would also understand your decision if you choose to rather seek rapid publication elsewhere at this stage. Should you find that the requested revisions are not feasible within the constraints outlined here and prefer, therefore, to submit your paper elsewhere, we would welcome a message to this effect.

Please also contact us as soon as possible if similar work is published elsewhere. If other work is published, we may not be able to extend the revision period beyond three months.

I look forward to receiving your revised manuscript.

***** Reviewer's comments *****

Referee #1 (Remarks for Author):

This manuscript submitted by Mei et al describes a new signaling pathway in mediating the adverse effects of homocysteine (Hcy). Because reducing Hcy levels do not always effectively alleviate hyperhomocysteinemia-related pathology, the authors started to seek out additional molecular intermediators in the Hcy metabolic pathway. They focused on methionyl-tRNA synthetases (MARSS). Maternal blood of human NTDs and CHDs, several cell models and the all-trans retinoic acid (ATRA)-induced rat spinal bifida model were used in the comprehensive studies described in this paper. Increased copy numbers of both cytosolic-MARS and mitochondrial-MARS2 were associated with human NTDs and CHDs. Experimental data supported the hypothesis that MARSS converted Hcy into a reactive intermediate Hcy thiolactone (HTL), which inactivated two antioxidant enzymes, SOD1 and SOD2 through lysine homocysteinylation (N-Hcy). Most interestingly, they developed a small molecule acetyl Hcy thioether (AHT) in inhibiting NARSs activity. AHT reduced ATRA-induced NTDs in the rat. Convincible data support the conclusion that MARSS are critical parts of Hcy sensing and mediate the teratogenicity of ATRA. The findings are novel and the development of AHT in targeting

MARS activity may have broad implications on preventing hyperhomocysteineemia-related diseases. Revealing N-Hcy-modified SOD1 and SOD2 provided new mechanistic insights for Hcy-induced oxidative stress, apoptosis and growth inhibition. Overall, the findings would advance the field in the origin and cause of birth defects.

However, there are concerns related to the low quality of some of the data, obscure methodology and the unconvincing link of the discovered pathway with increased canonical Wnt signaling.

- 1) The Abstract misses some of the key findings. Suggest re-writing to reflect the key points;
- 2) The information for the human study is unclear. Need a Table to specify the types of birth defects. When was maternal blood collected? Were Hcy and HTL levels determined? The correlation analysis is not very convincing for the relation between MARS and birth defects.
- 3) In Fig. 1A, where do the error bars come from if individual samples were quantified? Assay replicates do not represent biological replicates. Graph should depict averages and sample size. The rest panels of Figure 1 need quantification for intensities and Panel B needs a large magnification.
- 4) Several cell lines were used but not well justified. The closest one should be the NE4C cell line yet only few data generated from this cell line.
- 5) The method used for total ROS levels is not optimal. Preferably, the author should measure superoxide because SOD1 and SOD2 activity is impaired by the Hcy-MARS-N-Hcy pathway.
- 6) There is entirely unclear how SOD1 and SOD2 activity were determined.
- 7) Literature supports the notion that oxidative stress suppresses the canonical Wnt signaling pathway. There are both increased beta-catenin levels and DKK1, the Wnt inhibitor. The argument that Hcy and HTL decrease DVL1-NRX interaction is not well founded. Generally, the data do not strongly support the hypothesis that Hcy and HTL enhance Wnt signaling. The blots in Fig. 5E, F, G, H and I were saturated and the difference was hardly visible. Need data on the NE4C cell line.
- 8) In Fig. 6, Panel B, the right blot may be a failed run because the Hcy signals were not clearly visible.
- 9) In the rat study, given the inhibitor AHT at embryonic day 10.5, which is half way through neurulation and maybe completed posterior neural tube closure, was not justified. In the text, it said that AHT treatment is along with ATRA, which was given from E6.5. Most of the measurement was done in the brain while ATRA exclusively induces spinal bifida. Suggest repeating the experiment by measuring the endpoints in neurulation stage embryos.

Referee #2 (Comments on Novelty/Model System for Author):

There are many experimental methods, or how experimental results were normalized, that are not adequately described. These are pointed out in the comments to the authors.

Referee #2 (Remarks for Author):

This is an interesting study to test the hypothesis that methionyl-tRNA synthetases (MARSs) gain of function (in patients by increased gene copy numbers) increase neural tube defects (NTDs) and congenital heart defects (CHDs), not through increased substrate for protein synthesis, but through sensing homocysteine (Hcy) and modifying superoxide dismutases (SODs) by lysine homocysteinylation (N-Hcy). SOD1/2 N-Hcy thereby decreases their activities and increases reactive oxygen species (ROS), which are associated with NTDs and perhaps CHDs. The progression of several *in vitro* (cell free and in cultured cells) and *in vivo* (animal model) experiments is logical, although the experiments were designed to prove the hypothesis correct, and do not consider alternative explanations or activities of MARSs and involvement in NTDs and CHDs. The authors also should address that, while ROS induced by altered metabolism, such as maternal diabetes, perhaps folate deficiency, retinoic acid, and some drugs, may be sufficient to cause NTDs and CHDs, there may be other genetic and environmental causes of these defects that do not increase ROS. Many of the experimental methods and reagents need more detailed descriptions, so it is not possible at this point to evaluate the validity of several of the experimental results. Although the decrease in NTDs caused by all-trans retinoic acid (ATRA) by a small molecule MARSs inhibitor in the rat is consistent with the hypothesis (although effects on CHDs or litter sizes were not reported), it cannot be determined whether this was due to, or just correlated with, restoration of SOD1/2 activities and resulting ROS levels, because treatment of a group of ATRA-treated rats with an antioxidant was not included. The manuscript, as well as the supplement, needs better proof reading for spelling, punctuation, grammar, missing words, and spaces between

text and reference citations. Care needs to be taken to accurately cite the literature and to include additional references in some cases.

Specific comments:

1. Although inhibition of MARSs with a small molecule reduced (but did not prevent) NTDs in some of the embryos of ATRA-treated rats, there may be some causes (genetic or environmental) of NTDs that are not mediated by MARS activities and in which MARSs inhibition would be ineffective. Therefore, the title of the manuscript seems over stated and should be revised.
2. If the hypothesis is correct, there should have been an increase in CHDs in the embryos of ATRA-treated rats, unless ATRA does not stimulate MARSs activities in embryonic heart progenitors. More information needs to be provided regarding effects of ATRA and AHT (the MARS inhibitor) on fetal CHDs and on litter sizes (because fetuses with CHDs might die before E18.5), and what effects, if any, ATRA has on MARSs activities in embryonic heart.
3. The results in the rat model are consistent with the hypothesis, but it cannot be concluded that MARSs effects on SOD1 or 2 activities are due to ROS, leading to NTDs unless it is shown that antioxidants (vitamin E or a glutathione precursor) have the same effect to reduce NTDs as AHT in a group of ATRA-treated rats.
4. Abstract, line 6: "Increased copy numbers of cytosolic-MARS and/or mitochondrial-MARS2" should be followed by "-encoding genes". Please take care to italicize when referring to genes.
5. Same sentence: "were detected in blood samples of patients with neural tube defects (NTDs) and congenital heart defects (CHDs)". It should be stated in how many patients out of how many screened.
6. Introduction, first sentence, Braly & Holford, 2003 is not a retrievable reference. The claim in the first sentence is not what the title of the reference says. It is advised to state what Hcy disorders are associated with Hcy (too much or too little?) and to provide valid references.
7. Hyperhomocysteinemia (HHcy) is not a nutritional disorder. Do the authors mean "metabolic disorder"? If they claim that it is common, they need to state the prevalence.
8. "which are caused by insufficient development of the neural tube or the heart". "Insufficient development" should be replaced with "maldevelopment". There may be insufficient development in some (but not necessarily all) of these defects. However, at most, it can be claimed that insufficient development is associated with these defects but not the cause of them.
9. P 4: "Being proteogenic enzymes that charges methionyl-tRNA, MARS and MARS2 expression can be elevated in proliferating cells, including actively differentiating embryonic cells and cancer cells, the latter having been confirmed in human colon cancer(Kushner et al, 1976)." References need to be provided for proliferating and embryonic cells.
10. Next sentence: "Notably, MARS and MARS2 gene duplication had been associated with increased ROS(Bayat et al, 2012), a consistent finding in NTDs(Weksler-Zangen et al, 2003)". I think the authors mean that increased ROS are a consistent finding in NTDs. However, this is not true (there can be causes of NTDs independent of increased ROS), and this not an appropriate reference for this claim. Diabetes can increase ROS, and maternal diabetes-induced ROS can induce NTDs, but this paper did not show this.
11. Same sentence: "cancers(Trachootham et al, 2009)". This paper doesn't say this. Also, although ROS-induced DNA damage can cause cellular transformation, ROS induced by chemotherapy or radiation can kill cancer cells.
12. Same sentence: "that is associated with increased apoptosis" needs a reference. If Dhalla, et al., preceding the clause is the reference, the sentence should be rewritten. However, it should be noted that if ROS-induced apoptosis is involved in cancer, were true in cancer, cells would die, not thrive.
13. Results, p 5, first paragraph: "we collected blood samples from clinically confirmed NTDs and CHDs cases known to be induced by HHcy" How could this be known? Or, do the authors mean "associated with"?
14. Is it known whether or not the amplified copies of MARS or MARS2 are wild type (in coding and regulatory elements)? This should be addressed.
15. P 6: "The increased copy number of MARSs in NTDs and CHDs indicated that overexpression of MARSs may deregulate development of these organs." NTDs and CHDs are not organs. Instead ""may deregulate development of the neural tube and the heart". Also, it does not necessarily follow that increased copy numbers of MARSs caused overexpression of the encoded proteins, especially if the amplified copies were not wild type. It is possible that increased copies disrupted genes that are necessary for normal development of the neural tube and the heart.
16. P 7: How was growth of neuronal NE4C stem cells measured? What are the units?
17. How was BCL2 over-expressed? How was prevention of apoptosis determined?

18. "showing that Hcy, but not Met, inhibits cell growth mainly by augmenting apoptosis." "showing" should be replaced with "suggesting".
19. How was MARS or MARS2 over-expressed? How much siRNA was transfected?
20. "Data showed that oxidative stress related pathways were enriched". Oxidative stress-related pathways were only some of the enriched pathways. This should be made clear. Perhaps, "oxidative stress-related pathways were among the pathways that were enriched."
21. Why were HEK293T all of a sudden used instead of NE4C cells for Fig. 2G? Rationale for use of one cell type or the other for different experiments should be provided throughout the manuscript.
22. Why is the format of Fig. 3D different from A-C? What do #1, 2, and 3 refer to?
23. Fig. 3 and subsequent, it's not clear at all how ROS were assayed, and what they were expressed relative to.
24. Fig. 3 and others: What meant by %HTL, Hcy, N-Hcy levels? What is the denominator?
25. Fig. S2C needs more controls to validate the anti- N-Hcy antibody, such as an anti-BSA ab, and secondary ab alone.
26. How were SOD1/2 activities assayed?
27. "Hcy-mimetic" meaning isn't clear in the figure 4 legend. Also, the legends don't make it clear which assays were performed in vitro and which in vivo.
28. The legend for Fig. 5B says Hcy, not HTL. Do the authors mean "transcription of" or "transactivation by"?
29. The Topflash assay needs a description and/or reference.
30. More details need to be provided for methods for entire Fig. 6 and Fig. S7.
31. P 11: "The abilities of AHT to reverse Hcy effects were further tested." Is not clear from the subsequent panels in Fig. 6. Do the authors mean the production of Hcy or the effects of Hcy? Or both?
32. Pp 11-12: "An all-trans retinoic acid (ATRA) NTDs-complicated pregnancy rat model, which is thought to induce NTDs in rats partly by increasing Hcy (Xu et al, 2010) and ROS concentrations(Mantymaa et al, 2000; Mathieu et al, 2005)". All of these references provide evidence that ATRA increases Hcy and/or ROS in different adult cancer cells, but none of them provide evidence that ATRA induces NTDs, or that ATRA-induced NTDs by increasing Hcy and/or ROS. Appropriate references should be provided or this statement should be revised.
33. Fig. S7C is of poor quality. One can't see spina bifida in most of the fetuses, even upon zooming in.
34. The Discussion could be significantly shortened. 1-1.5 pages should be sufficient.

Referee #3 (Comments on Novelty/Model System for Author):

Most mechanistic studies were carried out in cell lines that are not related to neural tube defects at all. Animal models of hyperhomocystenemia may be required.

Referee #3 (Remarks for Author):

In this manuscript Mei et al report the potential roles of methionyl-tRNA synthetase (MARS)-generated homocysteine signals in neural tube defects (NTDs). The major findings include that increased copy numbers of MARS/MARS2 in patients are associated with neural tube defects and congenital heart defects. In vitro experiments suggest that overexpression of MARSs potentiates Hcy-induced ROS accumulation and apoptosis, possibly acting through sequential regulation of related intermediates HTL, N-Hcy, and SOD1/2. MARSs overexpression also upregulates beta-catenin expression level. SOD1/2 overexpression partly reverses Hcy- or HTL-induced beta-catenin accumulation and abolishes the ability of HTL in decreasing DVL1-NRX interaction in a kidney cell line. Finally, they show that an Hcy analog AHT inhibits MARSs, HTL, N-Hcy, and ROS productions in vitro and decreases NTD rates induced by all-trans retinoic acid in rats. This is an important study with attractive in vivo results and abundant in vitro biochemical data, which may reveal the function and related mechanism of MARSs in homocysteine-induced neural tube defects. A few concerns might need to be addressed or clarified.

Major concerns:

1. Figures 1 and S1 show MARSs expression in developing or proliferating cells of human placenta, embryonic monkey/rat brain sections, and urothelial cell carcinoma cells, none of these are related to neural tube closure.
2. Most mechanistic studies were carried out in cell lines that are not related to neural tube defects at all. Where are MARSs expressed in vivo during neural tube closure? Do MARSs actually regulate SOD1/2, ROS, and apoptosis in neural tube closure?
3. MARS overexpression increased beta-catenin and Tcf levels in cell lines. Does overactivation of Wnt/beta-catenin signaling cause NTDs?
4. Mechanistic and prevention studies using animal models of hyperhomocystenemia may dramatically enhance the significance of their findings.
5. It is quite impressive that maternal administration of the MARS inhibitor AHT can reduce the rates of ATRA-induced spina bifida in rats, which is correlated with reduced maternal liver N-Hcy levels. However, fetal brains, not the spinal neural tubes, were used to measure the changes of N-Hcy and ROS level. Does AHT administration actually reduce these signals and apoptosis in the rat neural tube during closure?

Minor concerns:

1. Increased copy numbers of MARSs are possibly associated with increased risks of either NTDs or CHDs, but the latter is not addressed further in this study. Related statement in the text should be softened.
2. It might be misleading to use "overexpression" of MARSs in normal developing or proliferating tissues. What's the normal function of MARSs in neural tube closure or organ development?
3. The manuscript may require professional editing service for typos or grammatical issues.

1st Revision - authors' response

6 March 2019

***** Reviewer's comments *****

Referee #1 (Remarks for Author):

This manuscript submitted by Mei et al describes a new signaling pathway in mediating the adverse effects of homocysteine (Hcy). Because reducing Hcy levels do not always effectively alleviate hyperhomocysteinemia-related pathology, the authors started to seek out additional molecular intermediators in the Hcy metabolic pathway. They focused on methionyl-tRNA synthetases (MARSs). Maternal blood of human NTDs and CHDs, several cell models and the all-trans retinoic acid (ATRA)-induced rat spinal bifida model were used in the comprehensive studies described in this paper. Increased copy numbers of both cytosolic-MARS and mitochondrial-MARS2 were associated with human NTDs and CHDs. Experimental data supported the hypothesis that MARSs converted Hcy into a reactive intermediate Hcy thiolactone (HTL), which inactivated two antioxidant enzymes, SOD1 and SOD2 through lysine homocysteinylation (N-Hcy). Most interestingly, they developed a small molecule acetyl Hcy thioether (AHT) in inhibiting MARSs activity. AHT reduced ATRA-induced NTDs in the rat.

Convincible data support the conclusion that MARSs are critical parts of Hcy sensing and mediate the teratogenicity of ATRA. 1. The findings are novel and the development of AHT in targeting MARS activity may have broad implications on preventing hyperhomocysteineemia-related diseases. 2. Revealing N-Hcy-modified SOD1 and SOD2 provided new mechanistic insights for Hcy-induced oxidative stress, apoptosis and growth inhibition. Overall, the findings would advance the field in the origin and cause of birth defects.

However, there are concerns related to the low quality of some of the data, obscure methodology and the unconvinced link of the discovered pathway with increased canonical Wnt signaling.

Response: We appreciate the very positive assessment to our study by the reviewer. Following the reviewer's suggestion, we made changes as specified below in the revised manuscript.

Comment 1-1 The Abstract misses some of the key findings. Suggest re-writing to reflect the key points;

Response: Following the reviewer's suggestion, we had rewritten the abstract to include all the major findings, including those added during revision.

Comment 1-2 The information for the human study is unclear. Need a Table to specify the types of birth defects. When was maternal blood collected? Were Hcy and HTL levels determined? The correlation analysis is not very convincing for the relation between MARS and birth defects.

Response: Sorry for our unclear description. Following the suggestion from the reviewer, we provided the sample information of birth defect types as Appendix Table S1 in the revised manuscript. The blood samples were collected between 2010-2015 regardless of the maternal levels of Hcy and HTL. In the samples we analyzed, statistically significant MARS copy number variations (CNV) difference were found between CHDs/NTDs samples and control samples. We believe that more pronounced differences will be observed if the sample size of CHDs/NTDs can be bigger.

Comment 1-3 In Fig. 1A, where do the error bars come from if individual samples were quantified? Assay replicates do not represent biological replicates. Graph should depict averages and sample size. The rest panels of Figure 1 need quantification for intensities and Panel B needs a large magnification.

Response: We thank the reviewer for his data presentation suggestion. The original figure was a bit confusing, the upper panel is the western blot results of MARS expression in 6 pairs of samples and the lower panel was Q-PCR results of MARS2 mRNA levels of the same group of samples. Error bar indicated assay replicates. We thank the reviewer for suggesting a better way to present our data. We changed the original Fig. 1A to the format as the reviewer suggested (revised Fig. 1J, 1K and 1L) in the revised manuscript.

Comment 1-4 Several cell lines were used but not well justified. The closest one should be the NE4C cell line yet only few data generated from this cell line.

Response: Following the suggestions from the reviewer, we repeated most of the experiments in NE-4C cell line and recaptured the initial findings. Since this issue had been pointed out by reviewer #2 and #3, we performed the experiments in NE4C and replaced the original results in HEK293T cells with results from those from NE-4C cells in Fig 3A-K; Fig 4E-G; Fig 5A-I, Fig 6D, G-J; Fig EVA, B, D-H)

Comment 1-5 The method used for total ROS levels is not optimal. Preferably, the author should measure superoxide because SOD1 and SOD2 activity is impaired by the Hcy-MARS-N-Hcy pathway.

Response: We agree with the reviewer's opinion. We not only measured ROS using DCFH-DA staining, but also detected superoxide levels using DHE staining, we have repeated the DHE staining experiments in NE-4C cells and provided these data as Fig 3A-G and; Fig 6H; Fig 7J and P; Fig. EV 2A, B and E; Fig. EV4E in the revised manuscript.

Comment 1-6 There is entirely unclear how SOD1 and SOD2 activity were determined.

Response: We apologize for the confusion and we have provided details for the SOD activity assay in the revised methods.

Comment 1-7

Literature supports the notion that oxidative stress suppresses the canonical Wnt signaling pathway.

Response: Albeit there are reports that oxidative stress suppresses the canonical Wnt signaling pathway, existing references (Funato et al., 2006, 2010; Korswagen et al., 2006) also suggested that oxidative stress activates rather than suppresses the canonical Wnt signaling pathway. Oxidative stress causes dissociation of NRX from Dvl, which enables Dvl to activate the downstream Wnt signaling pathway. These findings are consistent with our findings that N-Hcy-induced ROS activate Wnt signaling pathway (Figure 5). We provided the links to these references at the end of rebuttal letter (Funato et al, 2006; Funato & Miki, 2010; Korswagen, 2006).

There are both increased beta-catenin levels and DKK1, the Wnt inhibitor.

Response: We agree with the reviewer that this is paradoxical. However, Niida et al. reported that DKK1, a negative regulator of Wnt signaling, is a target of the b-catenin/TCF pathway. (Niida et al, 2004) Our results in agree with their findings. Therefore, future studies are needed to better understand these phenomena. In order to avoid unnecessary confusion, we removed the DKK1 results in the revised Fig. EV2F.

The argument that Hcy and HTL decrease DVL1-NRX interaction is not well founded. Generally, the data do not strongly support the hypothesis that Hcy and HTL enhance Wnt signaling. The blots in Fig. 5E, F, G, H and I were saturated and the difference was hardly visible. Need data on the NE4C cell line.

Response: We appreciate the reviewer's careful assessment of our data. We have repeated these experiments in NE4C cells and obtained consistent results with our previous data. We replaced the old results of Fig. 5E, F, G, H and I with new ones which were repeated in NE4C cells in the revised manuscript.

Comment 1-8 In Fig. 6, Panel B, the right blot may be a failed run because the Hcy signals were not clearly visible.

Response: We adjusted the contrast of Fig. 6, Panel B to make the dots visible.

Comment 1-9 In the rat study, given the inhibitor AHT at embryonic day 10.5, which is half way through neurulation and maybe completed posterior neural tube closure, was not justified. In the text, it said that AHT treatment is along with ATRA, which was given from E6.5. Most of the measurement was done in the brain while ATRA exclusively induces spinal bifida. Suggest repeating the experiment by measuring the endpoints in neurulation stage embryos.

Response: It has been reported that Fusion of the neural folds is the most prominent process during the stage between E10.5 and E10.75 of rats(Christie, 1964; Freeman, 1972; Sakai, 1989). Not only spinal bifida but also cephalic malformations were observed in our ATRA induced NTDs model, we supplied these data in Revised Fig. EV4C and Table EV2. Following the review's suggestion, we also detected b-catenin levels in spinal cord (revised Appendix Fig S4B). We agree with the reviewer that it is more proper to give ATRA earlier, so we treated C57/B6 mice at E6.75 and detected N-Hcy, SOD activities, ROS, and b-catenin levels at E8.5 and E10 during neurulation in both cranial and spinal neural tubes (Fig 7A and J, Fig EV4G).

Referee #2 (Comments on Novelty/Model System for Author):

There are many experimental methods, or how experimental results were normalized, that are not adequately described. These are pointed out in the comments to the authors.

Referee #2 (Remarks for Author):

This is an interesting study to test the hypothesis that methionyl-tRNA synthetases (MARSs) gain of function (in patients by increased gene copy numbers) increase neural tube defects (NTDs) and congenital heart defects (CHDs), not through increased substrate for protein synthesis, but through sensing homocysteine (Hcy) and modifying superoxide dismutases (SODs) by lysine homocysteinylation (N-Hcy). SOD1/2 N-Hcy thereby decreases their activities and increases reactive oxygen species (ROS), which are associated with NTDs and perhaps CHDs. The progression of several in vitro (cell free and in cultured cells) and in vivo (animal model) experiments is logical, although the experiments were designed to prove the hypothesis correct, and do not consider alternative explanations or activities of MARSs and involvement in NTDs and CHDs. The authors also should address that, while ROS induced by altered metabolism, such as maternal diabetes, perhaps folate deficiency, retinoic acid, and some drugs, may be sufficient to cause NTDs and CHDs, there may be other genetic and environmental causes of these defects that do not increase ROS. Many of the experimental methods and reagents need more detailed descriptions, so it is not possible at this point to evaluate the validity of several of the experimental results. Although the decrease in NTDs caused by all-trans retinoic acid (ATRA) by a small molecule MARSs inhibitor in the rat is consistent with the hypothesis (although effects on CHDs or litter sizes were not reported), it cannot be determined whether this was due to, or just correlated with, restoration of SOD1/2 activities and resulting ROS levels, because treatment of a group of ATRA-treated rats with an antioxidant was not included. The manuscript, as well as the supplement,

needs better proof reading for spelling, punctuation, grammar, missing words, and spaces between text and reference citations. Care needs to be taken to accurately cite the literature and to include additional references in some cases.

Response: We are grateful to the reviewer for his/her very careful reading, positive assessments and very detailed corrections to the language of our manuscript. We have addressed every question the reviewer raised, include: 1, added alternative explanations in the discussion session as the reviewer suggested; 2, added details to the results; 3, added the results of an antioxidant (N-acetyl Cysteine) inhibited NTDs and CHDs.; 3 added in the discussion that ROS elevation by N-Hcy may not be the sole cause of NTDs and CHDs.

Detailed responses are as following:

Specific comments:

Comment 2-1 Although inhibition of MARSs with a small molecule reduced (but did not prevent) NTDs in some of the embryos of ATRA-treated rats, there may be some causes (genetic or environmental) of NTDs that are not mediated by MARS activities and in which MARSs inhibition would be ineffective. Therefore, the title of the manuscript seems over stated and should be revised.

Response: We fully agree with the reviewer's point. Following his/her suggestion, we modified the title of the manuscript to "Inhibiting Methionyl-tRNA synthetase reduces neural tube defect and congenital heart defect onsets."

Comment 2-2 If the hypothesis is correct, there should have been an increase in CHDs in the embryos of ATRA-treated rats, unless ATRA does not stimulate MARSs activities in embryonic heart progenitors. More information needs to be provided regarding effects of ATRA and AHT (the MARS inhibitor) on fetal CHDs and on litter sizes (because fetuses with CHDs might die before E18.5), and what effects, if any, ATRA has on MARSs activities in embryonic heart.

Response: As the reviewer correctly projected, we did observe reduced litter size (revised table EV2) and mal-developments such as TGA and smaller heart size in litters of 50mg/kg ATRA treated rats as shown in the blow. Moreover, these mal-developments were reduced by AHT. In light of the reviewer's suggestion, we reported that ATRA increase MARS protein levels in developing heart (revised Fig 7D), which was accompanied by CHDs such as TGA, DORV (revised Fig. 7F, detected by ink injection in the RV), and SVD, VSD (Fig. 7G, detected by HE staining) in the revised manuscript as Figure7. AHT reversed such defects in the revised Table 4.

Comment 2-3 The results in the rat model are consistent with the hypothesis, but it cannot be concluded that MARSs effects on SOD1 or 2 activities are due to ROS, leading to NTDs unless it is shown that antioxidants (vitamin E or a glutathione precursor) have the same effect to reduce NTDs as AHT in a group of ATRA-treated rats.

Response: We thank the reviewer for his thoughtful suggestion. In fact, a parallel study carried out in our lab showed that N-acetylcysteine (NAC), a widely used antioxidant, reduced ATRA induced ROS levels (showed in below) and NTDs onsets in rats, although we didn't show these data in the original submission. We have included these results as Figure EV4E and Table EV2 in the revised manuscript. Furthermore, our new data in mice ATRA model and chicken Hyperhomocysteinemia model also showed that AHT have similar effects on ROS levels and NTDs rates as NAC (Fig. 7 and Table 4, 5, 6.), suggested that MARSs effect on SOD1 or 2 activities and results in elevated ROS, leading to NTDs.

Comment 2-4 Abstract, line 6: "Increased copy numbers of cytosolic-MARS and/or mitochondrial-MARS2" should be followed by "-encoding genes". Please take care to italicize when referring to genes.

Response: We had changed the wording as the reviewer suggested in the revised manuscript.

Comment 2-5 Same sentence: "were detected in blood samples of patients with neural tube defects (NTDs) and congenital heart defects (CHDs)". It should be stated in how many patients out of how many screened.

Response: Increased copy numbers of MARS and/or MARS2 encoding genes were detected in blood samples of 196 patients with neural tube defects (NTDs) and 100 patients with congenital heart defects (CHDs). We didn't include the information of out of how many screened because our sample were obtained from hospital collection over few year and it is hard to find the total number of patients screened.

Comment 2-6 Introduction, first sentence, Braly & Holford, 2003 is not a retrievable reference. The claim in the first sentence is not what the title of the reference says. It is advised to state what Hcy disorders are associated with Hcy (too much or too little?) and to provide valid references.

Response: Sorry for our improper description, in order to avoid confusion, this sentence and the reference were deleted in the revised manuscript.

Comment 2-7 Hyperhomocysteinemia (HHcy) is not a nutritional disorder. Do the authors mean "metabolic disorder"? If they claim that it is common, they need to state the prevalence.

Response: Thanks for the suggestion. We agree with the reviewer and we had changed the "nutritional disorder" to "metabolic disorder". HHcy prevails as high as 66.4% to 76.2% in elderly women and men, respectively (Janson et al, 2002). We state the prevalence in the revised introduction.

Comment 2-8 "which are caused by insufficient development of the neural tube or the heart". "Insufficient development" should be replaced with "maldevelopment". There may be insufficient development in some (but not necessarily all) of these defects. However, at most, it can be claimed that insufficient development is associated with these defects but not the cause of them.

Response: We agree with the reviewer and we had changed it to maldevelopment in the revised manuscript.

Comment 2-9 P 4: "Being proteogenic enzymes that charges methionyl-tRNA, MARS and MARS2 expression can be elevated in proliferating cells, including actively differentiating embryonic cells and cancer cells, the latter having been confirmed in human colon cancer (Kushner et al, 1976)." References need to be provided for proliferating and embryonic cells.

Response: We apology for the confusion. In the revised manuscript, we changed this statement to "*MARS and MARS2 expression are elevated in cancer(Kushner et al, 1976)). There are many similarities such as fast cell proliferating and actively differentiating between tumor progression and embryo development(Pierce, 1983). As MARS play critical roles in protein synthesis, we suspected that MARS and MARS2 expression can be elevated during embryo development.*" In our study, our data confirmed that "MARS and MARS2 levels in proliferating chorion cells derived from human embryos were higher than in their corresponding maternal deciduae cells (Fig 1J-L). Moreover, mice Mars and Mars2 levels were higher in fetal rat brains than in the corresponding mother's brains (Fig 1M-O)."

Comment 2-10 Next sentence:" Notably, MARS and MARS2 gene duplication had been associated with increased ROS (Bayat et al, 2012), a consistent finding in NTDs (Weksler-Zangen et al, 2003)." I think the authors mean that increased ROS are a consistent finding in NTDs. However, this is not true (there can be causes of NTDs independent of increased ROS), and this not an appropriate reference for this claim. Diabetes can increase ROS, and maternal diabetes-induced ROS can induce NTDs, but this paper did not show this.

Response: Following the reviewer's suggestion, we had changed the description to "*Notably, MARS and MARS2 gene duplication have been associated with increased ROS(Bayat et al, 2012).*"

Oxidative stress-induced aberrant apoptosis may result in NTDs (Covarrubias et al, 2008) and CHDs (Wang et al, 2015) and is believed to be the pathologically causal mechanism of HHcy." in the revised manuscript.

Comment 2-11 Same sentence: "cancers (Trachootham et al, 2009)". This paper doesn't say this. Also, although ROS-induced DNA damage can cause cellular transformation, ROS induced by chemotherapy or radiation can kill cancer cells.

Response: Thank you for your correction. We deleted "cancers" and associated reference here.

Comment 2-12 Same sentence: "that is associated with increased apoptosis" needs a reference. If Dhalla, et al., preceding the clause is the reference, the sentence should be rewritten. However, it should be noted that if ROS-induced apoptosis is involved in cancer, were true in cancer, cells would die, not thrive.

Response: We have changed this sentence to "*Oxidative stress-induced aberrant apoptosis may result in NTDs (Covarrubias et al, 2008) and CHDs (Wang et al, 2015)*" and provided the relative reference.

Comment 2-13 Results, p 5, first paragraph: "we collected blood samples from clinically confirmed NTDs and CHDs cases known to be induced by HHcy" How could this be known? Or, do the authors mean "associated with"?

Response: We apologize for the confusion. The reviewer was correct that we meant "associated with". In order to avoid confusion, we changed the sentence to "we collected blood samples from clinically confirmed NTDs and CHDs cases".

Comment 2-14 Is it known whether or not the amplified copies of MARS or MARS2 are wild type (in coding and regulatory elements)? This should be addressed.

Response: We sequenced MARSs with amplified copies and found no mutations in coding and upstream non-coding regions of them. We changed the last sentence of this paragraph to "These observations, together with that all amplified MARSs bear no mutation, and an absence of MARS or MARS2 copy number loss being detected in samples from either birth defect cohort, indicated that increased MARS or MARS2 copy numbers are positively associated with increased risks of either NTDs or CHDs." in the revised manuscript.

Comment 2-15

1) P 6: "The increased copy number of MARSs in NTDs and CHDs indicated that overexpression of MARSs may deregulate development of these organs." NTDs and CHDs are not organs. Instead ""may deregulate development of the neural tube and the heart".

2) Also, it does not necessarily follow that increased copy numbers of MARSs caused overexpression of the encoded proteins, especially if the amplified copies were not wild type. It is possible that increased copies disrupted genes that are necessary for normal development of the neural tube and the heart.

Response:

a, We changed this statement to "*The increased copy number of MARSs in NTDs and CHDs indicated that increased MARS levels may deregulate development of the neural tube and the heart.*" following the reviewer's suggestion.

b, The amplified copies of MARS are wild type. Increased copy numbers were often associated with elevated protein expression (Heidenblad et al, 2005; Hyman et al, 2002). In the revised manuscript, we provided data that both mRNA (Fig 1A and B) and protein expression levels (Fig 1C-F) increased proportionally with copy number of MARS and MARS2.

Comment 2-16 P 7: How was growth of neuronal NE4C stem cells measured? What are the units?

Response: We measured growth of neuronal NE4C stem cells using the xCELLigence RTCA MP instrument with E-plate, which is normally used to monitor cell attachment and growth in real time. NE4C stem cells were seeded on E-Plate at a density of 1×10^4 cells/well and incubated overnight. These details were included in the methods section of the revised manuscript.

Comment 2-17 How was BCL2 over-expressed? How was prevention of apoptosis determined?

Response: BCL2 was overexpressed by transfecting BCL2-Flag plasmids. As the anti-apoptotic effect of BCL2 is widely reported, we didn't detect apoptosis levels in the original submitted data.

We repeated this experiment and examined the prevention of apoptosis after BCL2 over-expressed by detecting cleaved-caspase 3. We provide these details in revised Fig. 2B

Comment 2-18 "showing that Hcy, but not Met, inhibits cell growth mainly by augmenting apoptosis." "showing" should be replaced with "suggesting".

Response: Thanks for the correction. We replaced "showing" to "suggesting" as suggested by the reviewer.

Comment 2-19 How was MARS or MARS2 over-expressed? How much siRNA was transfected?

Response: MARS and MARS2 cDNA were cloned in pRK7-Flag vector to generate expression plasmids. Polyethylenimine (PEI, CAS Number: 9002-98-0, hedeBio) transfection reagent was used for transfecting HEK293T cells. Lipofectamine® 3000 transfection reagent (L30075, Thermo Fisher) were used for transfecting the MARS and MARS2 plasmids into the NE4C cells. For siRNA experiments, 4µg siRNA were transfected using the Lipofectamine™ 2000 Transfection Reagent (11668030, Thermo Fisher). These details were provided in the methods section of the revised manuscript.

Comment 2-20 "Data showed that oxidative stress related pathways were enriched". Oxidative stress-related pathways were only some of the enriched pathways. This should be made clear. Perhaps, "oxidative stress-related pathways were among the pathways that were enriched."

Response: We thank the reviewer for tutoring our English expression. We have changed the sentence to "oxidative stress-related pathways were among the pathways that were enriched."

Comment 2-21 Why were HEK293T all of a sudden used instead of NE4C cells for Fig. 2G? Rationale for use of one cell type or the other for different experiments should be provided throughout the manuscript.

Response: Thank you for your suggestion. We repeated most of the experiments including Fig.2G in NE-4C cells and provided them in the revised manuscript.

Comment 2-22 Why is the format of Fig. 3D different from A-C? What do #1, 2, and 3 refer to?

Response: The format of the originally submitted 3D was different from 3A-C because siRNA was used (#1,2,3 meant different siRNA oligos were used) in 3D. We replaced the 3D in the revised manuscript with a shRNA result to keep the format consistent with 3A-C.

Comment 2-23 Fig. 3 and subsequent, it's not clear at all how ROS were assayed, and what they were expressed relative to.

Response: In the original submitted Fig3, ROS levels were measured by monitoring the fluorescence of DCFH-DA (S0033, Beyotime) stained cells (lex = 485 nm, lem = 620 nm) following the manufacturer's instruction. Briefly, cells were harvested from triplicate or quadruplicate cultures, resuspended and incubated with DCFH-DA for 20 min in the dark, washed twice with PBS solution, and fluorescence recorded. All the ROS levels were expressed as relative levels to controls, which levels were set as 100% arbitrarily. When repeated these data in NE-4C cells, combining the suggestion of the other reviewer, we detected the superoxide by monitoring the fluorescence of dihydroethidium (DHE, S0063, Beyotime) stained cells (lex = 300 nm, lem = 610 nm). The details were provided in the revised Fig. 3 and the methods section of the revised manuscript.

Comment 2-24 Fig. 3 and others: What meant by %HTL, Hcy, N-Hcy levels? What is the denominator?

Response: The %HTL, Hcy, N-Hcy levels were all normalized to those of untreated groups, which were set as 100% arbitrarily. We indicated this in the figure legends of the revised manuscript.

Comment 2-25 Fig. S2C needs more controls to validate the anti- N-Hcy antibody, such as an anti-BSA ab, and secondary ab alone.

Response: Following the suggestion from the reviewer, we further validated the N-Hcy antibody using anti-BSA antibodies and found that the K-Hcy added in the antibody dilution only compete the signals of N-Hcy but not the BSA signals. These results were added to the revised Appendix Fig.S2A. Furthermore, we have reported that this anti-N-Hcy antibody recognize N-Hcy-OVA, but doesn't recognize OVA, Ac-OVA and suc-OVA(Mei et al, 2016; Zhang et al, 2018), suggested that the N-Hcy antibody is specific.

Specificity of the anti-K_{Hcy} antibody. Immunoblotting assay was carried out by incubating the anti-K_{Hcy} antibody with unmodified OVA, Suc-OVA, Ac-OVA, or N-Hcy-OVA (Mei et al, 2016).

Verification of specificity of the anti-K_{Hcy} antibody. Western blotting assay was carried out by incubating the anti-K_{Hcy} antibody with unmodified OVA (ovalbumin), acetylated-OVA, succinylated-OVA, or K-Hcy-OVA. (Zhang et al, 2018)

Comment 2-26 How were SOD1/2 activities assayed?

Response: The SOD1 and SOD2 activity assay was carried out with the SOD Assay Kit (Dojindo Molecular Technology Inc.), which use Dojindo's highly water-soluble tetrazolium salt (WST-1) as a substrate and proceed according to the manufacturer's instruction. Briefly, WST-1 produces a water-soluble formazan dye upon reduction with a superoxide anion. The rate of the reduction of O₂ free radical is linearly related to the xanthine oxidase (XO) activity, and is inhibited by SOD. The inhibition activity of SOD or SOD-like materials can be determined by a colorimetric method. These details were provided in the methods section of the revised manuscript.

Comment 2-27 "Hcy-mimetic" meaning isn't clear in the figure 4 legend. Also, the legends don't make it clear which assays were performed *in vitro* and which *in vivo*.

Response: Sorry for our unclear description. Hcy-mimetic indicated the K-W mutants, which simulate K-Hcy modification by bulky side chain size. We made this clear in the revised text. *in vitro* means Hcy/HTL treatment in tubes and *ex vivo* means Hcy/HTL treatment in cultured cells. We added more details to make the *in vitro* and *ex vivo* clearer in the revised figure legend.

Comment 2-28 The legend for Fig. 5B says Hcy, not HTL. Do the authors mean "transcription of" or "transactivation by"?

Response: The reviewer was correct. We have changed the "transcription" to "transactivation".

Comment 2-29 The Topflash assay needs a description and/or reference.

Response: Following the suggestion from the reviewer, we added the details of Topflash assay to the methods session: "TCF/LEF reporter assay was used to examine the effect of Hcy/HTL on WNT-induced transcriptional activity. TOPFLASH and FOPFLASH plasmids were transfected in NE4C cells. 4 h after Hcy/HTL treatment, reporter activities were measured with a plate reader (LD400; Beckman Coulter, Fullerton, CA) using a dual luciferase reporter assay system (Promega). TCF/LEF activity was defined as the ratio of TOPFLASH: FOPFLASH reporter activities."

Comment 2-30 More details need to be provided for methods for entire Fig. 6 and Fig. S7.

Response: Following the suggestion from the reviewer, we have provided the details of methods in the revised methods.

Comment 2-31 "The abilities of AHT to reverse Hcy effects were further tested." Is not clear from the subsequent panels in Fig. 6. Do the authors mean the production of Hcy or the effects of Hcy? Or both?

Response: We meant the effect of Hcy, and we have changed the description "The abilities of AHT to reverse the effects of Hcy were further tested." in the revised manuscript.

Comment 2-32 "An all-trans retinoic acid (ATRA) NTDs-complicated pregnancy rat model, which is thought to induce NTDs in rats partly by increasing Hcy (Xu et al, 2010) and ROS concentrations (Mantymaa et al, 2000; Mathieu et al, 2005)". All of these references provide evidence that ATRA increases Hcy and/or ROS in different adult cancer cells, but none of them provide evidence that ATRA induces NTDs, or that ATRA-induced NTDs by increasing Hcy and/or ROS. Appropriate references should be provided or this statement should be revised.

Response: We apologize for the confusion. Our experiments results showed that ATRA treatment increase Hcy levels in the liver homogenates of pregnant rats and mice (revised Fig EV4A and B). However, references for ATRA increases Hcy and/or ROS in NTDs or other birth defects are lacking. We altered the statement to "All-trans retinoic acid (ATRA) NTDs-complicated pregnancy rat (Li et al, 2012) and mice models (Zhang et al, 2009) were employed." in the revised manuscript.

Comment 2-33 Fig. S7C is of poor quality. One can't see spina bifida in most of the fetuses, even upon zooming in.

Response: Spina bifida pictures with higher qualities were provided in the revised Fig EV3C.

Comment 2-34 The Discussion could be significantly shortened. 1-1.5 pages should be sufficient.

Response: We have shortened the discussion during the revision.

Referee #3 (Remarks for Author):

In this manuscript Mei et al report the potential roles of methionyl-tRNA synthetase (MARS)-generated homocysteine signals in neural tube defects (NTDs). The major findings include that increased copy numbers of MARS/MARS2 in patients are associated with neural tube defects and congenital heart defects. In vitro experiments suggest that overexpression of MARSs potentiates Hcy-induced ROS accumulation and apoptosis, possibly acting through sequential regulation of related intermediates HTL, N-Hcy, and SOD1/2. MARSs overexpression also upregulates beta-catenin expression level. SOD1/2 overexpression partly reverses Hcy- or HTL-induced beta-catenin accumulation and abolishes the ability of HTL in decreasing DVL1-NRX interaction in a kidney cell line. Finally, they show that an Hcy analog AHT inhibits MARSs, HTL, N-Hcy, and ROS productions in vitro and decreases NTD rates induced by all-trans retinoic acid in rats. This is an important study with attractive in vivo results and abundant in vitro biochemical data, which may reveal the function and related mechanism of MARSs in homocysteine-induced neural tube defects. A few concerns might need to be addressed or clarified.

Response: We thank the reviewer for his/her positive assessment to our study. We had addressed all his questions and suggestions during revision.

Major concerns:

Comment 3-1 Figures 1 and S1 show MARSs expression in developing or proliferating cells of human placenta, embryonic monkey/rat brain sections, and urothelial cell carcinoma cells, none of these are related to neural tube closure.

Response: We fully agree with the reviewer that these data were not neural tube closure-related. Our initial logic was that neural tube closure is a process of cell proliferating, therefore, the positive correlation of MARSs expression to the proliferation of cells may show the importance of MARSs expression in organs/tissues development. Following the reviewer's suggestion, we have provided double IHC staining for MARS and Ki67 (proliferation marker) of E8.5, E9, and E10 neural tube sections (revised Fig 1G and H, Fig EV1A). MARS were co-localized with Ki67 during neural tube closure.

Comment 3-2 Most mechanistic studies were carried out in cell lines that are not related to neural tube defects at all. Where are MARSs expressed in vivo during neural tube closure? Do MARSs actually regulate SOD1/2, ROS, and apoptosis in neural tube closure?

Response: We thank the reviewer for his/her thoughtful questions. First, following the reviewer's suggestion and the suggestion from reviewer #1 and review #2, we repeated many of the cell line-based experiments in NE4C, which may related to neural tube defects and successfully recaptured the findings we got from the cells we previously used. Second, we have detected MARS expression using IHC staining in E8.5, E9, E10 neural tube sections. We found that MARS were constitutively

expressed in neural tube, especially highly expressed in the neural epithelial cells (or neural progenitor cells) during neural tube closure. Our *extro vivo* experiments in NE4C showed that MARS overexpression promotes SOD N-Hcy and increase ROS and apoptosis levels, while knock-down MARS abolished Hcy's effects. Our *in vivo* data in the revised Fig 7 showed that MARS levels were elevated in ATRA treated mice neural tube tissues, accompanied by elevated SOD N-Hcy (Fig. 7I), ROS and apoptosis levels (Fig 7J, Fig EV4G), which were detected by western blot, DHE and TUNEL staining, respectively.

Comment 3-3 MARS overexpression increased beta-catenin and Tcf levels in cell lines. Does overactivation of Wnt/beta-catenin signaling cause NTDs?

Response: Wnt-beta-catenin signaling is tightly controlled during development. Both gain-of-function (GOF) and loss-of-function (LOF) of beta-catenin signaling can result in NTDs (Grigoryan et al, 2008). It has been reported that over-accumulation of beta-catenin and corresponding overactivated TCF and apoptosis results in NTDs (Gray et al, 2010; Li et al, 2015). Overactivation of Wnt/beta catenin signaling also causes CHDs (Hurlstone et al, 2003). In light of the reviewer's suggestion, we included these references in the revised manuscript.

Comment 3-4 Mechanistic and prevention studies using animal models of hyperhomocysteinemia may dramatically enhance the significance of their findings.

Response: We fully agree with the reviewer's point. As chicken hyperhomocysteinemia model were widely used to study Hcy induced NTDs and CHDs, we employed this model and found that AHT as well as NAC treatments reduced NTDs and CHDs onsets in the chicken embryo HHcy model. Moreover, Hcy increased while AHT decreased the protein N-Hcy levels and ROS levels in chicken embryos at 72h after incubation. These data strongly supported our claims and we included them as Table 5 and 6, Fig 7 O- Q in the revised manuscript.

Comment 3-5

It is quite impressive that maternal administration of the MARS inhibitor AHT can reduce the rates of ATRA-induced spina bifida in rats, which is correlated with reduced maternal liver N-Hcy levels. However, fetal brains, not the spinal neural tubes, were used to measure the changes of N-Hcy and ROS level. Does AHT administration actually reduce these signals and apoptosis in the rat neural tube during closure?

Response: Thanks for the reviewer's good suggestion. We have detected b-catenin levels in the rat spinal cord samples at E18.5, and we provided this data in the revised Appendix Fig S4B. Furthermore, we fully agree with the reviewer that it is more proper to collect the neural tube closure-related data. During the revision, as both cranial and spinal neural tube defects were observed in the mice ATRA model treated at E7, we detected the N-Hcy, ROS and apoptosis levels at E8.5 and E10 mice both in the cranial and spinal neural tube sections (revised Fig 7J, Fig EV4G).

Minor concerns:

Comment 3-6 Increased copy numbers of MARSs are possibly associated with increased risks of either NTDs or CHDs, but the latter is not addressed further in this study. Related statement in the text should be softened.

Response: During the revision, combing the suggestion of the reviewer and reviewer #2, we studied CHDs in ATRA induced mice model and chicken Hyperhomocysteinemia model. CHDs such as TGA, DORV (revised Fig. 7F), and SVD, VSD (revised Fig. 7G) ATRA treatments increase the MARS expression levels (Fig 7B) and Hcy levels (Fig EV4A). Inhibiting MARSs by AHT treatment or reducing ROS by NAC treatment reduced the CHDs onset both in ATRA and HHcy models (Table 4 and 6). We included these data in the revised manuscript.

Comment 3-7 It might be misleading to use "overexpression" of MARSs in normal developing or proliferating tissues. What's the normal function of MARSs in neural tube closure or organ development?

Response: Following the reviewer's suggestion, we have changes the "overexpression" to "elevated levels". The canonic functions of MARSs are to charge tRNA for protein synthesis. MARS has been identified as one of the essential genes for early zebrafish development (Amsterdam et al, 2004). Loss-of-function methionyl-tRNA synthetase mutations presented with a multi-organ dysfunction (van Meel et al, 2013). Heterozygous MARS mutations have been reported to cause

Charcot-Marie-Tooth disease(Niehues et al, 2015), axonal, type 2U (CMT2U). Homozygous or compound heterozygous mutations in MARS gene would cause interstitial lung and liver disease (ILLD), a severe disease onset in infancy or early childhood(Sun et al, 2017). As MARSs bind homocysteine and convert it to homocysteine thiolactone that modifies lysine to form N-Hcy is an error-checking process that prevent incorrect incorporation of Hcy to proteins. Deregulated MARSs levels may have the similar effects as deregulated homocysteine to induce an array of hyperhomocysteinemia- associated diseases such as NTDs and CHDs.

Comment 3-8 The manuscript may require professional editing service for typos or grammatical issues.

Response: Following the reviewer's suggestion, we asked a native English speaker who is a professor at Balor Medical School polished the English of our manuscript.

References mentioned in the letter:

Amsterdam A, Nissen RM, Sun Z, Swindell EC, Farrington S, Hopkins N (2004) Identification of 315 genes essential for early zebrafish development. *Proceedings of the National Academy of Sciences of the United States of America* **101**: 12792-12797

Bayat V, Thiffault I, Jaiswal M, Tetreault M, Donti T, Sasarman F, Bernard G, Demers-Lamarche J, Dicaire MJ, Mathieu J, Vanasse M, Bouchard JP, Rioux MF, Lourenco CM, Li ZH, Haueter C, Shoubridge EA, Graham BH, Brais B, Bellen HJ (2012) Mutations in the Mitochondrial Methionyl-tRNA Synthetase Cause a Neurodegenerative Phenotype in Flies and a Recessive Ataxia (ARSAL) in Humans. *Plos Biol* **10**

Christie GA (1964) Developmental Stages in Somite and Post-Somite Rat Embryos, Based on External Appearance, and Including Some Features of the Macroscopic Development of the Oral Cavity. *Journal of morphology* **114**: 263-283

Covarrubias L, Hernandez-Garcia D, Schnabel D, Salas-Vidal E, Castro-Obregon S (2008) Function of reactive oxygen species during animal development: passive or active? *Developmental biology* **320**: 1-11

Freeman BG (1972) Surface modifications of neural epithelial cells during formation of the neural tube in the rat embryo. *Journal of embryology and experimental morphology* **28**: 437-448

Funato Y, Michiue T, Asashima M, Miki H (2006) The thioredoxin-related redox-regulating protein nucleoredoxin inhibits Wnt-beta-catenin signalling through dishevelled. *Nature Cell Biology* **8**: 501-U135

Funato Y, Miki H (2010) Redox regulation of Wnt signalling via nucleoredoxin. *Free Radical Res* **44**: 379-388

Gray JD, Nakouzi G, Slowinska-Castaldo B, Dazard JE, Rao JS, Nadeau JH, Ross ME (2010) Functional interactions between the LRP6 WNT co-receptor and folate supplementation. *Human molecular genetics* **19**: 4560-4572

Grigoryan T, Wend P, Klaus A, Birchmeier W (2008) Deciphering the function of canonical Wnt signals in development and disease: conditional loss- and gain-of-function mutations of beta-catenin in mice. *Genes & development* **22**: 2308-2341

Heidenblad M, Lindgren D, Veltman JA, Jonson T, Mahlamaki EH, Gorunova L, van Kessel AG, Schoenmakers EF, Hoglund M (2005) Microarray analyses reveal strong influence of DNA copy number alterations on the transcriptional patterns in pancreatic cancer: implications for the interpretation of genomic amplifications. *Oncogene* **24**: 1794-1801

Hurlstone AF, Haramis AP, Wienholds E, Begthel H, Korving J, Van Eeden F, Cuppen E, Zivkovic D, Plasterk RH, Clevers H (2003) The Wnt/beta-catenin pathway regulates cardiac valve formation. *Nature* **425**: 633-637

- Hyman E, Kauraniemi P, Hautaniemi S, Wolf M, Mousses S, Rozenblum E, Ringner M, Sauter G, Monni O, Elkahlon A, Kallioniemi OP, Kallioniemi A (2002) Impact of DNA amplification on gene expression patterns in breast cancer. *Cancer research* **62**: 6240-6245
- Janson JJ, Galarza CR, Murua A, Quintana I, Przygoda PA, Waisman G, Camera L, Kordich L, Morales M, Mayorga LM, Camera MI (2002) Prevalence of hyperhomocysteinemia in an elderly population. *Am J Hypertens* **15**: 394-397
- Korswagen HC (2006) Regulation of the Wnt/beta-catenin pathway by redox signaling. *Dev Cell* **10**: 687-688
- Kushner JP, Boll D, Quagliana J, Dickman S (1976) Elevated methionine-tRNA synthetase activity in human colon cancer. *Proc Soc Exp Biol Med* **153**: 273-276
- Li BI, Matteson PG, Ababon MF, Nato AQ, Jr., Lin Y, Nanda V, Matise TC, Millonig JH (2015) The orphan GPCR, Gpr161, regulates the retinoic acid and canonical Wnt pathways during neurulation. *Developmental biology* **402**: 17-31
- Li H, Gao F, Ma L, Jiang J, Miao J, Jiang M, Fan Y, Wang L, Wu D, Liu B, Wang W, Lui VC, Yuan Z (2012) Therapeutic potential of in utero mesenchymal stem cell (MSCs) transplantation in rat fetuses with spina bifida aperta. *Journal of cellular and molecular medicine* **16**: 1606-1617
- Mei XY, He XD, Huang L, Qi DS, Nie J, Li Y, Si W, Zhao SM (2016) Dehomocysteinylation is catalysed by the sirtuin-2-like bacterial lysine deacetylase CobB. *The FEBS journal* **283**: 4149-4162
- Niehues S, Bussmann J, Steffes G, Erdmann I, Kohrer C, Sun L, Wagner M, Schafer K, Wang G, Koerdts SN, Stum M, Jaiswal S, RajBhandary UL, Thomas U, Aberle H, Burgess RW, Yang XL, Dieterich D, Storkebaum E (2015) Impaired protein translation in Drosophila models for Charcot-Marie-Tooth neuropathy caused by mutant tRNA synthetases. *Nature communications* **6**: 7520
- Niida A, Hiroko T, Kasai M, Furukawa Y, Nakamura Y, Suzuki Y, Sugano S, Akiyama T (2004) DKK1, a negative regulator of Wnt signaling, is a target of the beta-catenin/TCF pathway. *Oncogene* **23**: 8520-8526
- Pierce GB (1983) The cancer cell and its control by the embryo. Rous-Whipple Award lecture. *The American journal of pathology* **113**: 117-124
- Sakai Y (1989) Neurulation in the mouse: manner and timing of neural tube closure. *The Anatomical record* **223**: 194-203
- Sun Y, Hu G, Luo J, Fang D, Yu Y, Wang X, Chen J, Qiu W (2017) Mutations in methionyl-tRNA synthetase gene in a Chinese family with interstitial lung and liver disease, postnatal growth failure and anemia. *Journal of human genetics* **62**: 647-651
- van Meel E, Wegner DJ, Cliften P, Willing MC, White FV, Kornfeld S, Cole FS (2013) Rare recessive loss-of-function methionyl-tRNA synthetase mutations presenting as a multi-organ phenotype. *BMC medical genetics* **14**: 106
- Wang F, Fisher SA, Zhong J, Wu Y, Yang P (2015) Superoxide Dismutase 1 In Vivo Ameliorates Maternal Diabetes Mellitus-Induced Apoptosis and Heart Defects Through Restoration of Impaired Wnt Signaling. *Circulation Cardiovascular genetics* **8**: 665-676
- Zhang Q, Bai B, Mei X, Wan C, Cao H, Dan L, Wang S, Zhang M, Wang Z, Wu J, Wang H, Huo J, Ding G, Zhao J, Xie Q, Wang L, Qiu Z, Zhao S, Zhang T (2018) Elevated H3K79 homocysteinylation causes abnormal gene expression during neural development and subsequent neural tube defects. *Nature communications* **9**: 3436

Thank you for the submission of your revised manuscript to EMBO Molecular Medicine, and please accept my apologies for the delay in reaching a decision, which is due to the fact that we received a complete set of reports only recently.

As you will see from the enclosed reports, the referees are now overall supportive of publication, but however still have concerns that should fully be addressed before acceptance of your manuscript in EMBO Molecular Medicine.

1) Referee #2 provided a detailed report of his/her concerns that you will find attached. These concerns can mostly be addressed by modifying the text and adding clarifications. Please be as precise and concise as possible when answering these comments in your rebuttal letter and in the manuscript. Please refrain from using terms as "smaller" (point #2) or "dose-dependently" (point #30) when no quantification/numbers are provided.

2) Regarding the points from referee #2 that should be experimentally addressed:
- point #21: the use of different cell lines should either be better discussed in the text or addressed experimentally.
- point #25: the staining controls should be performed.

3) Please address the concerns from referee #3 regarding the quality of the immunofluorescence pictures.

4) As mentioned by referees #2 and #3, the manuscript would benefit from professional editing service and we would strongly encourage you to use such a service.

5) Please see below and refer to our author guideline when submitting your revised manuscript (<http://embomolmed.embopress.org/authorguide>).

I look forward to reading a new revised version of your manuscript as soon as possible.

***** Reviewer's comments *****

Referee #1 (Remarks for Author):

The authors have addressed my previous concerns.

Referee #2 (Remarks for Author):

The authors have made many revisions to the manuscript, including additional cell culture studies using one, most relevant cell line, additional chick embryo experiments, details of experimental methods, and revised figures and legends. However, some concerns that were raised in the first review still remain, and some of the new or revised text raise additional concerns and would need to undergo further revision.

Detailed report:

1. From previous review: "Although inhibition of MARSs with a small molecule reduced (but did not prevent) NTDs in some of the embryos of ATRA-treated rats, there may be some causes (genetic or environmental) of NTDs that are not mediated by MARS activities and in which MARSs inhibition would be ineffective. Therefore, the title of the manuscript [Inhibiting Methionyl-tRNA synthetase prevents neural tube defects] seems over stated and should be revised." The authors responded that they "fully agree", but revised the title to "Inhibiting Methionyl-tRNA synthetase reduces neural tube defect and congenital heart defect onsets". They changed "prevents" to "reduces", but missed the point that there may be some causes of NTDs that are not mediated by MARS activities and in which MARSs inhibition would be ineffective. And, they added another

congenital malformation that, in addition to NTDs, was related to MARS in their subjects. So, while “reduces” may be more conservative than “prevents”, the title is even more overstated.

2. From previous review: “there should have been an increase in CHDs in the embryos of ATRA-treated rats,... More information needs to be provided regarding effects of ATRA and AHT (the MARS inhibitor) on fetal CHDs and on litter sizes (because fetuses with CHDs might die before E18.5), and what effects, if any, ATRA has on MARS activities in embryonic heart.” The authors responded that they did observe smaller litter sizes (revised Table EV2), however, they did not perform statistics of litter sizes or resorptions. They showed examples of CHDs (revised Fig 7F), but did not show quantitation. They performed immunofluorescence for MARS (Fig 7D), but this is not quantitative (and differences between ATRA+/- AHT are not obvious), and shows protein levels, not enzyme activities.

3. From previous review: “The results in the rat model are consistent with the hypothesis, but it cannot be concluded that MARS effects on SOD1 or 2 activities are due to ROS, leading to NTDs unless it is shown that antioxidants (vitamin E or a glutathione precursor) have the same effect to reduce NTDs as AHT in a group of ATRA-treated rats.” The authors responded that they had shown that N-acetylcysteine (NAC) [which can be used as a glutathione precursor, although “NAC” is never defined, and a reference that it is a GSH precursor or can be used as an antioxidant is not provided in the revised manuscript], but had not shown the results in the original manuscript. However, effects of NAC on ROS levels in E18.5 fetal brain (revised Fig. EV4F) are not quantitative, and are well after formation of the neural tube, so are not relevant. Effects of NAC on ATRA-induced NTDs (revised Table EV2) are acceptable. Effects of NAC in Figure 7 and Tables 4, 5, and 6 are acceptable.

4. From previous review: “Abstract, line 6: “Increased copy numbers of cytosolic-MARS and/or mitochondrial-MARS2” should be followed by “-encoding genes”. Please take care to italicize when referring to genes.” The authors said that they made these changes, but they did not.

5. From previous review: “Same sentence: “were detected in blood samples of patients with neural tube defects (NTDs) and congenital heart defects (CHDs)”. It should be stated in how many patients out of how many screened.” The authors said that they indicated the numbers of NTDs and CHDs in the revised manuscript, but they did not. They do not have access to the total number of patients screened, which is a weakness, as the prevalence of these defects in the population seen cannot be determined from either the Abstract or the Materials and Methods.

6. From previous review: “Introduction, first sentence, Braly & Holford, 2003 is not a retrievable reference. The claim in the first sentence is not what the title of the reference says. It is advised to state what Hcy disorders are associated with Hcy (too much or too little?) and to provide valid references.” The authors have deleted the sentence and the reference, which is acceptable. However, they added a new clause and references to the second sentence, “Hyperhomocysteinemia prevails as high as 66.4% to 76.2% in elderly women and men, respectively (Janson et al, 2002),” which is irrelevant. Elderly women and men can't be pregnant, and it cannot be determined if these incidences of hyperhomocysteinemia are due to nutrition, genetics, aging, etc. This sentence and reference detracts from the focus of the manuscript, which is on specific birth defects.

7. Revision of “nutritional disorder” to “metabolic disorder” is acceptable.

8. Replacement of “Insufficient development” with “maldevelopment” is acceptable.

9. From previous review: “P 4: “Being proteogenic enzymes that charges methionyl-tRNA, MARS and MARS2 expression can be elevated in proliferating cells, including actively differentiating embryonic cells and cancer cells, the latter having been confirmed in human colon cancer(Kushner et al, 1976).” References need to be provided for proliferating and embryonic cells.” The need for references, that had not been cited, was apparently due to original sentence structure, because, in fact, the authors meant (P 4, revised manuscript), “As MARS play critical roles in protein synthesis, we suspected that MARS and MARS2 expression can be elevated during embryo development.” So, the expression of MARS and MARS2 in embryonic cells was not known. But, according to the added sentence, elevation of MARS and MARS2 expression occurs in NORMAL embryo

development. So, shouldn't further increase due to gene duplication be protective of embryo development? This added sentence seems to contradict the hypothesis.

10. From previous review: "Notably, MARS and MARS2 gene duplication had been associated with increased ROS (Bayat et al, 2012), a consistent finding in NTDs (Weksler-Zangen et al, 2003)." I think the authors mean that increased ROS are a consistent finding in NTDs. However, this is not true (there can be causes of NTDs independent of increased ROS), and this not an appropriate reference for this claim. Diabetes can increase ROS, and maternal diabetes-induced ROS can induce NTDs, but this paper did not show this." The authors have revised the second part of the sentence and replaced the Weksler-Zangen reference: "Oxidative stress-induced aberrant apoptosis may results in NTDs (Covarrubias et al, 2008) and CHDs (Wang et al, 2015) and is believed to be the pathologically causal mechanism of hyperhomocysteinemia." However, Covarrubias, et al., is mostly about the normal effects of ROS in development, not NTDs, and Wang, et al., is one (but not the original) reference for ROS in diabetes-CHDs. The latter part of the sentence "and is believed to be the pathologically causal mechanism of hyperhomocysteinemia" is not referenced and is not true.

11. Deletion of part of sentence (citing Trachootham) is acceptable.

12. From prior review: "that is associated with increased apoptosis" needs a reference. If Dhalla, et al., preceding the clause is the reference, the sentence should be rewritten. However, it should be noted that if ROS-induced apoptosis is involved in cancer, were true in cancer, cells would die, not thrive." The authors replied that they "changed this sentence to 'Oxidative stress-induced aberrant apoptosis may results in NTDs (Covarrubias et al, 2008) and CHDs (Wang et al, 2015)' and provided the relative regergence [sic]". However (see Comment #10), this sentence and references are not acceptable.

13. Deletion of "known to be induced by HHcy" is acceptable.

14. Response to whether amplified copies of MARS or MARS2 carry sequence mutations and revision in text is acceptable.

15. Response to "The increased copy number of MARSs in NTDs and CHDs indicated..." is acceptable.

16. From prior review: "P 7: How was growth of neuronal NE4C stem cells measured? What are the units?" The authors have described how growth was assayed, but the units ("Cell Index" in the revised manuscript) are not defined.

17. From prior review: "How was BCL2 over-expressed? How was prevention of apoptosis determined?" The authors replied "BCL2 was overexpressed by transfecting BCL2-FLAG plasmids." However, there is no information about the source or construction of the plasmid or transfection conditions in either the figure legends or Materials and Methods to allow critical evaluation or replication of the experiments. Assay of apoptosis is described (by cleavage of BCL2, according to the authors' response, and by flow cytometry or TUNEL assay, according to the revised figure legends or Materials and Methods). While flow cytometry and TUNEL methods are adequately described, assay of BCL2 cleavage is not adequately described or referenced to allow critical evaluation of the data or replication of the experiments. Even though BCL2 cleavage is an established method, it is essential that this information be provided.

18. Replacement of "showing" with "suggesting" is acceptable.

19. From prior review: "How was MARS or MARS2 over-expressed? How much siRNA was transfected?" The authors responded that MARS and MARS2 cDNA were cloned into pRK7-Flag vector, and described the lipofection reagent used. However, the amount of plasmid transfected was not described in the manuscript. The amount of siRNA, or how transfected were not described in the methods section of the revised manuscript, contrary to the authors' response. Nor were the siRNA their sequences, or how long cells were transfected before assay (which I did not ask for), which are important for assessment and replication of experiments.

20. Revision of “Data showed that oxidative stress related pathways were enriched” to “oxidative stress-related pathways were among the pathways that were enriched.” is acceptable.
21. One of the other reviewers, and I, questioned switching between HEK293 and NE4C cells. The authors repeated many of the cell culture experiments using NE4C cells, which is appropriate. However, both the Materials and Methods and the Supplement refer to NE4C, H9C2, HEK293TC, and NIH3T3 cells, and they were used for some figures (regular or EV figures). I agree that NE4C cells are most relevant to neural tube cells, but do not understand use of H9C2 cells. It appears that HEK293TC or NIH3T3 cells were used simply for a cell culture bioassay in the revised manuscript, but why the neural precursor NE4C cell line was not used is not clear. Explanation and justification of H9C2, HEK293TC, and NIH3T3 cells should be provided, or else, only 1 cell type should be used.
22. Revised format of Fig 3D, to align with Fig 3A-C (as well as cell line substitution) is acceptable.
23. From prior review: “Fig. 3 and subsequent, it's not clear at all how ROS were assayed, and what they were expressed relative to.” The authors clarified how ROS were assayed (DHE fluorescence) in this revised version, and expressed relative to controls (set at 100%). The legend to Fig 3 indicates that DHE fluorescence was assayed, and the Materials and Methods states that DHE was used to measure superoxide levels. “ROS” refers to a category of reactive molecules, not all of which are generated by the same stressors, and there is no one assay to measure all of them. Therefore, the Y-axes of such graphs should be labeled “relative superoxide levels” or something similar (just as Y-axes for Western blots are not labeled as “relative molecule levels”).
24. Revision of figure legends so that %HTL, Hcy, N-Hcy levels and their normalization can be understood is acceptable.
25. From prior review: “Fig. S2C needs more controls to validate the anti- N-Hcy antibody, such as an anti-BSA ab, and secondary ab alone.” The authors responded that they “further validated the N-Hcy antibody using anti-BSA antibodies... These results were added to the revised Appendix Fig. S2A.” Although these results were added to the revised Supplement, they still did not show results using secondary ab alone. Since there is high background detection of BSA with the N-Hcy ab, the validity and specificity of the antibody are questionable.
26. Methods to assay SOD activities added to the revised manuscript are acceptable.
27. From prior review: “‘Hcy-mimetic’ meaning isn't clear in the figure 4 legend. Also, the legends don't make it clear which assays were performed in vitro and which in vivo.” I inferred what the authors meant by “Hcy-mimetic” in the original manuscript. My point is that it is not a proper, universally understood scientific term, and may not be understood by readers. Furthermore, “mimetic” is not correct, because if the mutants do not have the same activities as the wild-type proteins, they are not “mimetic”. I still find the text in the revised figure legend unacceptable. “Unmodifiable K->W mutants” (or specify the modification that cannot occur on tryptophans) may be more appropriate. The authors’ use of “in vitro” and “extro vivo” are also not conventional, and still are not clear. “In solution” might be used when the authors mean “in tubes (in vitro)”. When treating cultured cells, it is not necessary to state “extro vivo” (which is not correct, anyway).
28. From prior review: “The legend for Fig. 5B says Hcy, not HTL. Do the authors mean “transcription of” or “transactivation by”?” This comment was meant to point out 2 problems. The authors have changed “Hcy” to “HTL” in the legend for Fig 5B. This is acceptable. However, they changed “transcription” to “transactivation”, but not “of” to “by”. “transactivation of” (P 10) is not correct, and is not acceptable.
29. From prior review: “The Topflash assay needs a description and/or reference.” The authors have added Topflash (and ‘Fopplash’ [sic]) assay methods to the Materials and Methods. However, they provide no citations or sources for the Topflash or Fopflash plasmids, or transfection conditions, that would allow critical evaluation or replication of the experiments.
30. From prior review: “More details need to be provided for methods for entire Fig. 6 and Fig. S7.” The authors responded that they provided the details of the methods in the revised methods. Fig. S7

was deleted from the revised manuscript, so I only evaluated the methods for Fig. 6. There is insufficient detail (or references cited for methods) for LC-MS, Western blot (as well as for Western blots in other figures), and relative ROS levels. (As noted for other figures, do they mean DHE fluorescence or superoxide levels? If so, Y-axes should be labeled accordingly.) There is no quantitation of Fig. 6G or Fig. EV3D (and given the large number of bands and my concerns stated above regarding the validity of the N-Hcy antibody, I'm not sure it is possible). Therefore, do not agree with the statement in the Results, "AHT dose-dependently inhibited N-Hcy levels (Fig 6G and Fig EV3D)."

31. Revision of sentence "The abilities of AHT to reverse Hcy effects were further tested." Is acceptable. 32. The authors apologized for the confusion from the sentence, "An all-trans retinoic acid (ATRA) NTDs-complicated pregnancy rat model, which is thought to induce NTDs in rats partly by increasing Hcy (Xu et al, 2010) and ROS concentrations(Mantymaa et al, 2000; Mathieu et al, 2005)". They revised the sentence and provided new references. However, one of them (Zhang, et al., 2009) is not cited in the References.

33. From prior review: "Fig. S7C is of poor quality. One can't see spina bifida in most of the fetuses, even upon zooming in." The authors responded that "Spina bifida pictures with higher qualities were provided in the revised Fig EV3C." I think they mean Fig EV4C. The images of spina bifida and exencephaly are acceptable.

34. I felt that the Discussion could be significantly shortened (from 2.5 pages to 1-1.5 pages). The Discussion is now 2 pages, which is OK.

Other:

It is not clear what is meant by "EV" figures and why they are not combined into the other figures or added to the Supplement. Although I did not comment on typographical and grammatical errors, I agree with Reviewer 3 that "The manuscript may require professional editing service for typos or grammatical issues." The authors replied, "Following the reviewer's suggestion, we asked a native English speaker who is a professor at Balor [sic] Medical School polished the English of our manuscript." (In addition to misspelling "Baylor", this sentence is not grammatically correct!) The typos, missing words, and grammatical errors are still excessive in this revised manuscript. This makes it difficult for reviewers to focus on the science, rather than be distracted by trying to infer the authors' intent.

Referee #3 (Comments on Novelty/Model System for Author):

A chicken model of hyperhomocysteinemia has been incorporated in the revised study.

Referee #3 (Remarks for Author):

The major concerns have been adequately addressed in the revised manuscript. The authors have made impressive efforts and carried out substantial revision experiments, including the incorporation of a chicken model of hyperhomocysteinemia for in vivo studies and the replacement of a neuronal cell line for most in vitro studies, which all support their conclusions for the roles of MARSs in hyperhomocysteinemia-associated neural tube defects and congenital heart defects.

The remaining minor concerns include the seemingly high background or non-specific staining of immunofluorescence for MARS and other markers; unmatched color for merged images, (e.g. Figure 1G should be yellow for red/green doubly labeled cells, and the DNA staining can be removed in 1G as it masked the double labelling cells); and numerous typos and grammatical issues throughout the revised manuscript.

2nd Revision - authors' response

1 August 2019

***** Reviewer's comments *****

Referee #1 (Remarks for Author):

The authors have addressed my previous concerns.

Response: We thank the reviewer for supporting the publication of our manuscript.

Referee #2 (Remarks for Author):

The authors have made many revisions to the manuscript, including additional cell culture studies using one, most relevant cell line, additional chick embryo experiments, details of experimental methods, and revised figures and legends. However, some concerns that were raised in the first review still remain, and some of the new or revised text raise additional concerns and would need to undergo further revision.

Response: We thank the reviewer for acknowledging our revising efforts and according to his comments, we further revised our manuscript.

Comment 1. From previous review: “Although inhibition of MARSs with a small molecule reduced (but did not prevent) NTDs in some of the embryos of ATRA-treated rats, there may be some causes (genetic or environmental) of NTDs that are not mediated by MARS activities and in which MARSs inhibition would be ineffective. Therefore, the title of the manuscript [**Inhibiting Methionyl-tRNA synthetase prevents neural tube defects**] seems over stated and should be revised.” The authors responded that they “fully agree”, but revised the title to “**Inhibiting Methionyl-tRNA synthetase reduces neural tube defect and congenital heart defect onsets**”. They changed “prevents” to “reduces”, but missed the point that there may be some causes of NTDs that are not mediated by MARS activities and in which MARSs inhibition would be ineffective. And, they added another congenital malformation that, in addition to NTDs, was related to MARS in their subjects. So, while “reduces” may be more conservative than “prevents”, the title is even more overstated.

Response : We thank the reviewer for the comment. We have changed the title to “**Inhibiting MARSs reduces hyperhomocysteinemia-associated neural tube and congenital heart defects**” to better reflect the content of our study.

Comment 2 From previous review: “there should have been an increase in CHDs in the embryos of ATRA-treated rats, More information needs to be provided regarding effects of ATRA and AHT (the MARS inhibitor) on fetal CHDs and on litter sizes (because fetuses with CHDs might die before E18.5), and what effects, if any, ATRA has on MARSs activities in embryonic heart.” The authors responded that they did observe smaller litter sizes (revised Table EV2), however, they did not perform statistics of litter sizes or resorptions.

Response : We have provided the statistics of live fetuses and resorption in Table EV2 of revised manuscript.

They showed examples of CHDs (revised Fig 7F), but did not show quantitation.

Response : We have added the quantitation of each kind of CHDs in Table 4 and Table 6 of the revised manuscript.

They performed immunofluorescence for MARS (Fig 7D), but this is not quantitative (and differences between ATRA+/-AHT are not obvious) and shows protein levels, not enzyme activities.

Response : We thank the reviewer for pointing out our inappropriate presentation of data.

1, We previously presented the quantitative data of Fig 7D in Fig. 7C. This was certainly confusion. We had put the immunofluorescence data as Fig. 7C and put the corresponding quantitative data as Fig. 7D during this revision. Albeit the staining was not obvious by bare eye, instrumental quantitation showed significant differences (Fig. 7D).

2, We not only detected the protein levels but also detected enzyme activities of MARS on transforming Hcy to HTL. It has been reported that HTL added to plasma disappears within~ 1 h(Jakubowski, 1999), 25-fold faster than expected from the rate of spontaneous Hcy-thiolactone hydrolysis (Jakubowski, 1997). After 3 h most of the HTL is incorporated covalently into protein (Jakubowski, 1997; Jakubowski, 1999). So we detected the enzyme activities of MARS on transforming Hcy to HTL by immunofluorescence for N-Hcy, since HTL react with proteins immediately.

Comment 3. From previous review: “The results in the rat model are consistent with the hypothesis, but it cannot be concluded that MARSs effects on SOD1 or 2 activities are due to ROS, leading to NTDs unless it is shown that antioxidants (vitamin E or a glutathione precursor) have the same effect to reduce NTDs as AHT in a group of ATRA-treated rats.” The authors responded that they

had shown that N-acetylcysteine (NAC) [which can be used as a glutathione precursor, although “NAC” is never defined, and a reference that it is a GSH precursor or can be used as an antioxidant is not provided in the revised manuscript but had not shown the results in the original manuscript. However, effects of NAC on ROS levels in E18.5 fetal brain (revised Fig. EV4F) are not quantitative and are well after formation of the neural tube, so are not relevant. Effects of NAC on ATRA-induced NTDs (revised Table EV2) are acceptable. Effects of NAC in Figure 7 and Tables 4, 5, and 6 are acceptable.

Response : We fully agree with the reviewer’s points. Following the reviewer’s suggestions, the following modifications were made.

- 1, A reference in which NAC was reported as a GSH precursor (Issels et al, 1985) was added.
- 2, E10.5 fetal rats were generated and stained by DHE, quantitation showed that NAC successfully decreased ROS levels in E10.5 fetal brain as it did in E18.5 fetal brain as we previously found. The E10.5 fetal brain results with quantification were used in the revised manuscript to replace previous E18.5 fetal brain results (Figure EV4E).

Comment 4. From previous review: “Abstract, line 6: “Increased copy numbers of cytosolic-MARS and/or mitochondrial-MARS2” should be followed by “-encoding genes”. Please take care to italicize when referring to genes.” The authors said that they made these changes, but they did not.

Response : Thanks for the reviewer’s comment. Due to the length limit of abstract (175 words), there was no space to insert “-encoding genes”. So we change the font of “MARS and MARS2” to italic when referring to genes in the abstract, and added “-encoding genes” in the revised results, discussion, “The paper explained” and “Synopsis” sections.

Comment 5. From previous review: “Same sentence: “were detected in blood samples of patients with neural tube defects (NTDs) and congenital heart defects (CHDs)”. It should be stated in how many patients out of how many screened.” The authors said that they indicated the numbers of NTDs and CHDs in the revised manuscript, but they did not. They do not have access to the total number of patients screened, which is a weakness, as the prevalence of these defects in the population seen cannot be determined from either the Abstract or the Materials and Methods.

Response : We have provided this information in the material and methods and supplemental Table S1 of the 1st revised version. Due to the length limit of abstract, we didn’t indicate the number of NTDs and CHDs patients in the abstract.

Comment 6. From previous review: “Introduction, first sentence, Braly & Holford, 2003 is not a retrievable reference. The claim in the first sentence is not what the title of the reference says. It is advised to state what Hcy disorders are associated with Hcy (too much or too little?) and to provide valid references.” The authors have deleted the sentence and the reference, which is acceptable. However, they added a new clause and references to the second sentence, “Hyperhomocysteinemia prevails as high as 66.4% to 76.2% in elderly women and men, respectively (Janson et al, 2002),” which is irrelevant. Elderly women and men can’t be pregnant, and it cannot be determined if these incidences of hyperhomocysteinemia are due to nutrition, genetics, aging, etc. This sentence and reference detracts from the focus of the manuscript, which is on specific birth defects.

Response: Thanks for the reviewer’s comment. We have deleted “prevails as high as 66.4% to 76.2% in elderly women and men, respectively (Janson et al, 2002)” and the reference in the revised manuscript.

Comment 7. Revision of “nutritional disorder” to “metabolic disorder” is acceptable.

Response: We thank the reviewer for agreeing with our revision.

Comment 8. Replacement of “Insufficient development” with “maldevelopment” is acceptable.

Response: We thank the reviewer for agreeing with our revision.

Comment 9. From previous review: “P 4: “Being proteogenic enzymes that charges methionyl-tRNA, MARS and MARS2 expression can be elevated in proliferating cells, including actively differentiating embryonic cells and cancer cells, the latter having been confirmed in human colon cancer (Kushner et al, 1976).” References need to be provided for proliferating and embryonic cells.” The need for references, that had not been cited, was apparently due to original sentence structure, because, in fact, the authors meant (P 4, revised manuscript), “As MARS play critical roles in protein synthesis, we suspected that MARS and MARS2 expression can be elevated during embryo development.” So, the expression of MARS and MARS2 in embryonic cells was not known.

But, according to the added sentence, elevation of MARS and MARS2 expression occurs in NORMAL embryo development. So, shouldn't further increase due to gene duplication be protective of embryo development? This added sentence seems to contradict the hypothesis.

Response: We thank the reviewer for his/her thoughtful question. Elevated MARSs in developing tissues is helpful to protein synthesis that is required for development, however, the elevated MARSs expression will have second effects other than protein synthesis, that is, enhances N-homocysteinylation. Under normal conditions, the detrimental effects of N-homocysteinylation may be tolerated. In the case of *MARS* and *MARS2* duplication, the detrimental effects are beyond the plasticity of cells and thus inhibit cell growth and cause birth defects. Therefore, our conclusion is coherent.

Comment 10. From previous review: “Notably, *MARS* and *MARS2* gene duplication had been associated with increased ROS (Bayat et al, 2012), a consistent finding in NTDs (Weksler-Zangen et al, 2003), I think the authors mean that increased ROS are a consistent finding in NTDs. However, this is not true (there can be causes of NTDs independent of increased ROS), and this not an appropriate reference for this claim. Diabetes can increase ROS, and maternal diabetes-induced ROS can induce NTDs, but this paper did not show this.” The authors have revised the second part of the sentence and replaced the Weksler-Zangen reference: “Oxidative stress-induced aberrant apoptosis may results in NTDs (Covarrubias et al, 2008) and CHDs (Wang et al, 2015) and is believed to be the pathologically causal mechanism of hyperhomocysteinemia.” However, Covarrubias, et al., is mostly about the normal effects of ROS in development, not NTDs, and Wang, et al., is one (but not the original) reference for ROS in diabetes-CHDs.

Response: Thanks for the comment. We have replaced the references in this sentence “Oxidative stress-induced aberrant apoptosis may results in NTDs (Covarrubias et al, 2008) and CHDs (Wang et al, 2015)” to “Oxidative stress-induced aberrant apoptosis may results in NTDs(Yang et al, 2008) and CHDs (Morgan et al, 2008).”

The latter part of the sentence “and is believed to be the pathologically causal mechanism of hyperhomocysteinemia” is not referenced and is not true.

Response: We have deleted “and is believed to be the pathologically causal mechanism of hyperhomocysteinemia” in the sentence.

Comment 11. Deletion of part of sentence (citing Trachootham) is acceptable.

Response: We thank the reviewer for agreeing with our revision.

Comment 12. From prior review: “‘that is associated with increased apoptosis’ needs a reference. If Dhalla, et al., preceding the clause is the reference, the sentence should be rewritten. However, it should be noted that if ROS-induced apoptosis is involved in cancer, were true in cancer, cells would die, not thrive.” The authors replied that they “changed this sentence to ‘Oxidative stress-induced aberrant apoptosis may results in NTDs (Covarrubias et al, 2008) and CHDs (Wang et al, 2015)’ and provided the relative reference [sic]”. However (see Comment #10), this sentence and references are not acceptable.

Response: Thanks for the comments. We have replaced the references in this sentence “Oxidative stress-induced aberrant apoptosis may results in NTDs (Covarrubias et al, 2008) and CHDs (Wang et al, 2015)” to “Oxidative stress-induced aberrant apoptosis may results in NTDs(Yang et al, 2008) and CHDs (Morgan et al, 2008)”

Comment 13. Deletion of “known to be induced by HHcy” is acceptable.

Response: We thank the reviewer for agreeing with our revision.

Comment 14. Response to whether amplified copies of MARS or MARS2 carry sequence mutations and revision in text is acceptable.

Response: We thank the reviewer for agreeing with our revision.

Comment 15. Response to ““The increased copy number of MARSs in NTDs and CHDs indicated...” is acceptable.

Response: We thank the reviewer for agreeing with our revision.

Comment 16. From prior review: “P 7: How was growth of neuronal NE4C stem cells measured? What are the units?” The authors have described how growth was assayed, but the units (“Cell Index” in the revised manuscript) are not defined.

Response: Thanks for the reviewer’s comment. We have added the definition of Cell Index, of which the impedance of electron flow caused by adherent cells is reported using a unitless parameter called Cell Index (CI), where $CI = (\text{impedance at time point } n - \text{impedance in the absence of cells}) / \text{nominal impedance value}$ to the methods section. Cell index responds to changes in cell number and cell adhesion. The more cells there are, the larger the cell index is.

Comment 17. From prior review: “How was BCL2 over-expressed? How was prevention of apoptosis determined?” The authors replied “BCL2 was overexpressed by transfecting BCL2-FLAG plasmids.” However, there is no information about the source or construction of the plasmid or transfection conditions in either the figure legends or Materials and Methods to allow critical evaluation or replication of the experiments.

Assay of apoptosis is described (by cleavage of BCL2, according to the authors’ response, and by flow cytometry or TUNEL assay, according to the revised figure legends or Materials and Methods). While flow cytometry and TUNEL methods are adequately described, assay of BCL2 cleavage is not adequately described or referenced to allow critical evaluation of the data or replication of the experiments. Even though BCL2 cleavage is an established method, it is essential that this information be provided.

Response: 1, Mouse Bcl-2 cDNA (NM_009741.5) cloned in pcDNA3.1-Flag vector was a gift from lab of Dr. Jiahui Han at Xiamen University, China. 5 μ g pcDNA3.1-Flag-Bcl-2 plasmids were transfected in 1×10^6 NE4C cells using Lipofectamine® 3000 transfection reagent (L30075, Thermo Fisher, Waltham, MA, USA). Production of Bcl-2 proteins was confirmed by Western blot analysis using anti-Flag antibody. This information had been added to the revised methods and Appendix Table S7.

2, We appreciate the reviewer’s suggestion. The description and references of Bcl-2’s effects on cleaved caspase3 (Swanton et al, 1999) and cleaved-caspase3 on evaluating apoptosis (Erhardt & Cooper, 1996) were provided in the 2nd revised results (Fig. 2B) and methods.

Comment 18. Replacement of “showing” with “suggesting” is acceptable.

Response: We thank the reviewer for agreeing with our revision.

Comment 19. From prior review: “How was MARS or MARS2 over-expressed? How much siRNA was transfected?” The authors responded that MARS and MARS2 cDNA were cloned into pRK7-Flag vector, and described the lipofection reagent used. However, the amount of plasmid transfected was not described in the manuscript. The amount of siRNA, or how transfected were not described in the methods section of the revised manuscript, contrary to the authors’ response. Nor were the siRNA their sequences, or how long cells were transfected before assay (which I did not ask for), which are important for assessment and replication of experiments.

Response: 1, We transfected 5 μ g MARS/MARS2 plasmids in 1×10^6 cells. We added this information in the 2nd revised methods.

2, We are sorry that we misunderstood the reviewer’s previous suggestion (see previous *Comment 22 (Why is the format of Fig. 3D different from A-C? What do #1, 2, and 3 refer to?)*). In the siMARS/siMARS2 experiment, we transfected 30pmol siRNA in 1×10^6 cells, the cells were harvest 48h after transfection. The siRNA sequences were shown as below. We have replaced the 3D in the revised manuscript with a shRNA result to keep the format consistent with 3A-C.

Table, MARS and MARS2 siRNA sequences

siRNA Sequences	
siMARS-#1	UUA CAC UGA GGC UUC UUA AGC UGG A
	UCG AGC UUA AGA AGC CUC AGU GUA A
siMARS-#2	UUU CUA GGC CCU UCU CCC AAG CAG U
	ACU GCU UGG GAG AAG GGC CUA GAA A
siMARS-#3	AAC UGA UAC AGG UCC ACU UGC UCU G
	CAG AGC AAG UGG ACC UGU AUC AGU U

siMARS2-#1	GUG CUA CAA GGG CGU CUA UTT
	AUA GAC GCC CUU GUA GAG CTT
siMARS2-#2	GCU AUA CCG UGG AUG GCU UTT
	AAG CCA UCC ACG GUA UAG CTT
siMARS2-#3	CCA AAU GCU GAG UUC AAA UTT
	AUU UGA ACU CAG CAU UUG GTT
Negative control siRNA	UUC UCC GAA CGU GUC ACG UTT
	ACG UGA CAC GUU CGG AGA ATT

Comment 20. Revision of “Data showed that oxidative stress related pathways were enriched” to “oxidative stress-related pathways were among the pathways that were enriched.” is acceptable.

Response: We thank the reviewer for agreeing with our revision.

Comment 21. One of the other reviewers, and I, questioned switching between HEK293 and NE4C cells. The authors repeated many of the cell culture experiments using NE4C cells, which is appropriate. However, both the Materials and Methods and the Supplement refer to NE4C, H9C2, HEK293TC, and NIH3T3 cells, and they were used for some figures (regular or EV figures). I agree that NE4C cells are most relevant to neural tube cells, but do not understand use of H9C2 cells. It appears that HEK293TC or NIH3T3 cells were used simply for a cell culture bioassay in the revised manuscript, but why the neural precursor NE4C cell line was not used is not clear. Explanation and justification of H9C2, HEK293TC, and NIH3T3 cells should be provided, or else, only 1 cell type should be used.

Response: We appreciate the reviewer’s comments. As the reviewer correctly pointed out, NE4C cells were used because they are relevant to neural tube cells. H9C2 is rat embryonic cardiomyoblast cell line thus may be relevant to heart development and CHDs. We fully agree that all bioassays should be performed in one cell line. However, it was more feasible to carry out some bioassays, such as affinity protein purification (SODs, Fig 4C and D; MARSs, Fig 6B and C), in HEK293T cells, because this cell line has high transfection and expression efficiency. We realized that it’s not proper to use NIH3T3, so we deleted the data in NIH3T3 (1st revised Fig. EV3D), and confirm AHT’s effect in H9C2 (2nd revised Fig. EVD-F). Following the reviewer’s suggestion, we have added an explanation of cell line applications in the 2nd revised methods.

Comment 22. Revised format of Fig 3D, to align with Fig 3A-C (as well as cell line substitution) is acceptable.

Response: We thank the reviewer for agreeing with our revision.

Comment 23. From prior review: “Fig. 3 and subsequent, it's not clear at all how ROS were assayed, and what they were expressed relative to.” The authors clarified how ROS were assayed (DHE fluorescence) in this revised version, and expressed relative to controls (set at 100%). The legend to Fig 3 indicates that DHE fluorescence was assayed, and the Materials and Methods states that DHE was used to measure superoxide levels. “ROS” refers to a category of reactive molecules, not all of which are generated by the same stressors, and there is no one assay to measure all of them. Therefore, the Y-axes of such graphs should be labeled “relative superoxide levels” or something similar (just as Y-axes for Western blots are not labeled as “relative molecule levels”).

Response: Thanks for the reviewer’s suggestion. We have changed the Y-axis labels in DHE detecting experiments to “Relative superoxide levels”.

Comment 24. Revision of figure legends so that %HTL, Hcy, N-Hcy levels and their normalization can be understood is acceptable.

Response: We thank the reviewer for agreeing with our revision.

Comment 25. From prior review: “Fig. S2C needs more controls to validate the anti- N-Hcy antibody, such as an anti-BSA ab, and secondary ab alone.” The authors responded that they “further validated the N-Hcy antibody using anti-BSA antibodies... These results were added to the revised Appendix Fig. S2A.” Although these results were added to the revised Supplement, they still did not show results using secondary ab alone. Since there is high background detection of BSA with the N-Hcy ab, the validity and specificity of the antibody are questionable.

Response: Thanks for the comment. We have supplied the results using secondary antibodies alone in the 2nd revised Appendix Figure S2A. The high background detection of BSA with the N-Hcy

antibodies in the 1st revised Appendix Fig. S2A may results from the light N-Hcylation of BSA, which due to the short reaction time (30min) and neutral pH (pH=7.0) of BSA and HTL. When the reaction condition changed to 12h, pH8.0, the N-Hcy levels were more heavily (indicates by strong signal and molecular weight shift) and much stronger than the background detection of BSA.

Comment 26. Methods to assay SOD activities added to the revised manuscript are acceptable.

Response: We thank the reviewer for agreeing with our revision.

Comment 27. From prior review: “‘Hcy-mimetic’ meaning isn't clear in the figure 4 legend. Also, the legends don't make it clear which assays were performed in vitro and which in vivo.” I inferred what the authors meant by “Hcy-mimetic” in the original manuscript. My point is that it is not a proper, universally understood scientific term, and may not be understood by readers. Furthermore, “mimetic” is not correct, because if the mutants do not have the same activities as the wild-type proteins, they are not “mimetic”. I still find the text in the revised figure legend unacceptable. “Unmodifiable K->W mutants” (or specify the modification that cannot occur on tryptophans) may be more appropriate.

Response: Thanks for the suggestion. We have changed the Hcy-mimetic to “Unmodifiable K->W mutants” in the revised manuscript.

The authors’ use of “in vitro” and “extro vivo” are also not conventional, and still are not clear. “In solution” might be used when the authors mean “in tubes (in vitro)”. When treating cultured cells, it is not necessary to state “extro vivo” (which is not correct, anyway).

Response: Thanks for the suggestion. We have changed the “in tubes” to “in solution”, and deleted the “extro vivo”.

Comment 28. From prior review: “The legend for Fig. 5B says Hcy, not HTL. Do the authors mean “transcription of” or “transactivation by”?” This comment was meant to point out 2 problems. The authors have changed “Hcy” to “HTL” in the legend for Fig 5B. This is acceptable. However, they changed “transcription” to “transactivation”, but not “of” to “by”. “transactivation of” (P 10) is not correct, and is not acceptable.

感谢: The authors have changed “Hcy” to “HTL” in the legend for Fig 5B. This is acceptable.

Response: Sorry for our mistake. We have changed “the transactivation of T-cell factor (TCF)” to “activities of β -catenin-dependent Tcf transcription factors”.

Comment 29. From prior review: “The Topflash assay needs a description and/or reference.” The authors have added Topflash (and ‘Fopflash’ [sic]) assay methods to the Materials and Methods. However, they provide no citations or sources for the Topflash or Fopflash plasmids, or transfection conditions, that would allow critical evaluation or replication of the experiments.

Response: Thanks for the suggestion and sorry for our unclear description. The Topflash plasmid was a gift from Dr. Tao Zhong (Fudan University, China) (Ni et al, 2011). The Fopflash plasmids were purchased from Millipore. 2.5 μ g Topflash and 2.5 μ g Fopflash were transfected in 1×10^6 NE4C cells using Lipofectamine® 3000 transfection reagent (L30075, Thermo Fisher) for 48h. These details had been added in the 2nd revised methods and Appendix Table S7.

Comment 30. From prior review: “More details need to be provided for methods for entire Fig. 6 and Fig. S7.” The authors responded that they provided the details of the methods in the revised methods. Fig. S7 was deleted from the revised manuscript, so I only evaluated the methods for Fig. 6. There is insufficient detail (or references cited for methods) for LC-MS, Western blot (as well as for Western blots in other figures), and relative ROS levels. (As noted for other figures, do they mean DHE fluorescence or superoxide levels? If so, Y-axes should be labeled accordingly.) There is no quantitation of Fig. 6G or Fig. EV3D (and given the large number of bands and my concerns stated above regarding the validity of the N-Hcy antibody, I’m not sure it is possible). Therefore, do not agree with the statement in the Results, “AHT dose-dependently inhibited N-Hcy levels (Fig 6G and Fig EV3D).”

Response: 1, Sorry for our unclear description. According to the reviewer’s comments, we added more details and references in the LC-MS and western blot in the 2nd revised methods.

2, We have added a reference which reports DHE can be used to detect intracellular superoxide formation in endothelial cells and intact tissues (Erhardt & Cooper, 1996) in the 2nd revised methods. We have changed the Y-axes to “Superoxide levels”.

3, We have added the quantitation data of Fig. 6G and Fig. EV3D, which were provided as 2nd revised Fig. 6H and Fig. EV3E. We cultured the cells in DMEM without serum to avoid the signal of BSA detected by the N-Hcy antibody.

4, We have deleted the “dose-dependently”.

Comment 31. Revision of sentence “The abilities of AHT to reverse Hcy effects were further tested.” Is acceptable.

Response: We thank the reviewer for agreeing with our revision.

Comment 32. The authors apologized for the confusion from the sentence, “An all-trans retinoic acid (ATRA) NTDs-complicated pregnancy rat model, which is thought to induce NTDs in rats partly by increasing Hcy (Xu et al, 2010) and ROS concentrations (Mantymaa et al, 2000; Mathieu et al, 2005)”. They revised the sentence and provided new references. However, one of them (Zhang, et al., 2009) is not cited in the References.

Response: Thanks for the comment. We have cited the reference (Zhang, et al., 2009) in the 2nd revised manuscript.

Comment 33. From prior review: “Fig. S7C is of poor quality. One can't see spina bifida in most of the fetuses, even upon zooming in.” The authors responded that “Spina bifida pictures with higher qualities were provided in the revised Fig EV3C.” I think they mean Fig EV4C. The images of spina bifida and exencephaly are acceptable.

Response: Sorry for our mistake. Spina bifida pictures with higher qualities were provided in the revised Fig EV4C.

Comment 34. I felt that the Discussion could be significantly shortened (from 2.5 pages to 1-1.5 pages). The Discussion is now 2 pages, which is OK. Other: It is not clear was is meant by “EV” figures and why they are not combined into the other figures or added to the Supplement. Although I did not comment on typographical and grammatical errors, I agree with Reviewer 3 that “The manuscript may require professional editing service for typos or grammatical issues.” The authors replied, “Following the reviewer’s suggestion, we asked a native English speaker who is a professor at Balor [sic] Medical School polished the English of our manuscript.” (In addition to misspelling “Baylor”, this sentence is not grammatically correct!) The typos, missing words, and grammatical errors are still excessive in this revised manuscript. This makes it difficult for reviewers to focus on the science, rather than be distracted by trying to infer the authors’ intent.

Response: 1. EMBO Press strongly encourages authors to select a limited number (typically 5) of supplementary figures for inclusion in the article proper as Expanded View (EV) figures in order to improve their accessibility, visibility and utility.

2. Thanks for the reviewer’s suggestion. We have sent the manuscript out for professional editing services.

Referee #3 (Comments on Novelty/Model System for Author):

A chicken model of hyperhomocysteinemia has been incorporated in the revised study.

Referee #3 (Remarks for Author):

The major concerns have been adequately addressed in the revised manuscript. The authors have made impressive efforts and carried out substantial revision experiments, including the incorporation of a chicken model of hyperhomocysteinemia for in vivo studies and the replacement of a neuronal cell line for most in vitro studies, which all support their conclusions for the roles of MARSs in hyperhomocysteinemia-associated neural tube defects and congenital heart defects.

Comment 1 The remaining minor concerns include the seemingly high background or non-specific staining of immunofluorescence for MARS and other markers;

Response : Thanks for the reviewer’s comment. We had tried to decrease the non-specific staining during our experimentations. As we all know, the quality of the staining is heavily dependent on the quality of the antibodies which was beyond our control. We had made efforts to make qualities of our staining comparable to the published ones such as β -Catenin(Li et al, 2015; Zechner et al, 2003; Zhao et al, 2014) , Ki67/MCM2(Ma et al, 2013; Sorrells et al, 2018; Wei et al, 2011; Yang et al, 2015) and TUNEL (Yang et al, 2015)

Comment 2 unmatched color for merged images, (e.g. Figure 1G should be yellow for red/green doubly labeled cells, and the DNA staining can be removed in 1G as it masked the double labelling cells);

Response : Thanks for the reviewer's comment. We have deleted the DNA staining (DAPI, blue) signals in the merged image of 2nd revised Fig 1G and H, Fig 7A and C, Fig EV1A and B.

Comment 3 numerous typos and grammatical issues throughout the revised manuscript.

Response : Thanks for reminding. We have sent the manuscript out again for professional editing services and made proofread by multiple person.

References

- Erhardt P, Cooper GM (1996) Activation of the CPP32 apoptotic protease by distinct signaling pathways with differential sensitivity to Bcl-xL. *The Journal of biological chemistry* **271**: 17601-17604
- Issels RD, Bourier S, Biaglow JE, Gerweck LE, Wilmanns W (1985) Temperature-dependent influence of thiols upon glutathione levels in Chinese hamster ovary cells at cytotoxic concentrations. *Cancer research* **45**: 6219-6224
- Jakubowski H (1997) Metabolism of homocysteine thiolactone in human cell cultures. Possible mechanism for pathological consequences of elevated homocysteine levels. *The Journal of biological chemistry* **272**: 1935-1942
- Jakubowski H (1999) Protein homocysteinylation: possible mechanism underlying pathological consequences of elevated homocysteine levels. *FASEB journal : official publication of the Federation of American Societies for Experimental Biology* **13**: 2277-2283
- Li BI, Matteson PG, Ababon MF, Nato AQ, Jr., Lin Y, Nanda V, Matise TC, Millonig JH (2015) The orphan GPCR, Gpr161, regulates the retinoic acid and canonical Wnt pathways during neurulation. *Developmental biology* **402**: 17-31
- Ma T, Wang C, Wang L, Zhou X, Tian M, Zhang Q, Zhang Y, Li J, Liu Z, Cai Y, Liu F, You Y, Chen C, Campbell K, Song H, Ma L, Rubenstein JL, Yang Z (2013) Subcortical origins of human and monkey neocortical interneurons. *Nature neuroscience* **16**: 1588-1597
- Morgan SC, Relaix F, Sandell LL, Loeken MR (2008) Oxidative stress during diabetic pregnancy disrupts cardiac neural crest migration and causes outflow tract defects. *Birth defects research Part A, Clinical and molecular teratology* **82**: 453-463
- Ni TT, Rellinger EJ, Mukherjee A, Xie S, Stephens L, Thorne CA, Kim K, Hu J, Lee E, Marnett L, Hatzopoulos AK, Zhong TP (2011) Discovering small molecules that promote cardiomyocyte generation by modulating Wnt signaling. *Chemistry & biology* **18**: 1658-1668
- Sorrells SF, Paredes MF, Cebrian-Silla A, Sandoval K, Qi D, Kelley KW, James D, Mayer S, Chang J, Auguste KI, Chang EF, Gutierrez AJ, Kriegstein AR, Mathern GW, Oldham MC, Huang EJ, Garcia-Verdugo JM, Yang Z, Alvarez-Buylla A (2018) Human hippocampal neurogenesis drops sharply in children to undetectable levels in adults. *Nature* **555**: 377-381
- Swanton E, Savory P, Cosulich S, Clarke P, Woodman P (1999) Bcl-2 regulates a caspase-3/caspase-2 apoptotic cascade in cytosolic extracts. *Oncogene* **18**: 1781-1787
- Wei B, Nie Y, Li X, Wang C, Ma T, Huang Z, Tian M, Sun C, Cai Y, You Y, Liu F, Yang Z (2011) Emx1-expressing neural stem cells in the subventricular zone give rise to new interneurons in the ischemic injured striatum. *The European journal of neuroscience* **33**: 819-830
- Yang P, Zhao Z, Reece EA (2008) Activation of oxidative stress signaling that is implicated in apoptosis with a mouse model of diabetic embryopathy. *American journal of obstetrics and gynecology* **198**: 130 e131-137

Yang SL, Yang M, Herrlinger S, Liang C, Lai F, Chen JF (2015) MiR-302/367 regulate neural progenitor proliferation, differentiation timing, and survival in neurulation. *Developmental biology* **408**: 140-150

Zechner D, Fujita Y, Hulsken J, Muller T, Walther I, Taketo MM, Crenshaw EB, 3rd, Birchmeier W, Birchmeier C (2003) beta-Catenin signals regulate cell growth and the balance between progenitor cell expansion and differentiation in the nervous system. *Developmental biology* **258**: 406-418

Zhao T, Gan Q, Stokes A, Lassiter RN, Wang Y, Chan J, Han JX, Pleasure DE, Epstein JA, Zhou CJ (2014) beta-catenin regulates Pax3 and Cdx2 for caudal neural tube closure and elongation. *Development* **141**: 148-157

3rd Editorial Decision

15 August 2019

Thank you for the submission of your revised manuscript to EMBO Molecular Medicine. We have now received the report from referee #2, who is supportive of publication of your study. I am therefore pleased to inform you that we will be able to accept your manuscript pending the following minor editorial amendments.

I look forward to reading a new revised version of your manuscript as soon as possible.

***** Reviewer's comments *****

Referee #2 (Remarks for Author):

This reviewer appreciates the significant work that the authors have put into revising the manuscript, including additional experiments. I have no more major concerns

3rd Revision - authors' response

16 December 2019

Authors made the requested editorial changes.

Corresponding Author Name: Shimin Zhao
 Journal Submitted to: EMBO Molecular Medicine
 Manuscript Number: EMM-2018-09469-V2